EMBO
Molecular Medicine

# ENOblock synergizes with colistin to treat *Acinetobacter baumannii* infections

Irene Molina Panadero [iD][1], Antonio Moreno Rodríguez[1], Angela Rey Hidalgo [iD][1], Mercedes de la Cruz[2], Pilar Sánchez[2], Laura Tomás Gallardo [iD][3], Thanadon Samernate[4], Milan Sencanski [iD][5], Sanja Glisic[5], Olga Genilloud[2], Poochit Nonejuie [iD][4], Antonio J Pérez-Pulido[1,6], Abdelkrim Hmadcha[6,7] & Younes Smani [iD][1,6 ✉]

## Abstract

**High-throughput screening studies provide an additional approach to discovering repurposed drugs for antimicrobial treatments. In this work, we report the identification of ENOblock, an anticancer drug, as an antimicrobial agent. We computationally and experimentally validated that ENOblock synergizes with colistin, the last resort antibiotic. Additionally, we identified enolase as the potential bacterial target for ENOblock. The in silico and in vitro antibacterial activity of ENOblock translated into potent in vivo efficacy in an animal infection model. Collectively, the preclinical data support the selection of ENOblock as a promising candidate for antimicrobial development, with the potential to address the urgent threat of infections caused by *Acinetobacter baumannii*.**

**Keywords** ENOblock; Enolase; Drug Repurposing; *Acinetobacter baumannii*
**Subject Categories** Microbiology, Virology & Host Pathogen Interaction; Pharmacology & Drug Discovery

## Introduction

Gram-negative bacteria (GNB) are a significant concern in healthcare settings due to their ability to cause a wide range of infections such as pneumonia, bloodstream infections, wound or surgical site infections, and meningitis (Morris and Cerceo, 2020), which are often difficult to treat. GNB infections, especially hospital acquired ones, represent a significant burden on healthcare systems worldwide, with reported costs of $136 million per year (Balasubramanian et al, 2023), requiring ongoing efforts in surveillance, infection prevention and control, antibiotic stewardship, and research into new treatment options to mitigate their impact.

One of the main difficulties in tackling GNB is their high efficiency in acquiring antimicrobial resistance (AMR) encoded by genomic, transcriptomic and proteomic changes (Yelin and Kishony, 2018). The challenge of AMR is further exacerbated by the alarming number of deaths attributed to bacterial AMR, coupled with the urgent threat posed by the declining discovery and development of new antibiotics. The World Health Organization (WHO) and other European institutions have recently underscored the dangers posed by these infections (ECDC and WHO, 2023). Consequently, a perfect storm is converging regarding these infections: increasing antimicrobial resistance with a decreased new drug development (ECDC and WHO, 2023). This context is likely the best example of the purported "Post-Antibiotic Era", with relevance even in non-specialized media. It is clear that new policies and actions are necessary to avoid the forecasts for 2050 that attribute ten million deaths worldwide to antimicrobial resistance (O'Neill, 2016). This is particularly critical for pathogens such as *Acinetobacter baumannii* (Ab), *Escherichia coli* and *Klebsiella pneumoniae*, among others, which have shown a continuous increase in antimicrobial resistance and are responsible for a significant proportion of reported infections and associated deaths (ECDC and WHO, 2023).

Approaches using computational methods and high-throughput screening (HTS) have recently been developed for antibiotic discovery (Wong et al, 2024; Boulaamane et al, 2024; Wan et al, 2024; Miethke et al, 2021). For example, screening small-molecule libraries has revealed new antimicrobial agents that belong to existing or new antibiotic classes (Bakker et al, 2024; Blasco et al, 2024; Zampaloni et al, 2024). Recently, HTS studies have been developed to discover repurposed drugs for antimicrobial treatments (Huang and Zhang, 2022).

Within this context, increasing attention has been given to targeting enzymes. One of the most promising targets to emerge is enolase a highly conserved metalloenzyme that plays a crucial role in the glycolytic pathway, catalyzing the reversible conversion of 2-phosphoglycerate into phosphoenolpyruvate (Pancholi, 2001).

[1]Centro Andaluz de Biología del Desarrollo, Universidad Pablo de Olavide/CSIC/Junta de Andalucía, Seville 41013, Spain. [2]Fundación MEDINA, Parque Tecnológico Ciencias de la Salud, Granada 18016, España. [3]Proteomics and Biochemistry Platform. Centro Andaluz de Biología del Desarrollo. Universidad Pablo de Olavide/CSIC/Junta de Andalucía, Seville 41013, Spain. [4]Institute of Molecular Biosciences, Mahidol University, Salaya, Nakhon Pathom 73170, Thailand. [5]Laboratory for Bioinformatics and Computational Chemistry, Institute of Nuclear Sciences VINCA, National Institute of the Republic of Serbia, University of Belgrade, Belgrade 11001, Serbia. [6]Departamento de Biología Molecular e Ingeniería Bioquímica, Universidad Pablo de Olavide, Seville 41013, Spain. [7]Biosanitary Research Institute (IIB-VIU), Valencian International University (VIU), Valencia 46021, Spain. ✉E-mail: ysma@upo.es

This enzyme is found across all domains of life, including both prokaryotic and eukaryotic organisms, where it contributes not only to central metabolism but also, in many cases, to non-glycolytic functions such as cell surface adhesion, plasminogen binding, and stress response (Pancholi, 2001). In pathogenic bacteria, enolase is often identified on the cell surface, where it may facilitate host colonization and immune evasion (Ehinger et al, 2004). Furthermore, enolase has been implicated in maintaining membrane dynamics and virulence of several bacterial species (Ayón-Núñez et al, 2018; Liu et al, 2021).

Specific enolase inhibitors, such as 2-aminothiazoles, disrupt bacterial ATP production and viability in *Mycobacterium tuberculosis* (Wescott et al, 2018). Three tropolone derivatives showing 53–78% enolase inhibition displayed antibacterial activity against major Gram-negative pathogens (*A. baumannii, E. coli, Pseudomonas aeruginosa*, and *K. pneumoniae*) with MICs of 11.3–45.2 mg/L, and induced filamentation in *E. coli*, suggesting effects on cell wall biosynthesis or division (Krucinska et al, 2019a). The natural inhibitor SF2312, from *Micromonospora*, showed limited activity (MIC from 50 to >400 mg/L) but improved potency against *E. coli* and *Staphylococcus aureus*, but not against *A. baumannii* and *P. aeruginosa*, when glucose-6-phosphate was added (Krucinska et al, 2019b). Additionally, PEIP-expressing bacteria exhibited growth attenuation and thinner cell walls due to impaired peptidoglycan synthesis of *Bacillus subtilis* (Zhang et al, 2022). Moreover, ENOblock has been identified as an inhibitor of enolase with pleiotropic biological activities. Previous studies have demonstrated its anticancer effects, where ENOblock suppresses tumor progression by disrupting glycolytic metabolism (Jung et al, 2013). It has also shown anti-inflammatory properties by modulating cytokine expression and reducing macrophage activation (Polcyn et al, 2020). In addition, ENOblock influences metabolic regulation, including the attenuation of hyperglycemia and improvement of insulin sensitivity in diabetic models (Cho et al, 2017). The evolutionary conservation of these enzymes in multiple Gram-negative pathogens underscores its potential as a therapeutic target, while its multifunctional role in virulence further enhances its clinical relevance (Krucinska et al, 2019b).

In this work, we report the identification of ENOblock, an anticancer drug, as an antimicrobial agent. We computationally and experimentally validated that ENOblock synergizes with colistin, the last resort antibiotic. Additionally, we identified enolase as the potential bacterial target for ENOblock. The in silico and in vitro antibacterial activity of ENOblock translated into potent in vivo efficacy in the *Galleria mellonella* animal infection model. Collectively, these preclinical data could support the selection of ENOblock as a promising candidate for antimicrobial development with the potential to address the urgent threat of infections caused by Ab.

## Results

### High-throughput screening for repurposing drugs as antimicrobial agents

We developed and validated a high-throughput screen assay using 2464 bioactive compounds from the EU-OPENSCREEN pilot library, dispensed in 384-well plates at 100 μM final concentration, and the *A. baumannii* Ab ATCC 17978 and *E. coli* ATCC 25922 strains, as well as their respective MDR strains (Ab186 and MCR1+) (Fig. 1A). We identified 33 compounds (1.32% of the total compounds) with inhibitory activity of ≥70% against at least one or both MDR strains of Ab186 and *E. coli* MCR1+. Of these 33 compounds, 7 showed MICs of ≤100 μM against the reference and MDR strains of Ab (ATCC 17978 and Ab186) and *E. coli* (ATCC 25922 and MCR1+) (Fig. 1A). Among these 7 compounds, ENOblock (Fig. 1B) was chosen based on preliminary antimicrobial activity against *A. baumannii* and *E. coli* in our initial screens for further studies. This compound was active against the Ab ATCC 17978 and *E. coli* ATCC 25922 strains, with AC$_{50}$ values of 23.57 μM and 46.86 μM, respectively (Fig. 1C).

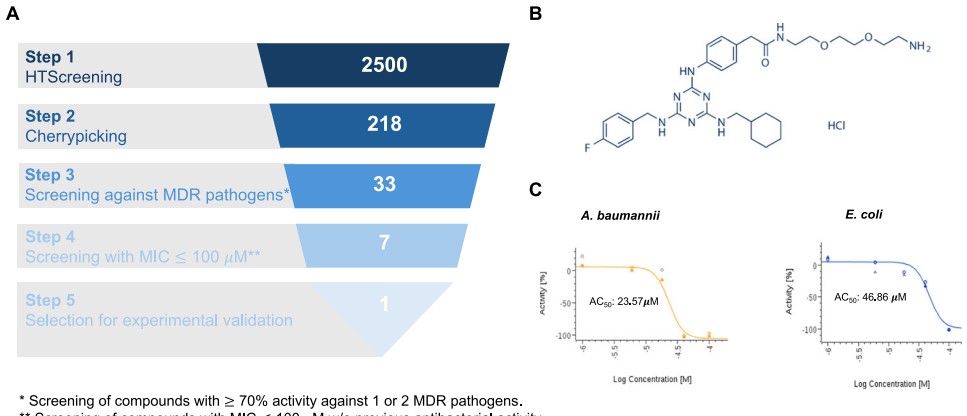

* Screening of compounds with ≥ 70% activity against 1 or 2 MDR pathogens.
** Screening of compounds with MIC ≤ 100 μM w/o previous antibacterial activity.

**Figure 1. HTS in antibiotic discovery.**

(A) Hit identification workflow using the EU-OPENSCREEN library. *Screening of compounds with ≥70% activity against 1 or 2 MDR pathogens. **Screening of compounds with MIC ≤100 μM w/o previous antibacterial activity. (B) Chemical structure of ENOblock. (C) Confirmation of ENOblock activity with the primary screening sample in dose–response mode against *A. baumannii* Ab ATCC 17978 and *E. coli* ATCC 25922 strains. AC$_{50}$: Concentration at which 50% of maximum activity is observed. Average values from two biological replicates are represented. Source data are available online for this figure.

**Table 1. Antibacterial activity of ENOblock in colistin-resistant *A. baumannii* strains.**

| Strains | Colistin resistance mechanism[c] | Colistin MIC (mg/L) | ENOblock MIC (mg/L) |
|---|---|---|---|
| #1[a] | pmrB | 256 | 16 |
| #10[a] | pmrB | >256 | 16 |
| #11[a] | pmrB | 256 | 16 |
| #14[a] | pmrB | 256 | 16 |
| #16[a] | pmrB | >256 | 16 |
| #17[a] | pmrB | 64 | 16 |
| #19[a] | pmrB | >256 | 32 |
| #20[a] | pmrB | >256 | 8 |
| #21[a] | pmrB | 256 | 8 |
| #22[a] | pmrB | >256 | 16 |
| #24[a] | pmrB | 64 | 16 |
| #99[a] | pmrB | >256 | 16 |
| #113[a] | pmrB | >256 | 16 |
| Ab CR17[b] | pmrA | 64 | 32 |
| MIC$_{50}$ | – | 256 | 16 |
| MIC$_{90}$ | – | >256 | 32 |

[a]From Valencia et al, 2009; [b]from López-Rojas et al, 2011b; [c]mutation in *pmrA* or *pmrB*.

**Table 2. Antibacterial activity of ENOblock in carbapenem-resistant *A. baumannii* strains.**

| Strains[a] | Carbapenem resistance mechanism[b] | Imipenem/ meropenem MIC (mg/L) | ENOblock MIC (mg/L) |
|---|---|---|---|
| #17 | OXA-58 | 8 | 16 |
| #37 | OXA-58 | 16 | 16 |
| #40 | OXA-24 | >64 | 16 |
| #53 | OXA-58 | 64 | 16 |
| #286 | ND | 16 | 16 |
| #288 | ND | 32 | 16 |
| #289 | ND | 8 | 16 |
| #295 | OXA-58 | 8 | 16 |
| #298 | OXA-58 | 16 | 32 |
| #299 | OXA-58 | 16 | 16 |
| #405 | OXA-58 | 32 | 32 |
| #410 | OXA-58 | 32 | 32 |
| #414 | OXA-58 | 32 | 32 |
| #416 | OXA-58 | 32 | 32 |
| #417 | OXA-58 | 16 | 16 |
| #440 | ND | 8 | 16 |
| #441 | ND | 8 | 16 |
| #448 | OXA-51, OXA-58 | 8 | 32 |
| MIC$_{50}$ | – | 16 | 16 |
| MIC$_{90}$ | – | 64 | 32 |

*ND* not determined.
[a]From GenBank Bioproject PRJNA422585; [b]carbapenemase.

## ENOblock is active against Ab

To confirm the activity of ENOblock against clinical isolates of Ab, ENOblock was tested against 14 and 18 colistin-resistant and carbapenem-resistant Ab isolates, respectively. The results of the MICs tests are displayed in Tables 1 and 2. The MICs ranged from 8 to 32 mg/L and 16 to 32 mg/L for ENOblock against colistin and carbapenem-resistant Ab, respectively. The reference strain Ab ATCC 17978 presents an ENOblock MIC of 8 mg/L. The MIC$_{50}$ and MIC$_{90}$ concentrations, which represent the lowest concentration of an antimicrobial agent that is required to inhibit the growth of 50 and 90% of the isolates tested, respectively, for ENOblock against colistin and carbapenem-resistant isolates were 16 and 32 mg/L, respectively. However, the MIC$_{50}$ and MIC$_{90}$ for colistin were 256 and >256 mg/L, and for carbapenems were 16 and 64 mg/L (Tables 1 and 2) Notably, the ENOblock MIC$_{90}$ (32 mg/L) is three times lower than the cytotoxic concentration 50% (CC$_{50}$) of ENOblock in HeLa (112.7 mg/L) and macrophage cells (121.3 mg/L) (Appendix Fig. S1), and the selective pressure by growing Ab ATCC 17978 strain in the presence of increasing ENOblock concentrations did not allow Ab to develop resistance to ENOblock (Fig. EV1). Using time-course assays, we evaluated the bactericidal activity of ENOblock against Ab ATCC 17978 and Ab CR17 strains. Figure 2A illustrates that ENOblock (2× and 4×MIC for Ab ATCC 17978 strain) exhibited bactericidal effect after 2, 4 and 8 h, reducing the bacterial count by over 3 log$_{10}$ CFU/mL compared to 0 h. For the Ab CR17 strain, ENOblock (1×, 2× and 4×MIC for Ab CR17 strain) demonstrated a bactericidal effect after 2, 4 and 8 h by reducing the bacterial count by over 3 log$_{10}$ CFU/mL, compared to 0 h.

It is well known that the development of new repurposed drugs includes the assessment of the presence of synergy with clinically used antibiotics. To this end, we conducted a virtual screening of ENOblock in combination with different antibiotic families (colistin, imipenem, ceftazidime and tigecycline) using the EIIP/AQVN criterion to overcome bacterial resistance (Appendix Table S2). Figure 2B suggests that colistin and ENOblock with similar electronic properties, as indicated by their EIIP and AQVN values, tend to exhibit synergistic effects. Conversely, no synergistic activity was observed between the rest of antibiotics and ENOblock with different electronic properties. To confirm experimentally the in silico synergistic effect of ENOblock with colistin against Ab, checkerboard and bacterial growth assays were performed (Fig. 2C,D). Checkerboard assay indicated that ENOblock had a synergistic effect with colistin by enhancing the activity of colistin against Ab ATCC 17978 and Ab CR17 strains, resulting in the fractional inhibitory concentration index (FICI) of ≤0.5. In contrast, the combination of ENOblock with other antibiotics such as imipenem, ceftazidime and tigecycline did not increase their activities, yielding a FICI > 0.5 (Fig. 2D). Moreover, bacterial growth data confirms the checkerboard data showing that sub-MIC of colistin combined with ENOblock (8 mg/L) abolished completely the growth of Ab ATCC 17978 and Ab CR17 strains (Fig. 2E).

## ENOblock inhibits the growth of Ab via different mechanism of action from conventional antibiotics

We employed the fluorescence microscopy-based BCP technique, as previously applied to Ab (Samernate et al, 2023; Nonejuie et al,

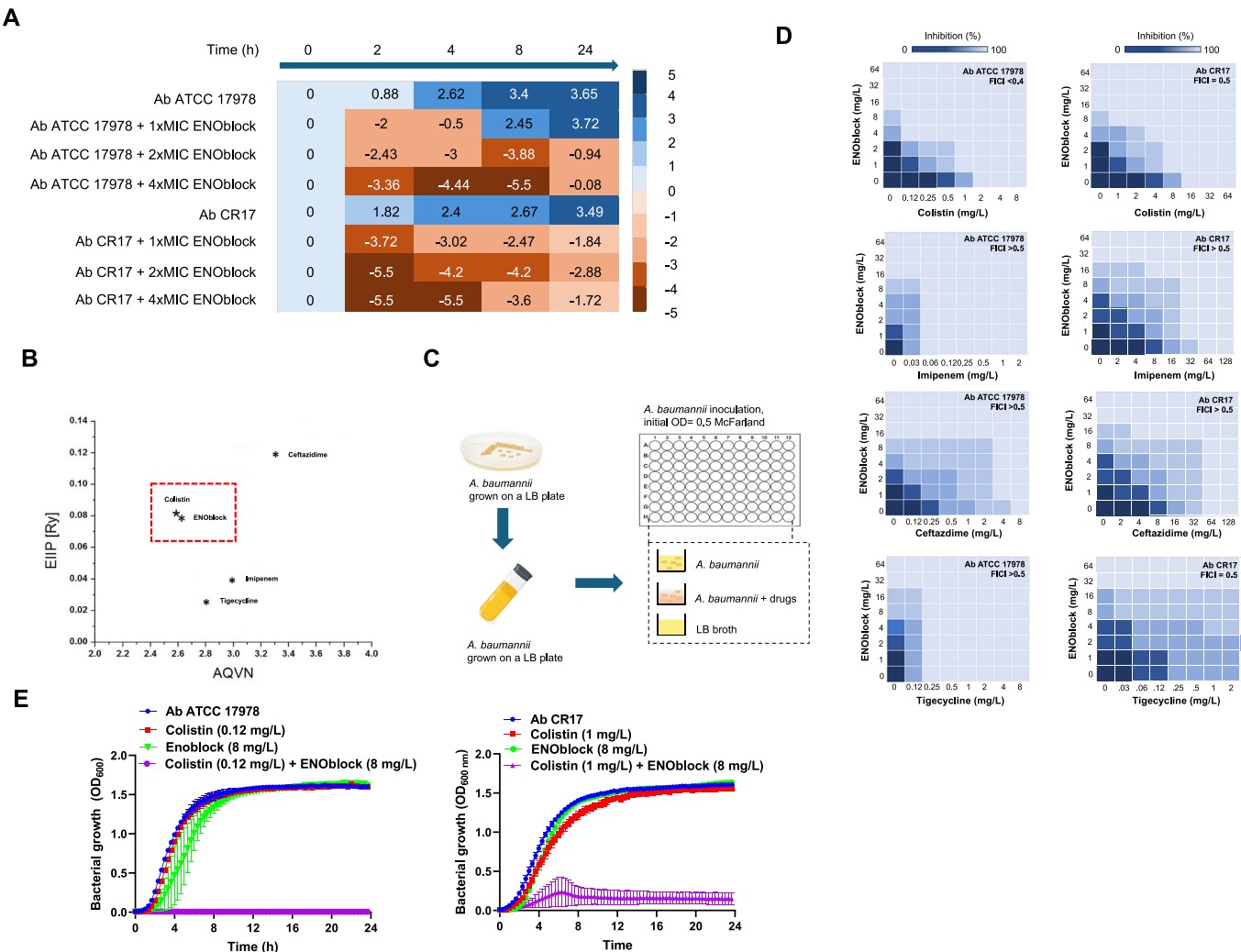

**Figure 2. ENOblock is active against Ab in monotherapy and in combination with colistin.**

(A) Time-kill curves of Ab ATCC 17978 and Ab CR17 strains in the presence of 1× and 1×, 2× and 4×MIC ENOblock for 24 h. Average values of Log₁₀ CFUs/mL at each time point from two biological replicates are represented as mean. Bacterial concentrations for each time point are expressed as the difference from time 0 h. (B) Schematic presentation of the EIIP/AQVN criterion for the selection of ENOblock and colistin combination (green-penicillins, blue-carbapenems, red-quinolones). (C, D) Representative heat plots of microdilution checkerboard assay for the combination of ENOblock and colistin against Ab ATCC 17978 and Ab CR17 strains. A representative example of two biological replicates is shown. (E) Dynamic bacterial growth curve plots of Ab ATCC 17978 and Ab CR17 strains in the absence and presence of colistin, ENOblock and colistin plus ENOblock. The bacterial growth is determined by bacterial density absorbance during 24 h. Average density absorbance values from two biological replicates ± SEM are represented. AQVN average quasi-valence number, EIIP electron-ion interaction potential. Source data are available online for this figure.

2013; Htoo et al, 2019), to investigate the mechanism of action of ENOblock against Ab ATCC 17978 strain. The morphological changes induced by ENOblock were compared with those previously reported profiles of standard antibiotics targeting major cellular pathways, including ciprofloxacin (DNA replication), rifampicin (RNA transcription), minocycline (protein translation), piperacillin and meropenem (cell wall synthesis), and colistin (membrane integrity). BCP results showed that ENOblock (2×MIC)-treated cells exhibit different morphological changes compared to those of antibiotic controls at 1×MIC. In particular, high SYTOX Green signal of ENOblock-treated cells was observed (Fig. 3A), indicating the loss of membrane integrity. However, the ENOblock-treated cells showed different morphological changes

from colistin-treated cells, a membrane integrity control, indicating that ENOblock interferes with bacterial membrane integrity but possibly in a manner different from colistin. Consistent with these differences, the image analysis profiles of ENOblock-treated cells clustered separately from those of untreated, other control antibiotics (Fig. 3B). It is conceivable that ENOblock inhibits pathways distinct from those targeted by the comparator antibiotics, which collectively represent the most common modes of antibacterial action, in a manner similar with previous studies that have observed distinct profiles (Htoo et al, 2022; Khunsri et al, 2023; Lin et al, 2015). Altogether, the results showed that the ENOblock profile is distinct from the six standard antibiotic profiles reported previously (Samernate et al, 2023).

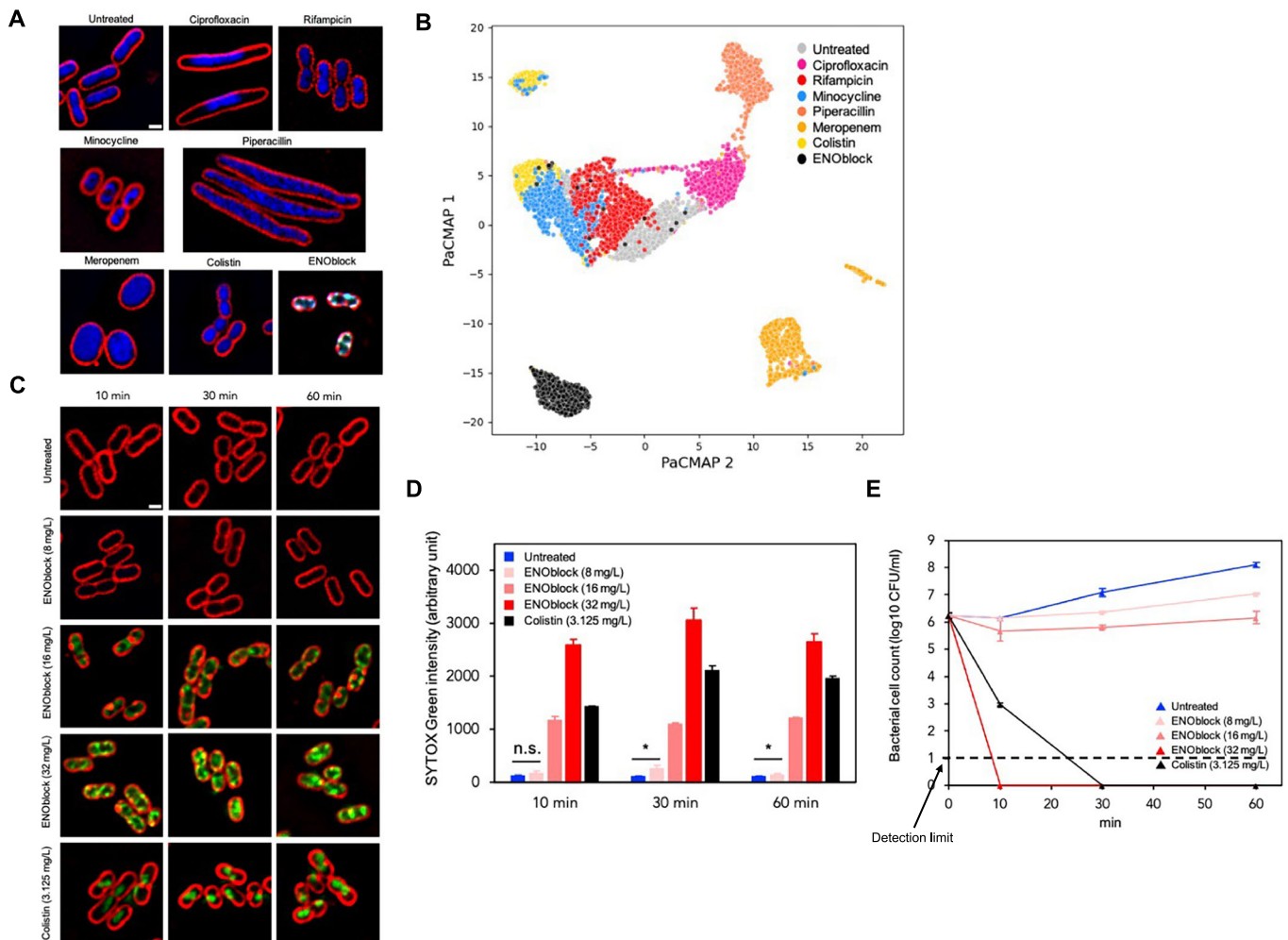

**Figure 3. ENOblock exhibits antibacterial activity against Ab via distinct mechanism of action and displays rapid permeabilization activity.**

(A) Representative images from three biological replicates (scale bar = 1 μm) and (B) a PaCMAP plot of Ab ATCC 17978 cells (>=1000 cells each condition) treated with 16 mg/L ENOblock (2×MIC) for 60 min, compared with antibiotic control data at 60 min exposure: 0.45 mg/L ciprofloxacin (1×MIC), 1 mg/L rifampicin (1×MIC), 0.5 mg/L minocycline (1×MIC), 40 mg/L piperacillin (1×MIC), 0.45 mg/L meropenem (1×MIC), and 0.625 mg/L colistin (1×MIC). All control data used for profile comparison, including images and single-cell profiles, were derived from previously published data (Samernate et al, 2023). In all images, cell membranes were stained with FM4-64 (red), DAPI (blue), and SYTOX Green (green). (C) Representative images from three biological replicates (scale bar = 1 μm) of Ab ATCC 17978 cells treated with 8 mg/L ENOblock (1×MIC), 16 mg/L ENOblock (2×MIC), 32 mg/L ENOblock (4×MIC), or 3.125 mg/L colistin (5×MIC) for 10, 30, and 60 min, stained with FM4-64 (red) and SYTOX Green (green). (D) Quantification of SYTOX Green intensities in untreated and ENOblock-treated cells at different time points. Values represent mean ± SEM of all individual nucleoids in each replicate (number of cells >30 per replicate, three biological replicates). *P = 0.0413 and 0.0419 for 30 and 60 min, respectively (two-tailed Student's t test); n.s. not significant. (E) Colony-forming unit (CFU) counts of Ab ATCC 17978 treated with 8 mg/L ENOblock (1×MIC), 16 mg/L ENOblock (2×MIC), 32 mg/L ENOblock (4×MIC), or 3.125 mg/L colistin (5×MIC) for 10, 30, and 60 min, compared with untreated control. Values represent mean ± SD from three biological replicates per condition. Source data are available online for this figure.

## ENOblock exhibits rapid permeabilization activity against Ab

The observation of a high proportion of ENOblock-treated cells displaying high SYTOX Green intensity prompts us to consider the possibility of observing this impact at earlier time intervals, as membrane permeabilization often happens swiftly within minutes. Prior studies utilizing BCP demonstrated that a compound with the ability to disrupt membranes could impact the bacterial membrane in a mere 10 min timeframe (Nonejuie et al, 2013; Htoo et al, 2022). Consequently, we conducted a temporal examination of SYTOX green staining in cells treated with 8, 16 and 32 mg/L of ENOblock

for 10, 30, and 60 min, in comparison with 3.125 mg/L colistin (5xMIC). The findings indicated that, at 10 min, cells treated with 8 mg/L of ENOblock did not exhibit a notable increase in SYTOX green intensity compared to the untreated sample, whereas treatment with 16 and 32 mg/L ENOblock resulted in increased intensity similar to a positive control, colistin (Fig. 3C,D). Regarding the temporal changes in SYTOX Green intensity, the results indicate that cells reached a saturation point by 30 min in all treated samples, with a slight decrease in SYTOX Green intensity observed at 60 min (Fig. 3D).

To determine whether membrane permeabilization, as indicated by SYTOX Green intensity, correlates with bacterial viability, CFU counts

were measured at each time point (Fig. 3E). No detectable CFUs were observed after treatment with 32 mg/L ENOblock and 3.125 mg/L colistin, starting at 10 and 30 min, respectively, indicating a bactericidal effect at these concentrations. In contrast, treatment with 8 mg/L ENOblock for 10 min (6.16 ± 0.03 log CFU/mL) showed no change in CFU counts compared to the untreated control (6.14 ± 0.00 log CFU/mL). Notably, CFU counts following treatment with 16 mg/L ENOblock at 10 (5.68 ± 0.38 log CFU/mL), 30 (5.82 ± 0.07 log CFU/mL), and 60 min (6.17 ± 0.23 log CFU/mL) indicated a bacteriostatic effect. This was evidenced by the slight reduction and minimal fluctuations in CFU counts compared to the initial value at 0 min (6.23 ± 0.10 log CFU/mL).

Altogether, these results indicate that ENOblock compromises the integrity of the bacterial cell envelope in a concentration- and time-dependent manner, with higher concentrations (≥16 mg/L) leading to rapid membrane permeabilization and loss of cell viability,

## ENOblock acts on Ab through the inhibition of enolase

In order to shed light on the ENOblock mechanism of action, we docked ENOblock in the C-terminal domain of Ab enolase. ENOblock exhibited a high docking score. The most stable pose shows that ENOblock binds to Ser371 and Asp207 amino acids through three hydrogen bonds (Fig. 4A). Moreover, we quantified the interaction between ENOblock and purified AbEnolase (Fig. EV2) using an ITC assay. As shown in Fig. 4B, ENOblock bound to AbEnolase with an affinity of 2.9 ± 0.78 μM. When Ser371 and Asp207 were together substituted with alanine, the binding of the ENOblock to the enolase is affected, making it impossible to calculate its $K_D$ constant, either because it decreases so much that the reaction do not reach the plateau on the binding curve (S371A) or because the protein loses stability causing its aggregation inside the cell (D207A and double mutant) as the ligand is added. (Figs. 4B and EV3). To investigate whether enolase in other Gram-negative bacteria, such as E. coli and K. pneumoniae, might also be a target of ENOblock, we performed in silico analyses to assess ENOblock's binding capacity to enolase from both microorganisms and evaluated their susceptibility to ENOblock using a collection of MDR clinical isolates. Although the ENOblock binding sites in E. coli (GLN166, SER249, ASP245, ASP290, ASP316) and K. pneumoniae (ASP317, SER372) differed from the binding of some sites in Ab (Asp207 and Ser371) (Appendix Fig. S2A,B), ENOblock exhibited similar activity against MDR clinical isolates of E. coli and K. pneumoniae, with $MIC_{50}$ values of 16 mg/L and 32 mg/L, respectively (Appendix Tables S3 and S4). As expected, ENOblock significantly reduced the growth of MDR E. coli Ec MCR1+ and K. pneumoniae Kp10 strains and increased their cell wall permeability (Fig. EV4A,B).

To confirm that Ab enolase is the potential target of ENOblock, we generated an enolase-deficient mutant. Deletion of the enolase gene in the Ab ATCC17978 strain (Δeno) first abolished completely the enolase activity (Fig. 4C) and subsequently increased the ENOblock MIC from 8 to 32 mg/L. The MIC of ENOblock remained unchanged in Δeno strain supplemented with phosphoenolpyruvate (Fig. 4D). Furthermore, we examined the antibacterial activity of ENOblock against the Ab ATCC 17978 and Δeno strains. Figure 4E reveals that the Ab ATCC 17978 strain exhibits rapid growth, reaching an OD of 1 within the first 4 h. However, a significant disparity in growth is observed between the untreated

cells and the cells treated with ENOblock, particularly at higher compound concentrations (16 and 32 mg/L). A different trend of growth inhibition is observed in the Δeno strain, which shows a higher OD value compared with Ab ATCC 17978 strain in the presence of ENOblock treatment (Fig. 4F). Considering these results, the difference in growth can be attributed to the resistance of the mutant strain to the ENOblock. The absence of enolase may hinder the compound's ability to exert its effect, as indicated by the findings of the molecular docking study. Similar results were observed with the E. coli enolase-deficient mutant (Ec Δeno) strain (Appendix Fig. S3).

## ENOblock affects the Ab-host interaction

To evaluate the effect of ENOblock on the interaction between Ab and host cells, we studied the adherence and invasion of the Ab ATCC 17978 and Ab CR17 strains on HeLa and macrophage cells for 2 h in the presence of ENOblock (Fig. 5A). We found that treatment with ENOblock at 1xMIC reduced the counts of adherent Ab ATCC 17978 and Ab CR17 strains on HeLa cells by 47% (P < 0.05) and 31% (P < 0.05), respectively, and on macrophage cells by 43% (P < 0.05) and 29% (P < 0.01) (Fig. 5B). Notably, a more significant reduction was observed in the invasion of both strains, with ENOblock treatment reducing invasive counts in HeLa cells by 76% (P < 0.01) and 46% (P < 0.05), respectively, and in macrophage cells by 67% (P < 0.01) and 41% (P < 0.05), respectively (Fig. 5C). Immunostaining of infected HeLa cells with the Ab ATCC 17978 and Ab CR17 strains, pretreated with ENOblock, showed a significant reduction in Ab attachment to HeLa cells (Fig. 5D). Similarly, ENOblock at 1xMIC significantly reduced the adherence of E. coli MCR1+ and K. pneumoniae Kp10 to HeLa cells (Fig. EV4C,D).

## Host cell interaction induces metabolic changes in Ab and upregulates enolase expression

Given that ENOblock reduces the interaction of Ab with host cells and the Δeno strain is defective in the adherence to host cells, we investigated whether enolase directly mediates this interaction. A TMT quantitative proteomics assay was performed on Ab ATCC 17978 strain alone and Ab ATCC 17978 strain in contact with epithelial and macrophage cells. Overall similarities and differences in protein expression between the two conditions (bacteria alone vs. bacteria in contact with host cells) were assessed using PCA (Fig. 6A). As shown, Ab ATCC 17978 strain in contact with epithelial and macrophage cells displayed distinct protein expression patterns compared to bacteria alone, reflecting the impact of host cell interaction. Pairwise comparisons of differentially expressed proteins (DEPs) (bacteria in contact with epithelial cells vs. bacteria in contact with macrophage cells) revealed two key findings. First, overlapping patterns of overexpressed and downregulated proteins were observed (Fig. 6B; Datasets EV1 and 2). Second, functional analysis of DEPs using Clusters of COG (Fig. 6C) identified a common behaviour of Ab ATCC 17978 strain when is in contact with host cells. Overexpressed clusters were enriched for proteins involved in amino acid, inorganic ion, lipid, carbohydrate, and coenzyme transport and metabolism, transcription, defense mechanisms, post-translational modification, protein turnover, chaperones, and secondary metabolites. Notably, the number of significantly overexpressed proteins (red) exceeded that

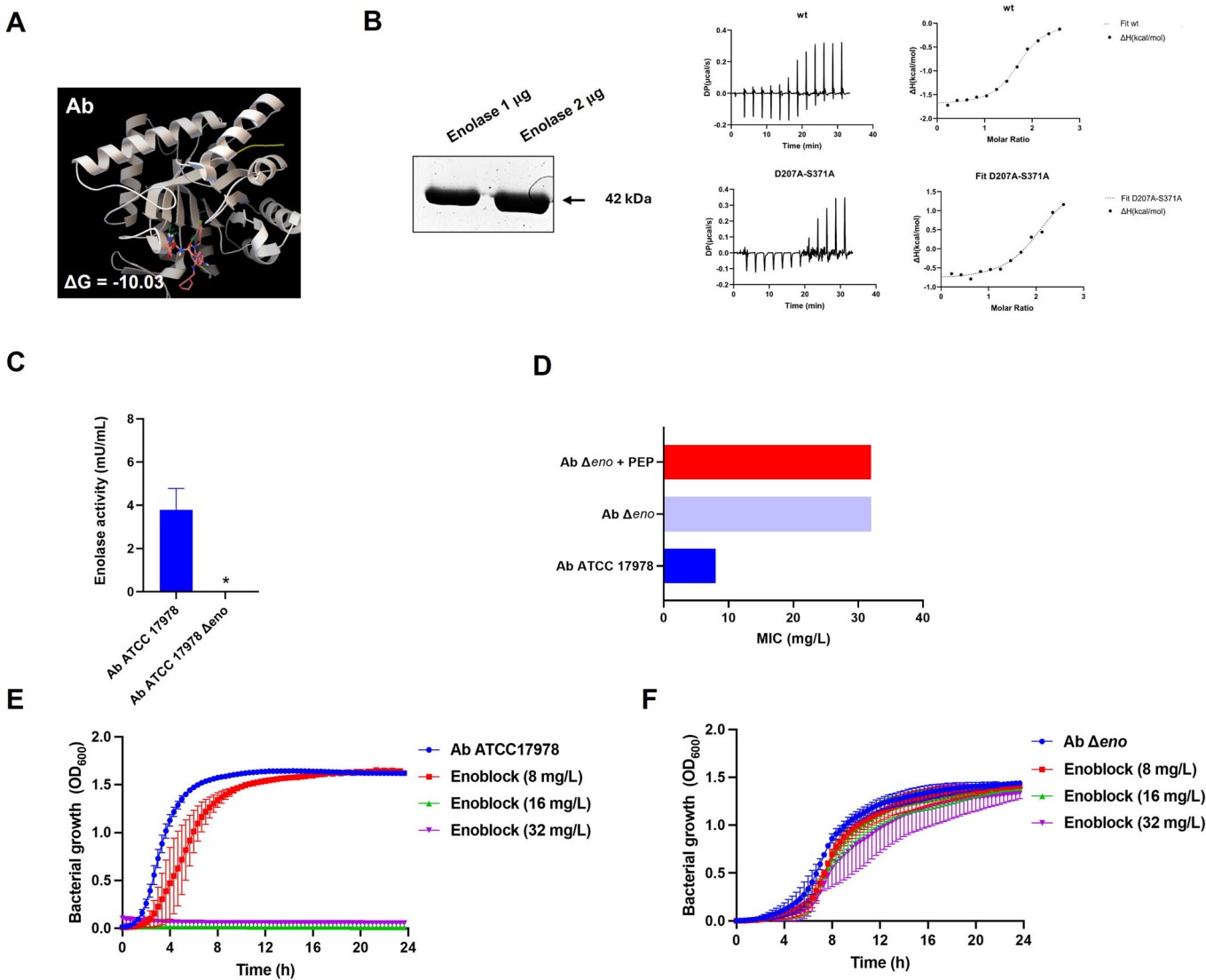

**Figure 4. ENOblock acts on Ab through the inhibition of enolase.**

(A) Structural models generated by docking of ENOblock into the C-domain of Ab enolase. ENOblock is displayed as sticks. ΔG: Glide score. (B) Isothermal titration calorimetry (ITC) titrations with integrated fitted heat plots of ENOblock binding with wt enolase (wt) or enolase (D207A-S371A). A representative example out of three biological replicates is shown. (C) Enolase activity in determined in Ab ATCC 17978 and Δeno strains. The data are presented as means of three biological replicates ± SEM, and Student $t$ test was used for statistical analysis. *$P = 0.005$, Ab ATCC 17978 vs Δeno (two-tailed Student's $t$ test). (D) MIC of ENOblock against Ab ATCC 17978 vs Δeno strains with or without supplementation of 4 mM phosphoenolpyruvate. A representative example out of two biological replicates. (E, F) Bacterial growth curve plots of Ab ATCC 17978 vs Δeno strains in the absence and presence of ENOblock treatment at different concentrations. Average values from two biological replicates ± SEM are represented. Source data are available online for this figure.

of downregulated proteins (blue) in response to bacterial interactions with epithelial and macrophage cells (Fig. 6D). Enolase, the target of ENOblock, was significantly upregulated in *A. baumannii* upon contact with epithelial cells ($\log_2$ fold change >2) and macrophage cells ($\log_2$ fold change >1). To validate the enolase's role in this interaction, we determined the adherence of the Δeno strain, with or without phosphoenolpyruvate supplementation, to HeLa cells. Deletion of the *enolase* gene in the Ab ATCC17978 strain reduced the Ab adherence to HeLa cells by more than 80% compared to the wt strain; however, supplementation of the Δeno strain with phosphoenolpyruvate restored adherence, reducing it by 67% (Fig. 6E). These findings suggest

that host cell interaction induces metabolic changes in Ab and might influence enolase expression.

## Enolase mediates the interaction of Ab with host cells via binding to host proteins

Given that Ab upregulates enolase expression when in contact with host cells, and considering that enolase of *P. aeruginosa* and *Streptococcus suis* binds to human plasminogen and fibronectin on host cells, respectively (Ceremuga et al, 2014; Esgleas et al, 2008), we sought to determine whether Ab enolase binds to host proteins (Plasminogen, fibronectin and fibrinogen) to mediate host cell

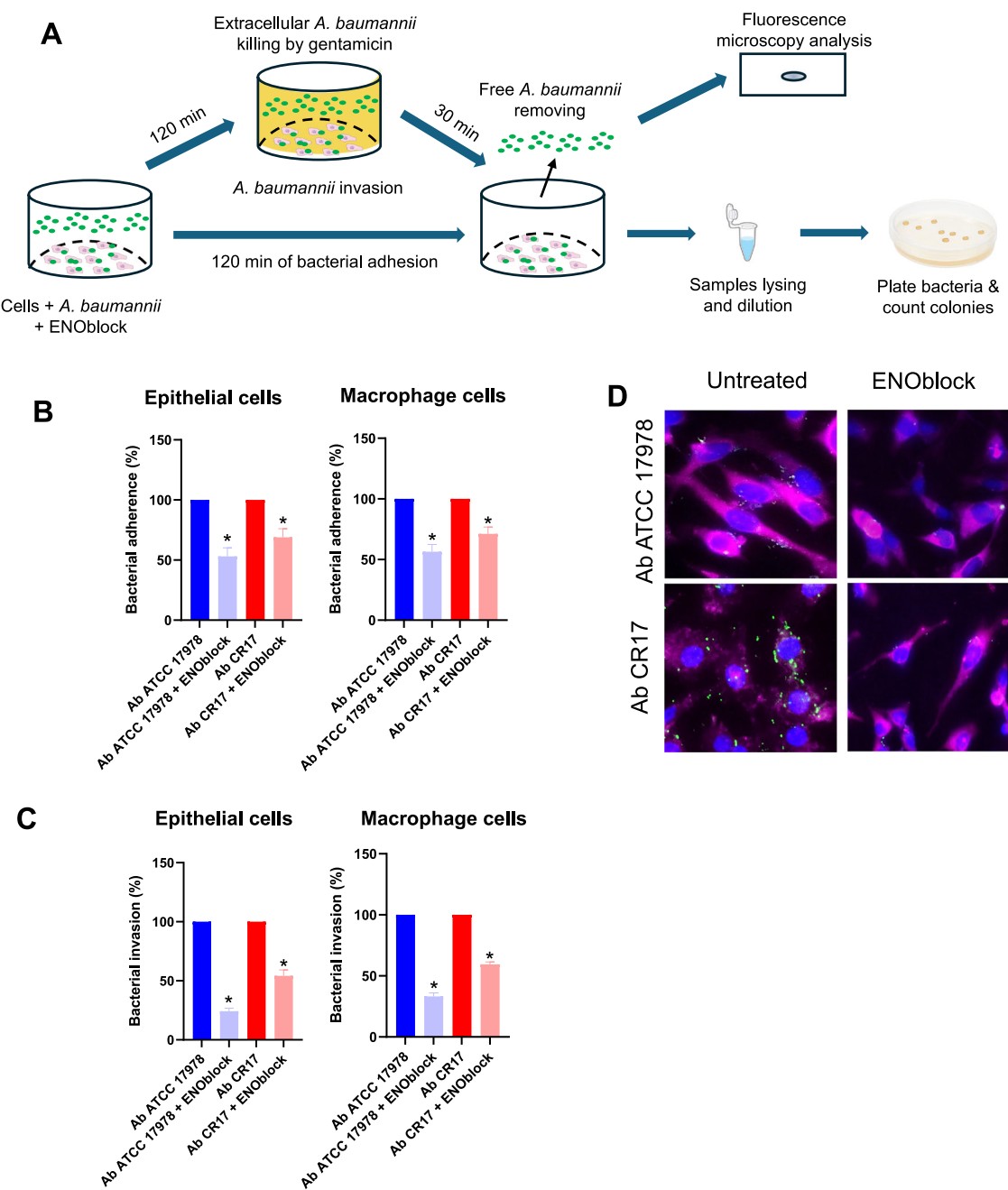

**Figure 5. ENOblock affects the Ab-host interaction.**

(A) Schematic of the bacterial adhesion/invasion assay. (B) Analysis of Ab ATCC 17978 and Ab CR17 strains adhesion into HeLa and macrophage cells with (1xMIC) and without ENOblock treatment. The data are presented as means of three biological replicates ± SEM, For HeLa cells, *$P = 0.023$: Ab ATCC 17978 vs Ab ATCC 17978 + ENOblock and *$P = 0.048$: Ab CR17 vs Ab CR17 + ENOblock (two-tailed Student's $t$ test). For macrophage cells, *$P = 0.018$: Ab ATCC 17978 vs Ab ATCC 17978 + ENOblock and *$P = 0.035$: Ab CR17 vs Ab CR17 + ENOblock (two-tailed Student's $t$ test). (C) Analysis of Ab ATCC 17978 and Ab CR17 strains invasion into HeLa and macrophage cells with (1×MIC) and without ENOblock treatment. The data are presented as means of three biological replicates ± SEM, For HeLa cells, *$P = 0.001$: Ab ATCC 17978 vs Ab ATCC 17978 + ENOblock and *$P = 0.011$: Ab CR17 vs Ab CR17 + ENOblock (two-tailed Student's $t$ test). For macrophage cells, *$P = 0.001$: Ab ATCC 17978 vs Ab ATCC 17978 + ENOblock and *$P = 0.002$: Ab CR17 vs Ab CR17 + ENOblock (two-tailed Student's $t$ test). (D) Immunostaining of fibronectin of HeLa cells (magenta) and Ab ATCC 17978 and Ab CR17 strains (green) pretreated with ENOblock (0× and 1×MIC), after bacterial adherence for 2 h, was performed by specific primary antibodies against both strains and their respective secondary antibodies. Blue staining shows the location of HeLa cell nuclei. A representative image out of three biological replicates is shown. Source data are available online for this figure.

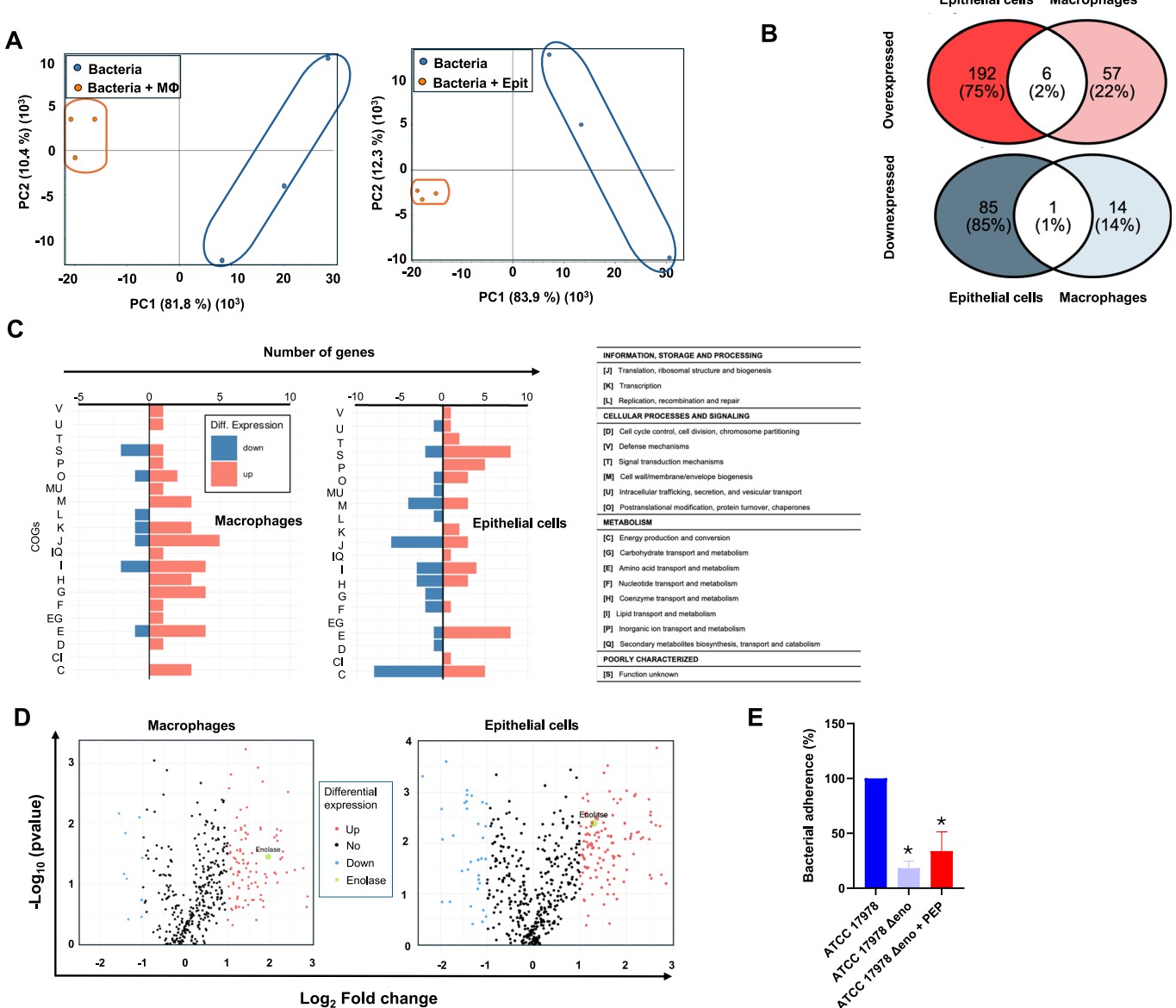

**Figure 6. Host cell interaction induces metabolic changes in Ab and upregulates enolase expression.**

(A) PCA plot of TMT data from Ab infected macrophages and epithelial cells of three biological replicates each group. (B) Venn diagrams showing the overlap of DEGs between the two comparisons: Ab infected macrophages vs Ab infected epithelial cells. The upper panel shows overlaps of overexpressed proteins, while the lower panel shows overlaps of downexpressed proteins. A total of six proteins were found to be commonly overexpressed in Ab upon exposure to both epithelial and macrophage cells: enolase, tyrosine–tRNA ligase, malonyl-CoA:acyl carrier protein transacylase, N-succinylarginine dihydrolase, carbonic anhydrase, and biotin carboxylase. Additionally, one protein, isocitrate dehydrogenase, was consistently downregulated in response to both cell types. (C) COG category distribution of DEPs. The overexpressed and downexpressed proteins in each comparison are categorized by their COG functional groups, with blue representing downexpressed proteins and orange representing overexpressed proteins. (D) Volcano plots depicting the DEPs for the two comparisons. Proteins significantly overexpressed are shown in red, while significantly downexpressed proteins are shown in blue. Enolase overexpression is shown in green. Non-significant proteins are indicated in black. Three biological replicates from each group were compared. (E) *A. baumannii* Ab ATCC 17978 and Δeno strains, with or without supplementation of 4 mM phosphoenolpyruvate, adhesion to HeLa cells. The data are presented as means of three biological replicates ± SEM, and Student *t* test was used for statistical analysis. *$P = 0.006$: Ab ATCC 17978 vs Δeno and *$P < 0.001$: Ab ATCC 17978 vs Δeno + PEP (two-tailed Student's *t* test). MΦ macrophage cells, Epit epithelial cells. Source data are available online for this figure.

interactions with *A. baumannii*. ISM analysis and purified AbEnolase (Fig. EV2) were used to assess interactions with host proteins. As shown in Fig. 7A–C, ISM-SM was employed to identify shared informational characteristics among Ab enolase, host proteins (Plasminogen, fibronectin and fibrinogen), and ENOblock. The ISM is based on the principle that interacting proteins and small molecules share common informational properties, appearing as cross-spectral peaks (Sencanski et al, 2022). Our consensus spectral analysis revealed a characteristic peak at frequency F(0.270) shared by plasminogen, enolase and ENOblock (Fig. 7A). The same peak was observed when fibronectin, enolase and ENOblock were analyzed (Fig. 7B). However, a not primary

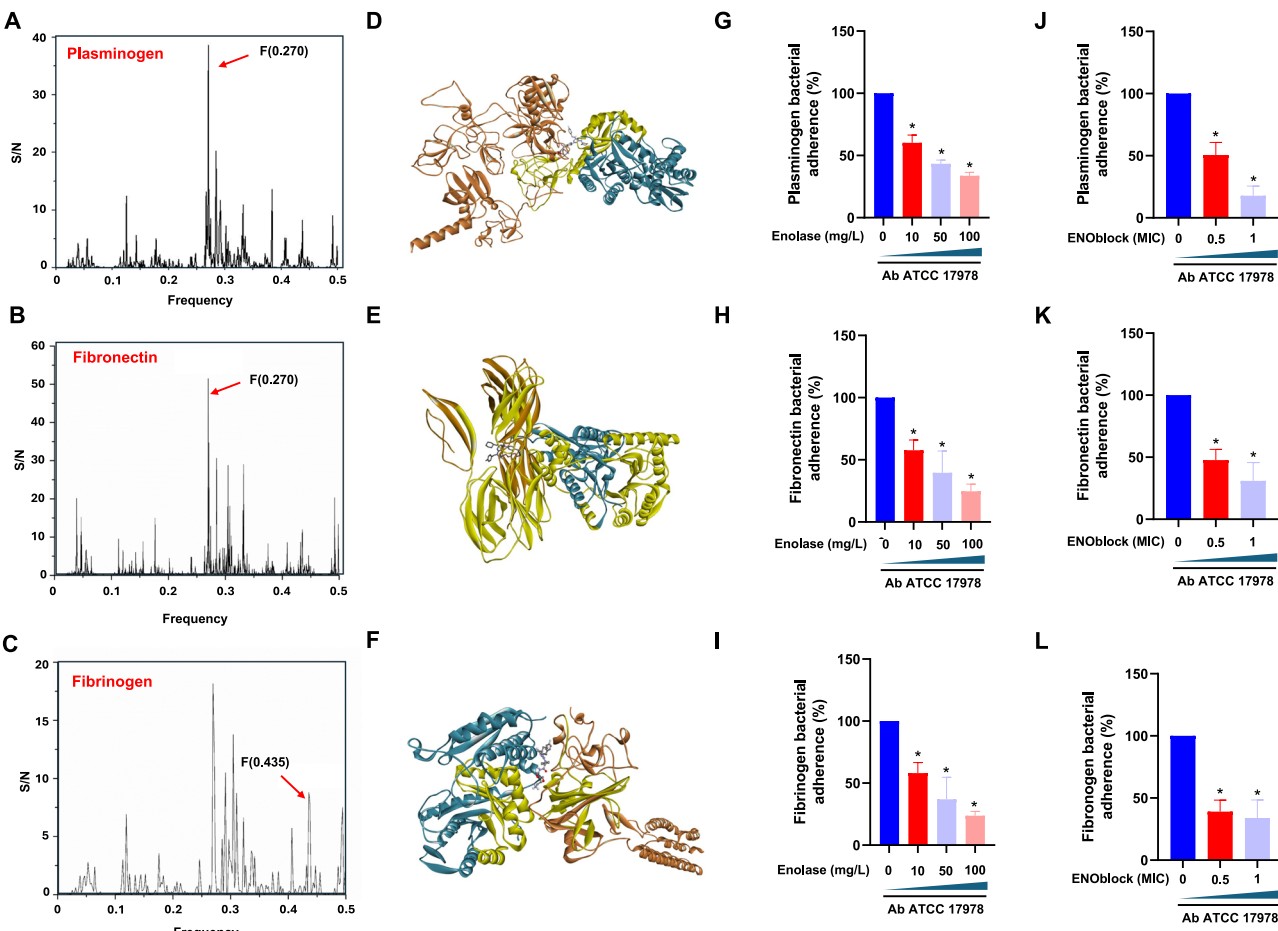

**Figure 7. Enolase mediates the interaction of Ab with host cells via binding to host proteins.**

(A–C) Consensus-spectrum (CIS) of enolase, plasminogen and ENOblock, enolase, fibronectin and ENOblock, and enolase, fibrinogen and ENOblock. (D–F) Structural models of protein-protein complex generated by Alphafold3 analysis by docking of ENOblock (ball and stick), enolase (pelorous ribbons) and plasminogen, fibronectin or fibrinogen (tangerine ribbons). The intermolecular interactions from the obtained complex were according to ISM method–detected interaction domains (yellow ribbons). (G–I) Inhibition of Ab adherence to immobilized plasminogen, fibronectin and fibrinogen by free enolase. Ab ATCC 17978 strain was incubated in plasminogen, fibronectin and fibrinogen-coated wells for 3 h at room temperature, containing increasing concentrations of free enolase (0, 10, 50 and 100 mg/L). The data are presented as means of three biological replicates ± SEM. For plasminogen, *$P = 0.023$: Ab ATCC 17978 vs Ab ATCC 17978 + enolase (10 mg/L), *$P = 0.002$: Ab ATCC 17978 vs Ab ATCC 17978 + enolase (50 mg/L) and *$P = 0.001$: Ab ATCC 17978 vs Ab ATCC 17978 + enolase (100 mg/L) (two-tailed Student's $t$ test). For fibronectin, *$P = 0.006$: Ab ATCC 17978 vs Ab ATCC 17978 + enolase (10 mg/L), *$P = 0.025$: Ab ATCC 17978 vs Ab ATCC 17978 + enolase (50 mg/L) and *$P = 0.005$: Ab ATCC 17978 vs Ab ATCC 17978 + enolase (100 mg/L) (two-tailed Student's $t$ test). For fibrinogen, *$P = 0.039$: Ab ATCC 17978 vs Ab ATCC 17978 + enolase (10 mg/L), *$P = 0.017$: Ab ATCC 17978 vs Ab ATCC 17978 + enolase (50 mg/L) and *$P = 0.002$: Ab ATCC 17978 vs Ab ATCC 17978 + enolase (100 mg/L) (two-tailed Student's $t$ test). (J–L) Inhibition of Ab adherence to immobilized plasminogen, fibronectin and fibrinogen by ENOblock. Ab ATCC 17978 strain was incubated in plasminogen, fibronectin and fibrinogen-coated wells for 2 h at room temperature, containing increasing concentrations of ENOblock (0.5× and 1×MIC). The data are presented as means of three biological replicates ± SEM, *$P < 0.05$: treatment vs no treatment; two-tailed Student's $t$ test. For plasminogen, *$P = 0.039$: Ab ATCC 17978 vs Ab ATCC 17978 + ENOblock (0.5×MIC) and *$P = 0.008$: Ab ATCC 17978 vs Ab ATCC 17978 + ENOblock (1xMIC) (two-tailed Student's $t$ test). For fibronectin, *$P = 0.026$: Ab ATCC 17978 vs Ab ATCC 17978 + ENOblock (0.5×MIC) and *$P = 0.042$: Ab ATCC 17978 vs Ab ATCC 17978 + ENOblock (1×MIC) (two-tailed Student's $t$ test). For fibrinogen, *$P$ 0.022 = Ab ATCC 17978 vs Ab ATCC 17978 + ENOblock (0.5×MIC) and *$P = 0.045$: Ab ATCC 17978 vs Ab ATCC 17978 + ENOblock (1xMIC) (two-tailed Student's $t$ test). Adherent bacteria to plasminogen, fibronectin and fibrinogen-coated wells were quantified by serial dilutions as described in materials and methods. Results were expressed as the percentage of total untreated Ab adhered to immobilized plasminogen, fibronectin and fibrinogen. Source data are available online for this figure.

frequency peak F(0.435) has been shared by fibrinogen, enolase and ENOblock. This peak was considered more relevant because the signal-to-noise (S/N) ratio between the enolase–ENOblock system and the fibrinogen–enolase–enoblock system shows a more pronounced change at F(0.435) than at F(0.270), with respective ratios of 2.58 and 1.45. (Fig. 7C; Appendix Table S5). These findings suggest that shared informational content at F(0.270) and F(0.435) indicates potential interactions between enolase and

human proteins. In the next step, the plasminogen, fibronectin or fibrinogen–enolase complexes were modelled. Obtained sequences of enolase and plasminogen, fibronectin or fibrinogen were sent to Alphafold3 to build a protein-protein complex. The intermolecular interactions from the obtained complex were according to ISM method-detected interaction domains (Fig. 7E–G, yellow ribbons). Finally, enoblock was docked into the plasminogen-enolase cavity, fibronectin-enolase cavity and fibrinogen–enolase cavity, using

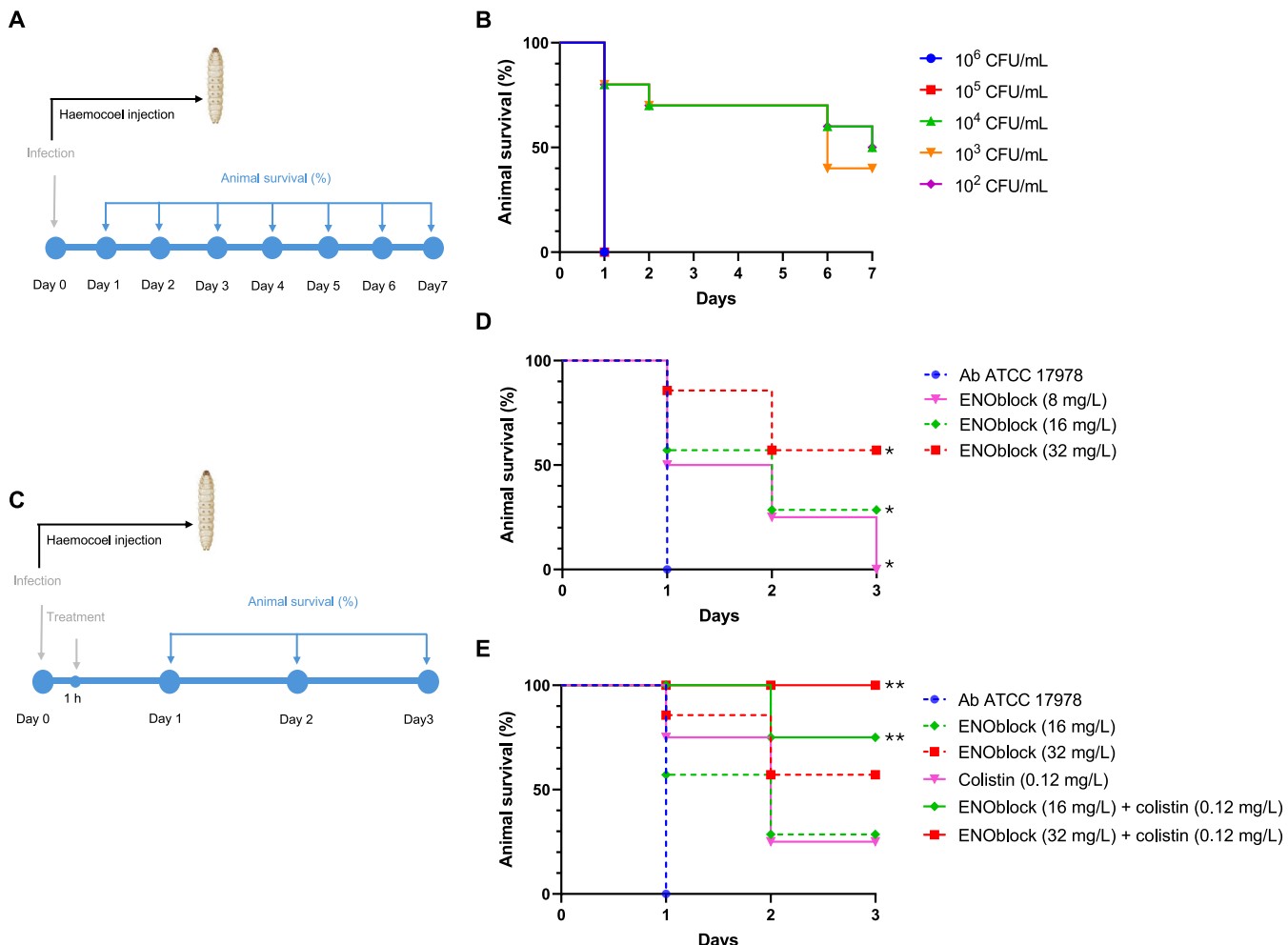

**Figure 8. ENOblock displays efficacy in _G. mellonella_ model of infection.**

(A) Experimental design for Ab ATCC 17978 strain lethal doses determination. (B) Seven days of mortality monitoring in _G. mellonella_ administered with different inoculum of Ab ATCC 17978 strain ($n = 8$ per group). (C) Experimental design for Ab ATCC 17978 strain infection and treatment. (D) Three days of mortality monitoring in _G. mellonella_ administered with $\mathrm{MDL}_{100}$ ($10^5$ CFU/mL) Ab ATCC 17978 and treated or not with ENOblock (8, 16 and 32 mg/L; $n = 7$–8 per group). $*P = 0.030$: Ab ATCC 17978 vs ENOblock (8 mg/L), $*P = 0.016$: Ab ATCC 17978 vs ENOblock (16 mg/L) and $*P = 0.001$: Ab ATCC 17978 vs ENOblock (32 mg/L); Kaplan–Meier test. (E) Three days of mortality monitoring in _G. mellonella_ administered with $\mathrm{MDL}_{100}$ ($10^5$ CFU/mL) Ab ATCC 17978 and treated or not with ENOblock (16 and 32 mg/L; $n = 8$ per group), colistin (0.12 mg/L; $n = 8$ per group) alone and in combination. $**P = 0.017$: ENOblock (16 mg/L) + colistin (0.12 mg/L) vs colistin (0.12 mg/L) and $**P < 0.0001$: ENOblock (32 mg/L) + colistin (0.12 mg/L) vs colistin (0.12 mg/L); Kaplan–Meier test. Source data are available online for this figure.

Autodock Vina software. The obtained confirmation had a docking energy of $-9.3$, $-7.9$ and $-9.9$ kcal/mol, respectively. To experimentally validate these observations, we first performed binding assays using purified AbEnolase with ECM proteins to outcompete binding sites of Ab. The adhesion of Ab ATCC 17978 strain to immobilized plasminogen, fibronectin or fibrinogen was significantly reduced with the addition of increasing concentrations of AbEnolase or ENOblock (Fig. 7G–L). However, the incubation of Ab ATCC 17978 strain with 0.5× and 1×MIC during 30 min did not reduce the bacterial concentration (Fig. EV5).

## ENOblock presents therapeutic efficacy in vivo

To confirm the in vitro effect of ENOblock in monotherapy and in combination with colistin against Ab, and to study this efficacy in a complete organism, we moved to an invertebrate model of infection by Ab. First, we determined the virulence of the Ab ATCC 17978 strain after haemocoel administration in _G. mellonella_ (Fig. 8A). The mortality rates of animals were inoculum concentration dependent. $\mathrm{LD}_{50}$ and $\mathrm{MLD}_{100}$ for the Ab ATCC 17978 strain were $10^2$ and $1 \times 10^5$ CFU/mL, respectively (Fig. 8B). Subsequently, in a _G. mellonella_ model of infection, we administered ENOblock (8, 16 and 32 mg/L) alone and in combination with colistin (0.12 mg/L) to animals after haemoceol administration of an $\mathrm{MLD}_{100}$ of the Ab ATCC 17978 strain (Fig. 8C). We observed that the treatment with ENOblock (8, 16 and 32 mg/L) showed a concentration-dependent increase in survival (0%, 28.57% and 57.14%) compared with untreated animals ($P < 0.05$, $P < 0.05$ and $P < 0.01$) (Fig. 8D). The combination of ENOblock (16 and 32 mg/L) with colistin significantly increased larval survival to 75% and 100%, respectively, compared

colistin monotherapy at 0.12 mg/L (25% survival) ($P < 0.05$ and $P < 0.0001$) (Fig. 8E). Of note, ENOblock (8, 16 and 32 mg/L) as control groups showed zero toxicity in *G. mellonella*.

## Discussion

Due to the antibacterial effects of various anticancer drugs on Ab (Miró-Canturri et al, 2019, 2021a, 2021b, 2021c), we hypothesized that ENOblock, identified in this study after a HTS of the EU-OPENSCREEN library, might exhibit strong antibacterial activity against colistin- or carbapenem-resistant Ab. After initial antimicrobial confirmation of ENOblock, we additionally tested its susceptibility against 32 clinical isolates of Ab resistant to colistin or carbapenems (Tables 1 and 2). The ENOblock $MIC_{90}$ is 32 mg/L, which is two to more than six times lower than the $MIC_{90}$ of carbapenem and colistin, respectively. This $MIC_{90}$ value falls within the range of other known antibiotics such as amikacin, amoxicillin-clavulanic acid, ceftazidime-avibactam, and fosfomycin, among others (EUCAST, 2024). Additionally, it is three times lower than the $CC_{50}$ values of 112,7 and 116.2 mg/L determined (Appendix Fig. S1) in epithelial and macrophages cells. This suggests that ENOblock, at $MIC_{90}$ concentration, is unlikely to be toxic or cause off-target effects in humans, despite targeting enolase, a highly conserved enzyme also present in human cells.

The comparison of the ENOblock cytotoxity with the cytotoxicity profiles of currently used antibiotics (e.g., colistin and tigecycline), showed that the $CC_{50}$ values of ENOblock fall above the range reported for both antibiotics. For instance, colistin has been reported to exhibit $CC_{50}$ values between 50 and 100 mg/L in HK-2 cells (Lee et al, 2015). Moreover, colistin at 8 mg/L and polymyxin B at 40 mg/L have been shown to reduce viability by 50% in both HeLa cells and PMA-differentiated THP-1 cells (Luo et al, 2013; Kagi et al, 2022). Regarding tigecycline, $CC_{50}$ values of 44 and 51 mg/L have been reported in NCM-356 cells and PC12 cells (Ruiz-Malagón et al, 2023; Huang et al, 2022), respectively. Furthermore, prolonged treatment with tigecycline (3 days) decreased its $CC_{50}$ to 1.7 mg/L in Jurkat T cells and to a range of 1.2–5.5 mg/L in PBMCs (Shao et al, 2025).

It is noteworthy that ENOblock demonstrated similar activity against MDR *E. coli* and *K. pneumoniae*, consistent with previously published data on the antibacterial activity of the anticancer drug family tamoxifen and its metabolites used in earlier studies (Miró-Canturri et al, 2021a, 2021b, 2021c).

Specific AQVN/EIIP domains, combined with structural properties, can serve as effective filters for the virtual screening of molecular libraries to identify new drug candidates, including new antibiotics. Using molecular descriptors, the EIIP and AQVN we have proposed suitable antibiotics for treating MDR bacterial infections (Veljkovic et al, 2016). In this study, we analyzed the electronic properties of ENOblock and antibiotics to which Ab ATCC 17978 is sensitive. Our analysis suggests that antimicrobials with similar electronic properties tend to act synergistically. However, the limited number of molecules analyzed restricts our ability to establish a criterion for predicting synergy in Ab. Moreover, studies show that the molecular mechanisms underlying such synergistic effects remain not fully understood (Sullivan et al, 2020). Nonetheless, the presented results, along with the observed tendency of ENOblock and colistin to exhibit synergy, suggest that

similar electronic properties may contribute to effective antibacterial combinations. Small molecules with similar AQVN and EIIP values have previously been shown to interact with the common therapeutic target (Veljkovic et al, 2011; Radosevic et al, 2019).

The antibacterial activity of ENOblock at 1×, 2×, and 4×MIC against the colistin-resistant strain (Ab CR17) is higher than against the colistin-susceptible reference strain (Ab ATCC 17978). This result may be related to differences in the cell wall structure of the two strains, where colistin-resistant strains of Ab are more permeable than colistin-susceptible strains (Miró-Canturri et al, 2020; López-Rojas et al, 2011a; Ayerbe-Algaba et al, 2019). In addition, the apparent CFU recovery at 8 mg/L of ENOblock in Fig. EV1 reflects survival of a subpopulation rather than complete inhibition, which is not uncommon near the MIC threshold. The ENOblock MIC determination is based on visible growth inhibition in broth culture, while CFU assays may detect surviving subpopulations even at MIC levels.

The antimicrobial activity of ENOblock identified in this study suggests promising potential that warrants further in vivo exploration after determining its pharmacokinetic parameters. However, in vitro bacterial growth showed progressive regrowth of the Ab ATCC 17978 strain after treatment with ENOblock at 1xMIC, suggesting that this strain may have acquired resistance to this compound. It is worth noting that the MIC of ENOblock against the Ab ATCC 17978 strain under these time-kill assay conditions is 8 mg/L, which is below the 2× and 4×MIC of ENOblock. Further investigations, including the determination of its concentration during the time-kill assay, are necessary to better understand the regrowth.

As with other antimicrobial agents, there is a risk of emerging resistance. Ab adaptation mechanisms could reduce susceptibility to ENOblock, either through mutations in enolase or compensatory metabolic pathways. However, we showed that selective pressure with sub-MIC concentrations of ENOblock did not induce resistance or adaptive growth over the days of exposure. Altogether, our results suggest that while stable genetic resistance was not observed under the tested conditions, transient phenotypic adaptations, potentially involving metabolic redirecting or stress responses, could affect susceptibility to ENOblock.

Additionally, BCP analysis showed that ENOblock-treated Ab exhibits distinct morphological changes compared to comparator antibiotics (Fig. 3A,B), suggesting that ENOblock inhibits pathways different from those targeted by the comparator antibiotics. Of note, the ENOblock concentration used was a higher MIC than that used for any of the antibiotics. Membrane-disruptive agents can be classified into various subcategories, each exhibiting a unique structure on BCP. Nevertheless, the BCP profile of ENOblock-treated cells observed in this study did not precisely correspond to any previously described profiles (Nonejuie et al, 2013; Htoo et al, 2019; Htoo et al, 2022; Khunsri et al, 2023). Consequently, the specific target on the membrane for ENOblock could not be determined, but the presence of SYTOX Green indicates that ENOblock can permeabilize membranes. Therefore, future studies should investigate the precise locations on the bacterial membrane where ENOblock is active.

It is widely known that ENOblock inhibits enolase activity in eukaryotic cells (Jung et al, 2013), and enolase, a cytoplasmic glycolytic enzyme, is a key component of the RNA degradosome in Gram-negative bacteria such as *E. coli* and *P. aeruginosa* (Canback

et al, 2002). For this reason, we decided to conduct a preliminary computational study to determine if ENOblock could potentially bind to enolase in Ab. As a result, ENOblock exhibited a better docking score. Deletion of *enolase* gene in Ab increased the ENOblock MIC fourfold and increased bacterial growth in the presence of ENOblock, suggesting that enolase could be a potential target of ENOblock in Ab. The interaction between AbEnolase and ENOblock was confirmed with a microcalorimetric assay where recombinantly expressed and purified AbEnolase showed a $K_D$ 2.9 μM (Fig. 4B). The computational study showed two potential amino acids (D207 and S371) in close proximity in the predicted 3D structure that could be part the binding site of ENOblock. To confirm this hypothesis, AbEnolase with both single and the double mutations were recombinantly expressed and purified, where the amino acids of interest were exchange for an alanine. These mutants showed a decrease in affinity against the ENOblock. The mutation in the Serine 370 caused a reduction in the affinity while the mutation in the Aspartic acid 207 led not only to decreased affinity but also to decreased protein stability causing protein aggregation from a ligand:enolase Molar Ratio of 1.5. The double mutation showed a stronger effect in protein stability and ligand affinity decrease.

Enolase is widely reported as essential in several bacterial species, including *A. baumannii* (Pancholi, 2001). However, conditional or partial loss of enolase function may still occur under certain experimental conditions, allowing the bacteria to survive through compensatory metabolic adaptations (Hottes et al, 2013). These adaptations likely differ in nature and timescale between our constructed Δeno mutant and the resistance assays of this study, where short-term selective pressure might not allow for the same degree of metabolic compensation.

Importantly, two studies have reported that enolase also resides on the cell wall outer membrane of *E. coli* and *P. aeruginosa* (Ceremuga et al, 2014; Witkowska et al, 2005) and mediates the binding of *P. aeruginosa* and *S. suis* to plasminogen and fibronectin on host cells (Ceremuga et al, 2014; Esgleas et al, 2008). In this study, we showed that ENOblock reduces the interaction of Ab with host cells, possibly by affecting the host proteins binding activity of Ab enolase. Other components of the *A. baumannii* cell wall such as Tuf, OmpA and chaperone-usher pathway pili binds also to plasminogen, fibronectin and fibrinogen, respectively (Smani et al, 2012; Tamadonfar et al, 2023; Koenigs et al, 2015). Consequently, the possibility of multi-targeting, with ENOblock binding to both cytosolic and membranal enolase, might contribute to its overall activity, which is an attractive aspect of ENOblock's antibacterial properties.

We have demonstrated through various assays that ENOblock can be repurposed as an antibacterial agent. However, its in vitro efficacy might not fully translate to in vivo conditions due to differences in pharmacokinetics, tissue distribution, and metabolic stability. Our preliminary in vivo experiments have shown that ENOblock, at 32 mg/L, has been able to increase animal survival in presence of Ab. This result aligns with previous observations that deletion of *enolase* gene abolishes the virulence of *P. aeruginosa* in a murine acute pneumonia model (Weng et al, 2016). Further in vivo studies and preclinical models such as murine acute pneumonia are necessary to confirm the therapeutic potential and safety profile of ENOblock, and future work should focus on detailed toxicity

assessments and the evaluation of ENOblock's selectivity for bacterial versus human enolase that showed higher similarity with Ab and *G. mellonella* enolase (Appendix Fig. S4).

In summary, this drug discovery approach ought to be viewed as the first step in creating a brand-antimicrobial drugs. The efficacy of ENOblock can be investigated further by combining this newly repurposed compound with clinically used antibiotics (like colistin) in in vivo experiments, with the aim of reducing mortality and improving therapeutic efficacy in cases of severe infections, even with the current antimicrobial therapies.

# Methods

### Reagents and tools table

| Reagent/resource | Reference or source | Identifier or catalog number |
|---|---|---|
| **Experimental models** | | |
| Clinical strains of Ab colistin-resistant | Valencia et al, 2009; López-Rojas et al, 2011b | N/A |
| Clinical strains of Ab carbapenem-intermediate/resistant | GenBank Bioproject PRJNA422585 | N/A |
| Clinical strains of *E. coli* colistin-resistant | Molina Panadero et al, 2025 | N/A |
| Clinical strains of *Klebsiella pneumoniae* carbapenem intermediate/resistant | Poirel Lab | N/A |
| Ab ATCC 17978 | LGC Standards | Cat# ATCC 17978 |
| Ec ATCC 25922 | LGC Standards | Cat# ATCC 25922 |
| Ab9 | Herrera-Espejo et al, 2024 | N/A |
| *E. coli* MG1655 | Murashko and Lin Chao, 2017 | N/A |
| Ab186 | Herrera-Espejo et al, 2024 | N/A |
| *E. coli* C1-7-LE | Herrera-Espejo et al, 2024 | N/A |
| *E. coli* MCR1+ | Herrera-Espejo et al, 2024 | N/A |
| *K. pneumoniae* Kp10 | Jayol et al, 2018 | N/A |
| *Pseudomonas aeruginosa* ATCC 27853 | LGC | Cat# ATCC-27853 |
| Ab CR17 | López-Rojas et al, 2011b | N/A |
| Ec Δeno strain | Murashko and Lin Chao, 2017 | N/A |
| *E. coli* Rosetta™ DE3 pRare | Merck 690 Millipore | N/A |
| *E. coli* DH5α | Andalusian Centre of Developmental Biology | N/A |
| HeLa Cell Line | LGC Standards | Cat# ATCC-CCL-2 |
| THP-1 cell line | LGC Standards | Cat# ATCC-TIB-202 |

| Reagent/resource | Reference or source | Identifier or catalog number |
|---|---|---|
| *Galleria mellonella larvae* | Artroposfera | Cat# 2400000003281 |
| **Antibodies** | | |
| Mouse anti-OmpA | ThermoFischer | Cat# MA5-47562 |
| Rabbit anti-human fibronectin | Merck | Cat# F3648 |
| Alexa488-conjugated goat anti-mouse IgG | Invitrogen | Cat# A-11001 |
| Alexa594-conjugated goat anti-rabbit IgG | Invitrogen | Cat# A-11008 |
| **Oligonucleotides and other sequence-based reagents** | | |
| Ab ATCC 17978 enolase wild-type gene | UniProtKB: A3M5Y1 | N/A |
| pET-30a(+) | GenScript | N/A |
| GenSmart Tool | GenScript | N/A |
| Primer: enolase IntUp | This study | N/A |
| Primer: enolase IntLw | This study | N/A |
| pGEM®-T Easy vector | Promega | Cat# A137A |
| T4 DNA ligase | Promega | Cat# M180A |
| *E. coli* DH5α | CABD | N/A |
| Primers upstream region enolase gene | This study | N/A |
| **Chemicals, enzymes and other reagents** | | |
| DAPI | Invitrogen | Cat# D1306 |
| DMSO | Merck | Cat# 472301 |
| Imipenem | Merck | Cat# I0160 |
| colistin | Merck | Cat# C4461 |
| ENOblock | MedChemExpress | Cat# HY-15858A |
| Luria–Bertani | Condalab | Cat# 1231.05 |
| Cation-adjusted Mueller-Hinton broth | Merck | Cat# 90922 |
| PBS | Applichem | Cat# A9201.0100 |
| Sheep blood agar | ThermoFisher | Cat# PB5306A |
| Ceftazidime | Merck | Cat# C3809 |
| Tigecycline | Merck | Cat# Y0001961 |
| Ampicillin | Merck | Cat# A6140 |
| IPTG | Genereas | Cat# Gen-s-02122 |
| Trizma | Merck | Cat# T6066 |
| HCl | Merck | Cat# 1758 |
| NaCl | Merck | Cat# S9899 |
| Imidazole | Merck | Cat# 56749 |
| MgCl$_2$ | Merck | Cat# 7791-18-06 |
| DTT | Thermo Scientific | Cat# R0862 |
| Enolase activity assay kit | Merck | Cat# MAK178-1KT |
| FBS | ThermoFisher | Cat# A4766801 |

| Reagent/resource | Reference or source | Identifier or catalog number |
|---|---|---|
| Ticarcillin | Merck | Cat# T5639 |
| DMEM | Biowest | Cat#L0104-500 |
| FM4-64 | Invitrogen | Cat# T13320 |
| SYTOX Green | Invitrogen | Cat# S7020 |
| Agarose | Invitrogen | Cat# 75510019 |
| Ethidium Homodimer-1 | ThermoFisher | Cat# E1169 |
| Vancomycin | Merck | Cat# SBR00001 |
| Gentamicin | Merck | Cat# G1397 |
| Amphotericin B | ThermoFisher | Cat# 15290-018 |
| HEPES | ThermoFisher | Cat# 15630-080 |
| RPMI | ThermoFisher | Cat# 22400089 |
| Penicillin–streptomycin | ThermoFisher | Cat# 15140122 |
| Phorbol 12-myristate 13-acetate | Merck | Cat# P8139 |
| 2-mercaptoethanol | Gibco | Cat# 21985023 |
| SDS | Merck | Cat# L3771 |
| DC Protein Assay Kit | Bio-Rad | Cat# 5000121 |
| Iodoacetamide | GE Healthcare | Cat# RPN6302 |
| TCEP | Meck | Cat# C4706 |
| Acetone | Merck | Cat# SHBR3623 |
| Triethylammonium bicarbonate | Merck | Cat# 18597 |
| Trypsin bovine | Promega | Cat# V5111 |
| Formic acid | Merck | Cat# F0507 |
| Acetonitrile | Merck | Cat# 1.00030 |
| Trifluoroacetic acid | Merck | Cat# 302031 |
| Phosphoenolpyruvate | Fluorochem | Cat# M02738 |
| Triton X-100 | ThermFisher | Cat# 13444259 |
| MTT | Merck | Cat# 475989 |
| Methanol | Merck | Cat# 34860 |
| Pork serum | Merck | Cat# P9783 |
| Bovine serum albumin | Merck | Cat# A8806 |
| Prolong Diamond Antifade Mounting | Invitrogen | Cat# P36965 |
| Fibrinogen | Merck | Cat# F3879 |
| Fibronectin | ThermoFisher | Cat# RP-43130 |
| Plasminogen | Meck | Cat# SRP6518 |
| Tripsin-EDTA | Biowest | Cat# X0930-100 |
| **Software** | | |
| Genedata Screener | Genedata | N/A |
| ImageQuant TL | GE Healthcare Life Sciences | N/A |
| SEQUEST® HT | ThermoFisher | N/A |
| Thermo Scientific™ Proteome Discoverer™ 2.2 | ThermoFisher | N/A |
| TopGO R package | Bioconductor | N/A |
| GraphPad Prism 9 | GraphPad | N/A |

| Reagent/resource | Reference or source | Identifier or catalog number |
|---|---|---|
| Autodock Tools 1.5.6 | Center for Computational Structural Biology | N/A |
| OpenBabel 3.1.1 | Open Babel | N/A |
| AlphaFold | Jumper et al, 2021 | N/A |
| ImageJ | https://imagej.net/ij/ | N/A |
| CellProfiler 4.0 software | CellProfiler | N/A |
| QuantileTransformer | Scikit-learn | N/A |
| Other | | |
| EU-OPENSCREEN chemical library | EU-OPENSCREEN | N/A |
| Echo 550® acoustic liquid handler | Beckman Coulter | N/A |
| EnVision™ microplate reader | Revvity | N/A |
| U-bottom microtiter plates | Deltalab | Cat# 900010 |
| 96-well plate | ThermoFisher | Cat# 130188 |
| Microtiter plate reader | Tecan Spark | N/A |
| Typhoon FLA 9000 laser scanner | GE Healthcare Life Sciences | N/A |
| Sonicator | Branson SFX550 698 sonifier, Emerson | N/A |
| Column HisTrap FF | Cytiva | N/A |
| ÄKTApure™ chromatography system | Cytiva | N/A |
| 3.5 K Slide-A-Lyzer Dialysis Cassette | Thermo Scientific | N/A |
| 10 K Amicon filter | Millipore | N/A |
| Auto-iTC200 isothermal titration calorimeter | MicroCal, Malvern-Panalytical | N/A |
| Humidified incubator | ThermoFisher | N/A |
| Isobaric standard tandem tag | ThermoFisher | Cat# 90064 |
| Tandem mass spectrometry | Bruker | N/A |
| OMIX C18 tips | Agilent Technologies | N/A |
| Nano-HPLC | Agilent Technologies | N/A |
| Thermo ScientificTM Easy nLC system | ThermoFisher | N/A |
| 50 cm C18 Thermo 802 Scientific™ EASY-Spray™ column. | ThermoFisher | N/A |
| Thermo ScientificTM Q Exactive™ Plus Orbitrap™ mass spectrometer | ThermoFisher | N/A |
| Fluorescence microscopy Zeiss Axio | Zeiss | N/A |

## Bacterial strains

A total of 32 clinical strains of Ab colistin-resistant ($n = 14$) (Valencia et al, 2009; López-Rojas et al, 2011b) or carbapenem-intermediate/resistant ($n = 18$) were collected from the "II Spanish Study of Ab GEIH-REIPI 2000–2010" multicenter study (GenBank Bioproject PRJNA422585), 13 clinical strains of E. coli colistin-resistant (Molina Panadero et al, 2025), 14 clinical strains of Klebsiella pneumoniae carbapenem-intermediate/resistant, and the reference strain Ab ATCC 17978 were used in this study.

HTS was performed with two reference strains of Ab and E. coli (Ab ATCC 17978 and ATCC 25922, respectively) and five clonally unrelated clinical strains: Ab9 (ST672), colistin- and tigecycline-susceptible; MDR Ab186 (ST208), colistin-susceptible, tigecycline-resistant; E. coli C1-7-LE (ST8671), colistin- and tigecycline-susceptible; and MDR E. coli MCR1+ (ST6108), colistin-resistant, tigecycline-susceptible (Herrera-Espejo et al, 2024); K. pneumoniae Kp10 KPC-2 producing (Jayol et al, 2018).

## EU-OPENSCREEN library and HTS validation

EU-OPENSCREEN provided a subset of 2,464 bioactive compounds from the ECBL pilot library (EU-OPENSCREEN: European Chemical Biology Library - Pilot Library) as 10 mM stock solutions in 100% DMSO. The library was screened in duplicate at a final concentration of 100 µM per well (1% DMSO). The antibacterial single-concentration HTS and dose–response susceptibility assays were conducted in 384-well plates, with bacterial cell density measured by optical density at 600 nm ($OD_{600}$).

A starting inoculum of $10^6$ colony-forming units (CFU)/mL was used, with an incubation time of 24 h for the Ab (ATCC 17978, 9 and 186) strains and E. coli (ATCC 25922, C17LE, and MCR1+) strains. Imipenem and colistin were used as internal controls for Ab and E. coli strains, respectively.

Compounds were distributed in duplicate on separate 384-well microtiter plates using an Echo 550® acoustic liquid handler (Beckman Coulter™, Indianapolis, IN) and inoculated with bacteria to a final concentration of $10^6$ CFU/mL, with a total assay volume of 25.25 µL. Plates were incubated with shaking for 24 h at 37 °C. Bacterial growth was measured by reading the $OD_{600}$ using an EnVision™ microplate reader (Revvity, Waltham, MA). The activity of the compounds was expressed as the percentage of bacterial growth inhibition, and it was calculated using the following normalization:

$$\%Inhibition = 100 \times \left\{ 1 - \frac{\left[(T_{f\,Sample} - T_{O\,Sample}) - (T_{f\,Blank} - T_{O\,Blank})\right]}{\left[(T_{f\,Growth} - T_{O\,Growth}) - (T_{f\,Blank} - T_{O\,Blank})\right]} \right\}$$

Where, $T_{0\,Sample}$ is the absorbance of the strain growth in the presence of compound measured at time zero, $T_{f\,Sample}$ is the absorbance of the strain growth in the presence of compound measured at final time, $T_{0\,Growth}$ is the absorbance of the strain growth in the absence of compound measured at time zero, $T_{f\,Growth}$ is the absorbance of the strain growth in the absence of compound measured at final time, $T_{0\,Blank}$ is the absorbance of the broth medium (blank) measured at time zero, $T_{f\,Blank}$: the absorbance of the broth medium (blank) measured at final time; $T_0$ is Time at 0 h and $T_f$ is Time at 24 h.

**Blank:** is composed of 25 µL of MHII and 0.25 µL of DMSO 20%
**Growth:** is composed of 25 µL of bacterial inoculum and 0.25 µL of DMSO 20%.
**Sample:** studied compound.

The Genedata Screener software (Genedata, Inc., Basel, Switzerland) was used to process and analyse all the screening data. Their reproducibility and sensitivity were supported by the statistical

values derived from all the experiments performed. Also, MIC 90% (MIC required to inhibit 90% of the growth of a microorganism) value for every Dose-Response Curve of reference antibiotic compounds (imipenem and colistin) was determined to assess consistent reproducible activity data within assay plates and between experiments. This software was used to calculate quality control parameters such as RZ' factor (RZ' factor ≥0.5), and signal/background ratio (S/B). The Z'factor predicts the robustness of an assay by considering the mean and standard deviation of both positive and negative controls (Zhang et al, 1999). The robust Z' factor (RZ' factor) is based on the Z' factor, but standard deviations and means are replaced by the robust standard deviations and medians, respectively.

The selected active compound was prioritized because it demonstrated the most consistent in vitro activity across multiple assays, making it the most suitable candidate for detailed mechanistic and time-kill studies.

## in vitro susceptibility testing

The MIC of ENOblock and colistin were determined against all studied Ab, *E. coli* and *K. pneumoniae* strains in two independent experiments using the broth microdilution method, following the standard guidelines of the European Committee on Antimicrobial Susceptibility Testing (EUCAST, 2023). Briefly, a $5 \times 10^5$ CFU/mL inoculum of each strain was cultured in Luria–Bertani (LB) and cation-adjusted Mueller-Hinton broth, and then added to U-bottom microtiter plates (Deltalab, Spain) containing ENOblock and colistin, respectively. The plates were incubated for 18 h at 37 °C. *Pseudomonas aeruginosa* ATCC 27853 was used as the positive reference control strain.

## Antimicrobial selection pressure

The diluted Ab ATCC 17978 inocula ($10^5$ CFU/mL) were incubated with sub-inhibitory concentrations of ENOblock, corresponding to dilutions onefold below the MIC, at 37 °C for 24 h. Bacterial concentration was determined in cultures showing positive growth, which were then re-adjusted to $10^5$ CFU/mL for further incubation with a twofold increased concentration of ENOblock. These steps were repeated until an ENOblock concentration was reached that completely inhibited bacterial growth.

## Time kill kinetic assays

To determine the bactericidal activity, duplicate time-kill curves were performed for Ab ATCC 179178 and Ab CR17 strains, for *E. coli* MCR1$^+$ (colistin-resistant) and *K. pneumoniae* Kp10 (carbapenem-resistant) strains, and for *E. coli* MG1655 and its isogenic deficient in enolase (Ec Δ*eno*) strain (Murashko and Lin Chao, 2017), as previously described (Miró-Canturri et al, 2020). An initial inoculum of $5 \times 10^5$ CFU/mL was added to LB in the presence of 1×MIC, 2×MIC and 4×MIC of ENOblock. A drug-free broth was evaluated in parallel as a control. Tubes of each condition were incubated at 37 °C with shaking, and viable counts were determined by serial dilution at 0, 2, 4, 8, and 24 h. Viable counts were determined by plating 100 μL of the control, test cultures, or the respective dilutions at the indicated times onto sheep blood agar plates (ThermoFisher, Spain). Plates were incubated for 24 h at 37 °C, and after colony counts, the $\log_{10}$ of viable cells (CFU/mL)

was determined. Bactericidal activity was defined as a reduction of ≥3 $\log_{10}$ CFU/mL from the initial inoculum.

## EIIP/AQVN filter

Specific recognition and targeting between interacting biological molecules at distances >5 Å were determined by the average quasi-valence number (AQVN) and the Electron-ion interaction potential (EIIP) derived from the general model pseudopotential (Veljkovic and Slavic, 1972)

$$EIIP = 0.25 \, Z^* \sin(1.04\pi Z^*) \quad (1)$$

where Z* is the AQVN determined by:

$$Z^* = \sum m(ni \, Zi/N) \quad (2)$$

where Zi is the valence number of the *i*th atomic component, ni is the number of atoms of the *i*th component, m is the number of atomic components in the molecule, and N is the total number of atoms. EIIP values are computed using Eqs. (1) and (2) and are expressed in Rydberg units (Ry).

AQVN and EIIP are unique physical properties that characterize long-range interactions between biological molecules among the 3300 molecular descriptors currently in use (Duran et al, 2009). It has been shown that the EIIP and AQVN of organic molecules strongly correlate with their biological activity (mutagenicity, carcinogenicity, toxicity, antibiotic and cytostatic activity, etc.) (Veljkovic, 1980).

## Checkerboard assay

The assay was performed on a 96-well plate in duplicate as previously described (Miró-Canturri et al, 2020). Colistin, imipenem, ceftazidime or tigecycline were twofold serially diluted along the *x* axis, whereas ENOblock was twofold serially diluted along the *y* axis to create a matrix, where each well consists of a combination of both agents at different concentrations. Bacterial cultures grown overnight were then diluted in saline to 0.5 McFarland turbidity, followed by 1:50 further dilution LB and inoculation on each well to achieve a final concentration of approximately $5.5 \times 10^5$ CFU/mL. The 96-well plates were then incubated at 37 °C for 18 h and examined for visible turbidity. The FIC of the colistin, imipenem, ceftazidime or tigecycline was calculated by dividing the MIC of colistin, imipenem, ceftazidime or tigecycline in the presence of ENOblock by the MIC of colistin, imipenem, ceftazidime or tigecycline alone. Similarly, the FIC of ENOblock was calculated by dividing the MIC of ENOblock in the presence of imipenem, ceftazidime or tigecycline alone. The FICI was the summation of both FIC values. FICI values of ≤0.5 and >5 were interpreted as synergistic and non-synergistic, respectively.

## Bacterial growth curves

To confirm the synergy between ENOblock and colistin, bacterial growth curves of the Ab ATCC 17978 and Ab CR17 strains were performed in duplicate in 96-well plate (ThermoFisher, Spain). An initial inoculum of $5 \times 10^5$ CFU/mL was prepared in LB in the

presence of colistin (0.12 or 1 mg/L) and ENOblock (8 mg/L) separately or together. A drug-free broth was evaluated in parallel as a control. Plates were incubated at 37 °C with shaking, and bacterial growth was monitored for 24 h using a microtiter plate reader (Tecan Spark, Austria). The result of bacterial density absorbance over 24 h did not contradict the classical MIC definition but rather highlights the complexity of ENOblock's activity and the possible involvement of tolerance or persistence phenomena.

## Bacterial cytological profiling

Overnight cultures of Ab ATCC 17978 were diluted 1:100 in LB broth and incubated on a roller at 30 °C until the $OD_{600}$ reached 0.2. ENOblock at 16 mg/L (2×MIC) was added to the bacterial cultures prior to observation under a fluorescence microscope at 60 min time points. Then, the cultures were stained with 2 mg/L FM4-64, 2 mg/L DAPI, and 0.5 μM SYTOX Green. The bacterial cells were then harvested by centrifugation at 6000 ×g for 1 min and resuspended in 1/10 of the original volume. A small amount of the concentrated bacterial cultures was placed on an agarose pad (1.2% agarose in 10% LB broth) on concave glass slides for microscopy. Consistent experimental settings and imaging parameters were maintained throughout all experiments included in the statistical analysis of the antibiotic training sets.

## SYTOX Green assay

Overnight cultures of *A. baumannii* ATCC 17978 were diluted 1:100 in LB broth and incubated at 30 °C on a roller until reaching an $OD_{600}$ of 0.2. ENOblock was then added at various concentrations: 8 mg/L (1×MIC), 16 mg/L (2×MIC), and 32 mg/L (4 × MIC), along with colistin at 3.125 mg/L (5×MIC). Samples were collected after 10, 30, and 60 min. Cells were stained with FM4-64 (2 mg/L), DAPI (2 mg/L), and SYTOX Green (0.5 μM), centrifuged at 6000 × g for 1 min, and resuspended in one-tenth of the original volume. A small aliquot was mounted on 1.2% agarose pads (in 10% LB) for microscopy. Experimental and imaging conditions were kept consistent across all datasets.

### Image data analysis

For BCP analysis, the raw images from the fluorescent microscope were preprocessed using ImageJ software (Schindelin et al, 2012), and cell features were extracted with CellProfiler 4.0 software (McQuin et al, 2018). After data extraction, the ENOblock-treated cell profiles were analyzed alongside antibiotic-treated cell profiles from previous studies using an analysis pipeline from previous research (Samernate et al, 2023). Briefly, the data was transformed using QuantileTransformer (Jiang et al, 2020), and outliers were removed using hierarchical density-based spatial clustering of applications with noise (HDBSCAN) (Campello et al, 2015). This analysis utilized a morphological feature set from previous studies (Samernate et al, 2023). Finally, the dimension of the data set was reduced and visualized through data clustering using pairwise controlled manifold approximation (PaCMAP) (Wang et al, 2021). All antibiotic control data used for profile comparison, including images (Fig. 3A) and single-cell profiles (Fig. 3B), were derived from previously published data (Samernate et al, 2023). For SYTOX green assays, the intensity within nucleoid outlines was analyzed for all treatment conditions (number of cells >30 per replicate, three

biological replicates). The average intensity was calculated from three biological replicates per condition.

## Membrane permeability assay

Membrane permeability of *E. coli* MCR1$^{+}$ and *K. pneumoniae* Kp10 strains was assessed in the presence of ENOblock following a previously established protocol (Miró-Canturri et al, 2020). Briefly, bacterial cells were grown in LB broth overnight and bacterial pellets were harvested by centrifugation at 4600 × g for 15 min, washed with 1X PBS, and centrifuged again under the same conditions. The bacterial pellets OD were adjusted to 0.2 and incubated with 0.5×MIC ENOblock. Then, they were resuspended in 100 μL of 1× PBS containing 10 μL of Ethidium Homodimer-1 (EthD-1) (ThermoFisher, Spain). After 10 min of incubation, 100 μL of the suspension was transferred to a 96-well plate, and fluorescence was measured over 3 h using a Typhoon FLA 9000 laser scanner (GE Healthcare Life Sciences, USA). Fluorescence intensity was quantified using ImageQuant TL software (GE Healthcare Life Sciences, USA). PBS with 1% DMSO was used as negative control.

## Docking and molecular modelling

The crystal structure of enolase C-terminal of Ab was predicted by AlphaFold (Jumper et al, 2021). This target protein was modified using Autodock tools 1.5.6 software; including ligand and water removal, hydrogen addition, and incorporation of Kollman charges. The resultant files were saved in pdbqt format. The ligand "ENOblock" was downloaded from Pubchem (https://pubchem.ncbi.nlm.nih.gov/) in SMILES format (https://pubchem.ncbi.nlm.nih.gov/compound/24012277). Openbabel 3.1.1 software was used to create the 3D chemical structure, energy minimization, hydrogen atoms addition and establishment of a neutral pH. Resulting ligand was saved in MOL.2 format. Gasteiger charges computation for the ligand structure was performed using Autodock tools 1.5.6, with the output saved in pdbqt format. Autodock tools 1.5.6 was used for docking and subsequent analysis of the docking results.

## Protein expression and purification of recombinant AbEnolase

The Ab ATCC 17978 enolase wild-type gene (UniProtKB: A3M5Y1), enolase mutant #1 (D207A: aspartate replaced by alanine at position 207), enolase mutant #2 (S371A: serine replaced by alanine at position 371) and enolase double mutant (D207A and S371) were synthesized and cloned into the pET-30a(+) vector with an N-terminal His6 tag using *NdeI* and *XhoI* restriction sites by GenScript. Prior to synthesis, the codon usage of the gene was optimized using the GenSmart Tool. *wild-type* and mutated AbEnolase overexpression were performed in *E. coli* Rosetta™ DE3 pRare (Merck Millipore, Spain). A single colony from freshly transformed cells was incubated overnight at 37 °C in LB supplemented with ampicillin, with shaking at 180 rpm. The culture was diluted 1:100 into fresh medium and grown at 37 °C until an $OD_{600}$ of 0.5 was reached. Protein expression was induced with 0.5 mM isopropyl-β-D-thiogalactopyranoside (IPTG) (Merck, Spain) for 3 h at 30 °C. Cells were harvested by centrifugation, washed with 50 mM Tris-HCl pH 8 and the pellet was stored at

$-80\,°C$ until use. The cell pellet was resuspended in Lysis Buffer (50 mM Tris, pH 8.0, 0.4 M NaCl, 5 mM imidazole, 5 mM $MgCl_2$) and disrupted by sonication (Branson SFX550 sonifier, Emerson, Spain). Cell debris was removed by centrifugation at $10,000\,g$ for 25 min. The soluble lysate was filtered through a 0.45-µm filter and applied to a nickel affinity column HisTrap FF (Cytiva, USA) equilibrated with Lysis Buffer using an ÄKTApure™ chromatography system (Cytiva, USA). Enolase was eluted using a 10-column volume gradient of lysis buffer supplemented with 250 mM imidazole. Fractions containing enolase were identified by SDS-PAGE, pooled, and dialyzed overnight at $4\,°C$ against Dialysis Buffer (25 mM Tris, pH 7.5, 100 mM NaCl, 5 mM $MgCl_2$, 0.5 mM dithiothreitol [DTT]) using a 3.5 K Slide-A-Lyzer Dialysis Cassette (Thermo Scientific, Spain). Before storage at $-80\,°C$, the protein was concentrated using a 10 K Amicon filter (Millipore, Spain) and quantified by densitometric analysis. Aliquots were prepared at a concentration of 5 mg/mL and stored at $-80\,°C$.

## Isothermal titration calorimetry (ITC)

Interaction of the *wild-type* Abenolase and mutants D207A, S371A and D207A + S371S with ENOblock was analyzed with an Auto-iTC200 isothermal titration calorimeter (MicroCal, Malvern-Panalytical) at a constant temperature of $25\,°C$. Protein solution was used at a final concentration of 75 µM in buffer 25 mM Tris pH 7.5, 100 mM NaCl, 5 mM $MgCl_2$. ENOblock solution at 1 mM in the same buffer was injected following programmed sequences of 13 injections of 3 µL with a spacing of 150 s and a stirring speed of 750. Three independent calorimetric titrations ($n=3$) were performed for each interaction, in order to assess reproducibility, as well as to estimate average values for the dissociation constants and their errors. The dissociation constants were obtained through nonlinear least squares regression analysis of the experimental data to a model considering a single ligand binding site. Appropriate controls (buffer into buffer; ligand dilution into buffer; buffer into enolase) were conducted to obtain a baseline for each experiment and determine the heat of dilution/mixing.

## Generation of enolase knockout from Ab ATCC 17978

To construct an *enolase* knockout from Ab ATCC 17978, we followed the protocol described previously (Smani et al, 2013; Smani et al, 2014). Briefly, an internal *enolase* 482-bp fragment obtained by PCR amplification with the primers enolase IntUp and enolase IntLw (Appendix Table S1) was cloned into pGEM-T (Promega, Spain) to give plasmid enolase-pGEM-T by using T4 DNA ligase (Promega, Spain). The resulting construct incorporated into *E. coli* DH5α was purified and electroporated into Ab ATCC 17978 to knock out the *enolase* gene. Transformants were selected on LB agar plates containing 80 µg/mL ticarcillin. The *enolase* gene disruption within the resulting strain, designated Ab Δ*eno*, was confirmed by PCR using a combination of primers matching the upstream region of *enolase* gene and the pGEM-T Easy vector.

## Bacterial enolase activity assay

Enolase activity was determined in three independent experiments following the instructions of Enolase Activity Assay kit (Merck,

Spain). Briefly, a bacterial inoculum of $10^6$ CFU/mL, after centrifugation, was incubated with a master mix composed by enolase substrate, peroxidase substrate, and the necessary converters for the colorimetric reaction at $25\,°C$. Enolase enzymatic activity, reported in mU/mL, was measured at $OD_{570nm}$ every 2–3 min for 1 h using a microtiter plate reader (Tecan Spark, Austria) and calculated using the specific formula: $\Delta A570 = (A570)_{final} - (A570)_{initial}$,

and the following equation:

$$enolase\ Activity = (B \times Sample\ Dilution\ Factor)/(Reaction\ Time \times V)$$

where: B = Amount (nmol) of $H_2O_2$ generated between $T_{initial}$ and $T_{final}$, Reaction Time = $T_{final} - T_{initial}$ (minutes), and V = volume of sample (mL) added to the well.

## Bacterial growth curves

To determine the antibacterial of ENOblock against *wild-type* and enolase-deficient *A. baumannii*, bacterial growth curves of the *A. baumannii* Ab ATCC 17978 and its isogenic deficient in enolase (Ab Δ*eno*) strain in similar conditions (see above). Bacteria was incubated in the presence of $1\times$, $2\times$ and $4\times$ MIC of ENOblock.

## Human cell culture

HeLa cells were grown in DMEM supplemented with 10% heat-inactivated fetal bovine serum (FBS), vancomycin (50 mg/L), gentamicin (20 mg/L), and amphotericin B (0.25 mg/L) (Invitrogen, Spain), and 1% HEPES in a humidified incubator with 5% $CO_2$ at $37\,°C$. The HeLa cells were routinely passaged every 3 or 4 days. Immediately before infection, HeLa cells were washed three times with prewarmed PBS and further incubated in DMEM without FBS and antibiotics (Parra-Millán et al, 2018). The human monocytic leukemia cell line THP-1 (ATCC, LGC Standards, Spain) was cultured in RPMI medium (ThermoFisher, Spain) supplemented with 10% FBS (ThermoFisher, Spain), 1% penicillin–streptomycin (Gibco, Spain), 1% HEPES, and 0.05 mM 2-mercaptoethanol (Gibco, Spain). Cells were maintained in a humidified incubator at $37\,°C$ with 5% $CO_2$ and were routinely passaged every 3 days.

## Differentiation of THP-1 monocytes into macrophages

To differentiate THP-1 monocytes into macrophages, the cells were seeded in RPMI medium supplemented with 10% FBS, 1% penicillin–streptomycin, and 40 ng/mL phorbol 12-myristate 13-acetate (PMA) (Merck, Spain). The cells were incubated at $37\,°C$ with 5% $CO_2$ for 2 days (Díez-Sainz et al, 2023). Before infection, macrophages were washed three times with prewarmed PBS and further incubated in RPMI medium without FBS or antibiotics (Parra-Millán et al, 2018).

## Tandem mass Tag (TMT) assay and analysis

Infected and non-infected HeLa and macrophage cells were lysed in a buffer containing 4% SDS, 100 mM Tris (pH 7.6), and water. Cell membranes were disrupted via sonication for 1.5 min with 5-s intervals on ice. The supernatant was collected after centrifugation at $4\,°C$ for 5 min. Protein concentration was determined using a DC

Protein Assay Kit (Bio-Rad, Spain). Samples were incubated with 10 mM TCEP in 100 mM Tris (pH 7.6) at 55 °C for 1 h and then treated in darkness with 375 mM iodoacetamide in 100 mM TrisHCl (pH 7.6) for 30 min. Proteins were precipitated with acetone and stored at −20 °C for at least 4 h. Samples were centrifuged at 16,000 × $g$ at 4 °C for 10 min, and the acetone was discarded by pellet desiccation. To analyse differential protein expression between bacteria alone and bacteria in contact with HeLa or macrophage cells, an isobaric standard tandem tag (ThermoFisher, Spain) was employed. Samples were analyzed via tandem mass spectrometry (MS/MS) (Bruker, USA) at the Proteomics Facility of the University Pablo de Olavide (Seville, Spain). Protein pellets were resuspended in triethylammonium bicarbonate and digested overnight at 37 °C using bovine trypsin (Sequencing Grade Modified Trypsin, Promega) in a ratio 1:12 enzyme-substrate. Reaction was stopped using formic acid to 0.5% and samples were labelled with the isobaric tags following manufacturer instructions, using channels 126, 127, 128, 129, 130 and 131. In all, 5 µg of every tagged sample were mixed in a single sample tube. OMIX C18 tips (Agilent Technologies) were used for concentrating and desalting tagged peptide extracts. Sample was dried and resuspended in 0.1% trifluoroacetic acid and injected in nano-HPLC system. Protein digested samples were separated in a Thermo ScientificTM Easy nLC system using a 50 cm C18 Thermo Scientific™ EASY-Spray™ column. The following solvents were employed as mobile phases: Water 0.1% Formic Acid (phase A) and Acetonitrile, 20% $H_2O$, 0.1% Formic Acid (phase B). Separation was achieved with an acetonitrile gradient from 10% to 35% over 360 min, 35% to 100% over 1 min, and 100% B over 5 min at a flow rate of 200 nL/min.

A Thermo ScientificTM Q Exactive™ Plus Orbitrap™ mass spectrometer was used for acquiring the top 10 MS/MS spectra in DDA mode. LC-MS data were analysed using the SEQUEST® HT search engine in Thermo Scientific™ Proteome Discoverer™ 2.2 software considering the modifications: static carbamidomethylation (C), dynamic oxidation (M) and dynamic N-terminus acetylation. Data were searched against the Uniprot *Acinetobacter_baumanii* or *Homo sapiens* (sp_canonical TaxID=9606) (v2024-03-27) protein databases and results were filtered using a 1% protein FDR threshold. Quantifiable proteins were those identified by >2 peptides with a confidence level >95%, a *P* value < 0.05, and an error factor <2 for each reference tag. Proteins with a fold change of <−1 or >1 were classified as downexpressed or overexpressed, respectively. Sample-to-sample distances were visualized through principal component analysis (PCA). Functional enrichment analyses included Clusters of Orthologous Groups (COG) and Gene Ontology (GO) terms that were assigned using EggNOG mapper. Enriched GO terms were identified using the enricher function in the topGO R package.

## Adhesion and invasion assays

HeLa and macrophage cells were infected with Ab ATCC 17978, Ab CR17 and Ab Δ*eno* strains, and for *E. coli* MCR1+ and *K. pneumoniae* Kp10 strains at a concentration of $1 \times 10^8$ CFU/mL, in the absence and presence of 1xMIC of ENOblock or 4 mM phosphoenolpyruvate at a multiplicity of infection (MOI) of 100. The infection was carried out for 2 h with 5% $CO_2$ at 37 °C in three independent experiments. After that, the infected HeLa and macrophage cells were washed five times with prewarmed PBS and lysed with 0.5% Triton X-100. Diluted lysates were plated onto

LB agar and incubated at 37 °C for 24 h to enumerate the developed colonies and determine the number of bacteria that had attached to the HeLa and macrophage cells (Parra-Millán et al, 2018).

In addition, to determine the number of colonies that entered inside the HeLa and macrophage cells, the infected cells for 2 h were washed with phosphate-buffered saline and incubated for additional 30 min in the presence of DMEM or RPMI plus gentamicin (256 µg/mL), in order to kill the bacteria, present in the area. Then, the wells were washed with phosphate-buffered saline to remove gentamicin. The number of colonies that entered inside HeLa and macrophage cells was determined as described above (Parra-Millán et al, 2018).

## Cellular toxicity of ENOblock

HeLa cells and macrophages differentiated from THP-1 cells (Díez-Sainz et al, 2023) were incubated with ENOblock at different concentrations ranged from 0.5 to 256 mg/L for 24 h with 5% $CO_2$ at 37 °C. Prior the evaluation of the ENOblock cytotoxicity, HeLa and macrophage cells were washed three times with prewarmed PBS 1×. Subsequently, quantitative cytotoxicity was evaluated by measuring the mitochondrial reduction activity using the 3-(4,5-dimethylthiazol-2-yl)-2,5-diphenyltetrazolium bromide (MTT) assay as described previously (Smani et al, 2013). The percentage of cytotoxicity was calculated from the absorbance at 570 nm as follows: [(Absorbance 570 nm of treated cells/Absorbance 570 nm mean of untreated cells) × 100]. The cytotoxic concentration 50% ($CC_{50}$) value was determined using GraphPad Prism 9.

## Immunofluorescence

Immunofluorescence assay was performed as described previously (Vila-Farrés et al, 2017). Briefly, the HeLa cells plated on coverslips were incubated with Ab CR17 and *E. coli* MCR1+ strains for 2 h, and later were incubated with ENOblock (0 and 1×MIC, 30 min) at 5% $CO_2$ and 37 °C. Bacterial cells were removed, and HeLa cells were washed five times with cold PBS. HeLa cells on the coverslips were fixed in methanol for 8 min at −20 °C, permeabilized with 0.5% Triton X-100 and blocked with 20% pork serum in PBS. Primary antibodies: mouse anti-OmpA of Ab (ThermoFischer, Spain) and rabbit anti-human fibronectin (Merck, Spain) were used at dilution of 1:25 sand 1:50, respectively, in PBS containing 1% bovine serum albumin (BSA) for 2 h. After washing with PBS, the coverslips were incubated with their respective secondary antibodies: Alexa488-conjugated goat anti-mouse IgG, and Alexa594-conjugated goat anti-rabbit IgG (Invitrogen, Spain) at dilution of 1:50, 1:50 and 1:100, respectively, in PBS containing 1% BSA for 1 h. The fixed coverslips were incubated for 10 min at room temperature with DAPI (Applichem, Germany) (0.5 µg/mL), washed with PBS, mounted in fluorescence mounting medium "Prolong Diamond Antifade Mounting" (Invitrogen, Spain), and visualized using fluorescence microscopy Zeiss Axio Imager 2 (Zeiss, Germany).

## Informational spectrum method

The Informational Spectrum Method (ISM), a virtual spectroscopy approach, was employed to investigate protein-protein interactions in silico to complement experimental data. ISM converts protein or DNA sequences into numerical signals derived from the EIIP of

their constituent amino acids or nucleotides. EIIP values capture electronic properties that mediate long-range molecular interactions (5–1000 Å) (Veljkovic et al, 2011). This virtual spectroscopy method enables functional analysis of protein sequences without requiring prior experimental input. The ISM extension for small molecules (ISM-SM) uses similar principles, translating small molecules represented in SMILES notation into EIIP-based arrays corresponding to atomic groups (Sencanski et al, 2022). By identifying shared frequencies in computational spectra, ISM-SM predicts potential interactions between proteins and small molecules and highlights likely binding regions on the protein. In this study, ISM-SM was applied to identify shared informational features among Ab enolase (UniProt: sp|B0VQI4| ENO_ACIBS), human plasminogen (UniProt: sp|P00747| PLMN_HUMAN), human fibrinogen (UniProt: sp|P02679| FIBG_HUMAN), human fibronectin (UniProt: sp|P02751| FINC_HUMAN) and ENOblock (PubChem CID: 24012277). In the next step, the plasminogen-enolase, fibrinogen–enolase, and fibronectin-enolase complexes were modelled. The obtained sequences of enolase and these extracellular matrix (ECM) proteins were sent to Alphafold3 to build a protein-protein complex.

## Coating human host proteins on wells

The coating of human host proteins (fibrinogen, fibronectin, and plasminogen) onto 96-well plates was performed as previously described (Smani et al, 2012). Wells were coated overnight at 4 °C with 125 µL of PBS containing 1.25 µg of plasmatic plasminogen, fibronectin or fibrinogen (10 µg/mL), or with bovine serum albumin (BSA) at 20 µg/mL as a control. After coating, wells were washed four times with 125 µL of 1% (w/v) BSA in PBS and then blocked for 1 h at room temperature with 125 µL of 1% BSA in PBS. Immediately before the addition of bacteria, wells were washed six times with sterile PBS.

## Human host proteins-AbEnolase binding assay

Binding assays between human host proteins and purified Ab enolase (AbEnolase) (Appendix Fig. S1) were performed as previously described with some modifications (Smani et al, 2012). Ab ATCC 17978 cells, grown overnight at 37 °C in LB broth, were resuspended in PBS, centrifuged at $5000 \times g$ for 10 min, and washed twice with sterile PBS. Fifty microliters of increasing concentrations of AbEnolase (10, 50, and 100 mg/L) were added to wells pre-coated with plasminogen, fibronectin or fibrinogen, and incubated for 2 h at room temperature. Wells were then rinsed six times with sterile PBS, and 50 µL of bacterial suspension was added to the wells for a 2 h incubation to facilitate bacterial adhesion. Non-adherent bacteria were removed by washing the wells six times with sterile PBS. Adherent bacteria were collected by adding 125 µL of sterile PBS containing 0.5% Triton X-100. The lysates were diluted and plated onto LB agar, followed by incubation at 37 °C for 24 h to enumerate developed colonies. This allowed determination of bacterial adhesion to plasminogen, fibronectin and fibrinogen in the presence of AbEnolase.

## Human host proteins-bacteria-ENOblock binding assay

Human host proteins-bacteria-ENOblock binding assays were performed as described above with some modifications. The Ab ATCC

### The paper explained

#### Problem
The emergence of infections caused by multidrug resistant Gram-negative bacilli such Acinetobacter baumannii (Ab) is a well-recognized global health threat, urgently requiring effective solutions. Last-resort treatments, such as colistin, are increasingly ineffective in many cases, leading to mortality in hospitalized patients who acquire these infections.

#### Results
We show that ENOblock presents strong antibacterial activity against multidrug-resistant Ab, with an $MIC_{50}$ of 16 mg/L, better than colistin and carbapenems. We demonstrate that ENOblock rapidly damages bacterial membranes and causes unique cellular changes, suggesting a distinct mechanism of action. Using docking studies and mutant analysis, we confirm that enolase is the target of ENOblock. We reveal that ENOblock also acts synergistically with colistin, reduces Ab interaction and invasion in host cells, and improves survival in infected Galleria mellonella larvae.

#### Impact
These preclinical data support the selection of ENOblock as a promising candidate for antimicrobial development, with the potential to address the urgent threat posed by infections caused by Ab.

17978 strain was grown overnight at 37 °C in LB, resuspended in PBS, and collected by centrifugation at $5000 \times g$ for 10 min. The bacteria were washed twice in sterile PBS and resuspended in the same sterile buffer. A 50 µL of bacterial suspension was incubated with 0.5× and 1×MIC ENOblock for 30 min, washed twice in sterile PBS, then added to plasminogen-, fibronectin- or fibrinogen-coated wells and incubated 2 h at room temperature for bacterial adsorption. The determination of bacterial adhesion to plasminogen, fibronectin and fibrinogen in the presence of ENOblock was performed as described above. In addition, we determined the concentration of Ab ATCC 17978 strain in presence of 0.5× and 1×MIC ENOblock during 30 min.

## Galleria mellonella infection model

G. mellonella infection model with Ab ATCC 17978 strain was established by haemocoel bacterial inoculation. Briefly, caterpillars obtained from Artroposfera (Toledo, Spain) were inoculated with 10 µL of the bacterial suspensions, which were incubated for 20–24 h in LB at 37 °C. The minimal bacterial lethal dose 100 (MLD100) and LD50 were determined by inoculating various groups of larvae (8 G. mellonella per group) with decreasing amounts of Ab ATCC 17978 strains inocula from $10^6$ to $10^2$ CFU/mL, and monitoring the survival of the larvae for 7 days.

## Therapeutic efficacy of ENOblock alone and in combination with colistin in G. mellonella infection model

The efficacy of the ENOblock treatment alone and in combination with colistin were tested in G. mellonella survival assay as previously described (Martínez-Guitián et al, 2020). Caterpillars were injected by the 10 µL of suspension containing MLD of Ab ATCC 17978. Treatment with 0.12 mg/L of colistin, 2× and 4xMIC of ENOblock alone and in combination with 0.12 mg/L of colistin

were injected 1 h post-infection. Groups of larvae injected with 10 µL of sterile PBS or ENOblock (1×, 2× and 4×MIC) were included as control. After inoculation, the larvae were incubated at 37 °C in the dark and death was assessed over 3 days.

## Statistical analysis

Group data are presented as means ± standard errors of the means (SEM). The Student $t$ test was used to determine differences between means using the GraphPad Prism 9 (version 9.3.1; GraphPad Software, LLC). For the *G. mellonella* survival model, a Kaplan–Meier test was performed to determine the difference between mortality rates. $P < 0.05$ was considered significant.

## Data availability

The data supporting the findings of this study are publicly available in https://doi.org/10.5281/zenodo.15533882.

The source data of this paper are collected in the following database record: biostudies:S-SCDT-10_1038-S44321-025-00331-2.

## Peer review information

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

## Acknowledgements

This research was funded by the Ministerio de Ciencia e Innovación, Agencia Estatal de Investigación, Fondo Europeo de Desarrollo Regional, MCIN/AEI/10.13039/501100011033/FEDER, UE (Grant PID2022-136357OBI00), and (Grant CEX2020-001088-M-20-5), by the Consejería de Universidad, Investigación e Innovación de la Junta de Andalucía (Grant ProyExcel_00116), by the National Research Council of Thailand (NRCT) and Mahidol University: N42A650368, and by the Ministry of Science, Technological Development and Innovation of the Republic of Serbia (Grant number 451-03-136/2025-03/200017). We acknowledge EU-OPENSCREEN ERIC for providing its compound collection and Fundación MEDINA HTS antimicrobial screening platform to support the discovery of the antibacterial activity of the compound described in the presented work. This article is based upon work from COST Action EURESTOP, CA21145, supported by COST (European Cooperation in Science and Technology). AMR is supported by a doctoral fellowship PRE2022-104318, from the Agencia Estatal de Investigación, Ministerio de Ciencia e Innovación.

## Author contributions

**Irene Molina Panadero**: Formal analysis; Investigation; Methodology. **Antonio Moreno Rodríguez**: Formal analysis; Investigation; Methodology. **Angela Rey Hidalgo**: Formal analysis; Investigation; Methodology. **Mercedes de la Cruz**: Formal analysis; Investigation; Methodology. **Pilar Sánchez**: Formal analysis; Investigation; Methodology. **Laura Tomás Gallardo**: Formal analysis; Investigation; Methodology. **Thanadon Samernate**: Formal analysis; Funding acquisition; Methodology. **Milan Sencanski**: Formal analysis; Investigation; Methodology. **Sanja Glisic**: Investigation; Methodology; Writing—review and editing. **Olga Genilloud**: Investigation; Writing—review and editing. **Poochit Nonejuie**: Investigation; Writing—review and editing. **Antonio J Pérez-Pulido**: Investigation; Writing—review and editing. **Abdelkrim Hmadcha**: Formal analysis; Investigation; Writing—review and editing. **Younes Smani**: Conceptualization; Supervision; Funding acquisition; Writing—review and editing.

Source data underlying figure panels in this paper may have individual authorship assigned. Where available, figure panel/source data authorship is listed in the following database record: biostudies:S-SCDT-10_1038-S44321-025-00331-2.

## Disclosure and competing interests statement

The authors declare that the research was conducted in the absence of any commercial or financial relationships that could be construed as a potential conflict of interest.

# Expanded View Figures

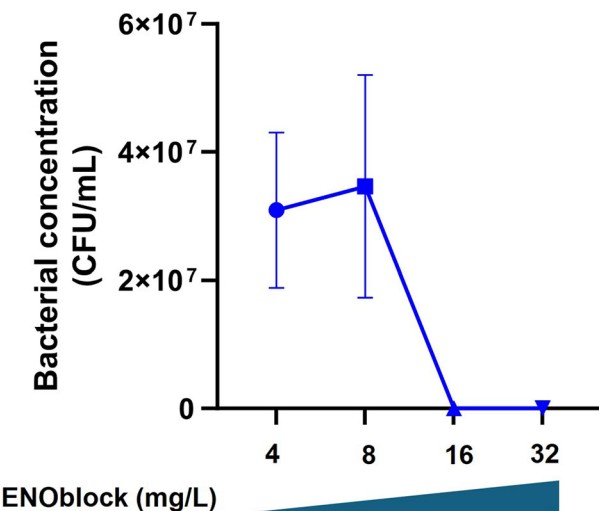

**Figure EV1.  ENOblock selective pressure.**

*A. baumannii* Ab ATCC 17978 concentrations after incubation with increasing concentrations of ENOblock.

**A**

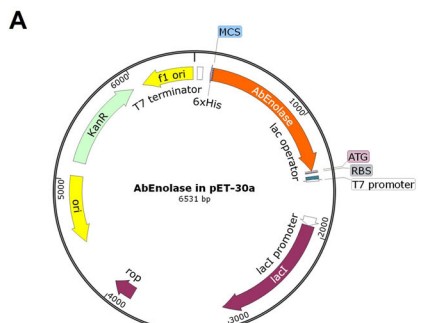

**B**

### Enolase sequence (42 kDa)

ATGTTATTTTGGAGTCCGGCGTGGTGGGCCGTGCCTGCGCACCGAGCGGTGCGAGCACTGGTTCACGCGAAGCTTTGGAGTTA
CGTGACGGTGACAAATCTAGATACCTGGGCAAAGGTGTTCGTACTGCGGTCCAGAATGTTAATTCGTCAATCCACGAACTGCTG
GTTGGCCAAAGCGTTTTTGAACAGAAAGCGCTGGACGAGAAAATGATTGCATTTGATGGTACAGAAAATAAATCGAAGCTTGGTG
CGAATGCAACCCTCGCCGGTGAGCCTGGCGGCTGCGCATGCAGCGGCTGCTGAACAAAAATTGCCGCTGTTCCAGTATATCGCG
AACTTGCGTGGTCAAACGACCCTAACGATGCCGGTTCCGATGATGAACATTTTAAACGGTGGTGCCCACGCAGATAATACCGTG
GATATCCAAGAATTTATGATTGAGCCGGTAGGCTTTACCTCCTTCGCGGAGGCTCTGCGCGCGGGCGCGGAGGTGTTCCACAG
CCTTAAGTCTGTTCTGAAGAAACAAGGTCTGAACACCGCGGTGGGCGACGAGGGCGGTTTCGCGCCAAATCTCCGCAGCAATG
AGGAAGCGATTACCGTGATTCTTCAGGCAATCGAGCAGACCGGTTACAAGGCTGGTTCCGATATCATGCTGGCTTTGGATTGCG
CGAGCAGCGAATTCTATAAGAACGGCCAATACATCTTGGAAGGTGAGGGCAACAAAAGCTTCACCTCTAACCAGTTTGCCGACT
ATCTGGCAGGTCTGGTGAAACAGTACCCGATTATCAGCATCGAGGACGGCCTGGACGAGTCCGACTGGGAAGGCTGGAGCTAT
CTGACCTCTATCTTGGGCGACAAGATCCAACTGGTTGGCGATGATCTGTTTGTCACCAATCCGAAAATCCTGCAGCGTGGTATT
GATGAAAAGGTGGGCAACAGCATTCTGATTAAGTATAACCAAATTGGTACGCTGACCGAGACTCTGGACGCTATCTACCTGGCG
AAAGCGAACGGCTACACCACCGTGATCAGCCATCGTAGCGGTGAAACCGAAGATAGCACCATCGCAGACCTGGCCGTAGGAAC
GGCCGCCAGGCCAGATTAAGACCGGTTCCCTGTGTCGTAGCGATCGTGTTTCCAAATACAACCAGCTGCTGCGTATTGAGGAACT
GACGAAAGCGGTCTATCGTGGTAAGGCTGAGTTCAAGGGCTTGAACTAA

**C**

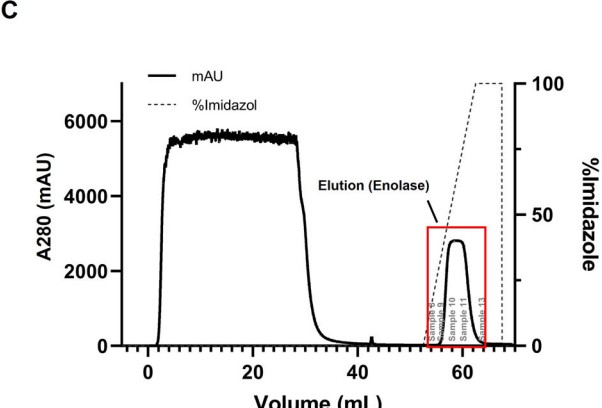

**D**

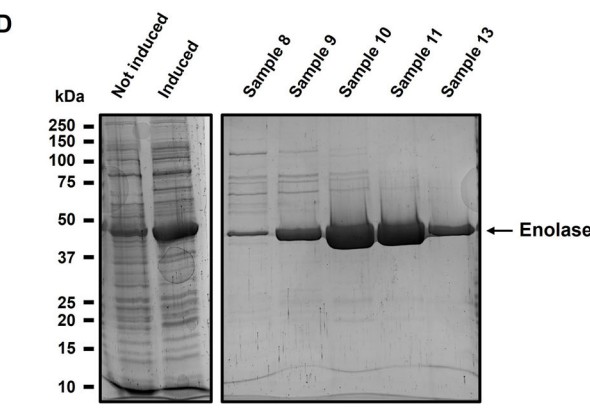

**Figure EV2.  Construction and characterization of *A.baumannii* enolase.**

(**A**) Illustration of the recombinant expression vector of enolase. (**B**) The sequence of the enolase. (**C**) Purification of enolase using the Histrap FF column. (**D**) SDS-PAGE gel analysis of the purified enolase.

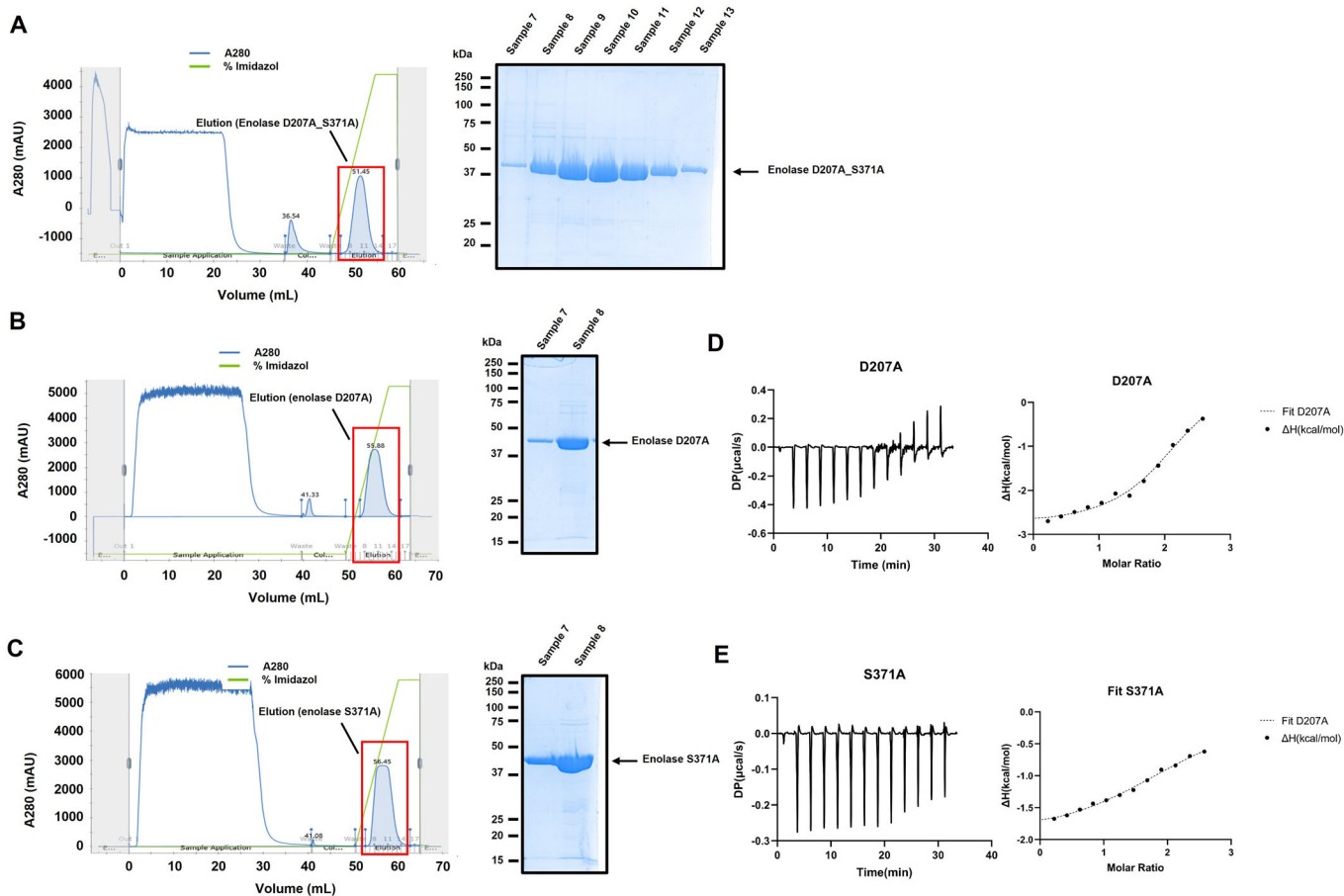

**Figure EV3.  Construction and characterization of *A. baumannii* enolase mutants.**

(A–C) Purification and SDS-PAGE gel analysis of the purified enolase D207_S371, enolase D207A and enolase S371 using the Histrap FF column. (D, E) Isothermal titration calorimetry (ITC) titrations with integrated fitted heat plots of ENOblock binding with enolase (D207A) or enolase (S371A).

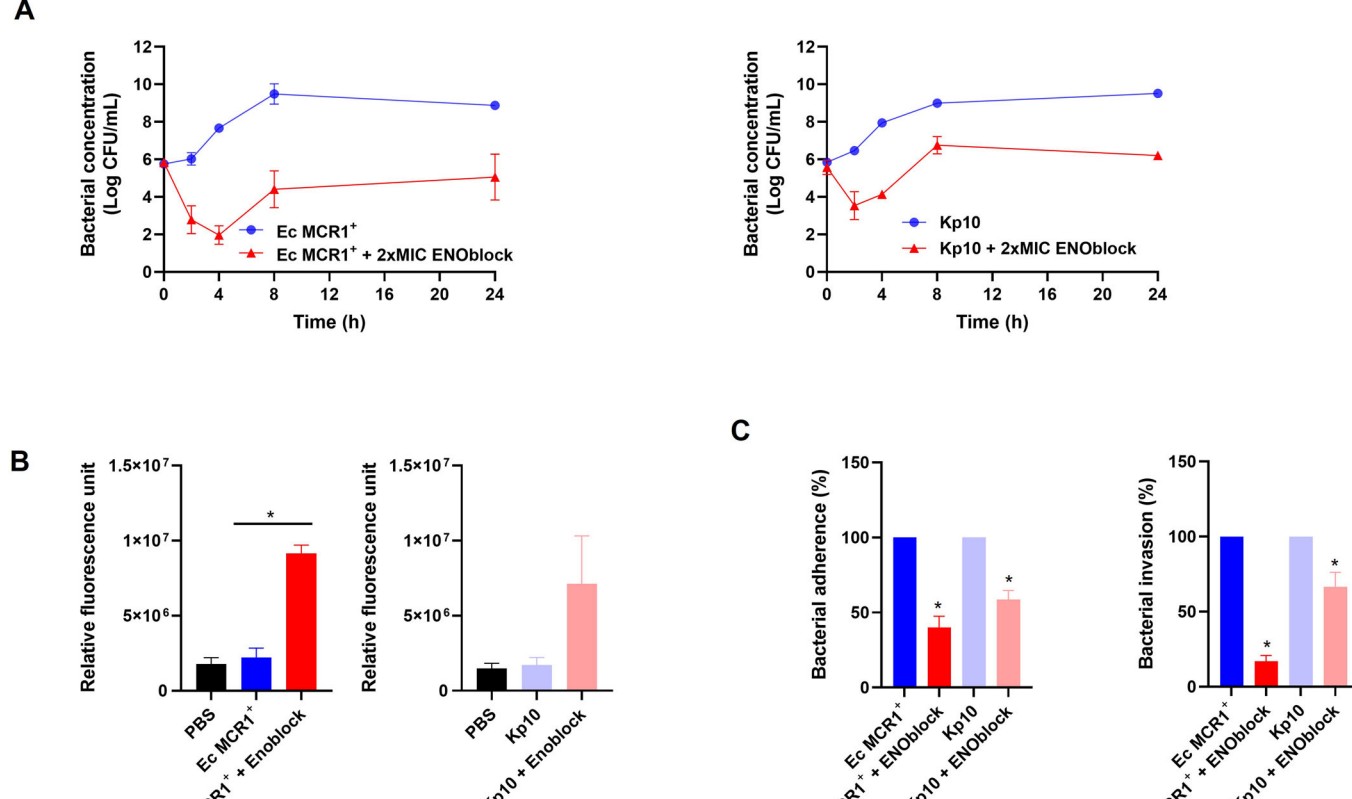

**Figure EV4. ENOblock is active against *E. coli* and *K. pneumoniae*.**

(**A**) Time-kill curves of *E. coli* Ec MCR1+ and Kp10 strains in the presence of 2xMIC ENOblock for 24 h. (**B**) Membrane permeabilization of *E. coli* Ec MCR1+ and Kp10 strains in the presence of 0.5xMIC ENOblock, incubated for 10 min, was quantified by Typhon Scanner. Data are represented as mean ± SEM from three independent replicates and experiments. *P = 0.001: Ec MCR1+ vs Ec MCR1+ + ENOblock(two-tailed Student's *t* test). (**C**) Analysis of *E. coli* Ec MCR1+ and Kp10 strains adhesion into HeLa cells with (1xMIC) and without ENOblock treatment. The data are presented as means ± SEM, *P = 0.015: Ec MCR1+ vs Ec MCR1+ + ENOblock and *P = 0.02: Kp10 vs Kp10 + ENOblock (two-tailed Student's *t* test). Analysis of *E. coli* Ec MCR1+ and Kp10 strains invasion into HeLa cells with (1xMIC) and without ENOblock treatment. The data are presented as means ± SEM, *P = 0.002: Ec MCR1+ vs Ec MCR1+ + ENOblock and *P = 0.037: Kp10 vs Kp10 + ENOblock treatment (two-tailed Student's *t* test).

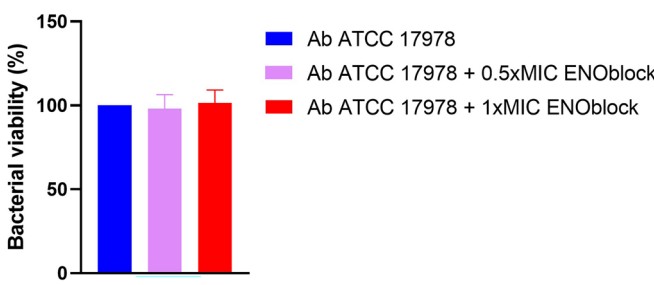

**Figure EV5.** Bacterial viability after incubation of ATCC with 0.5 and 1×MIC during 30 min with ENOblock.

