## [Peer Review File · EMBO Molecular Medicine]

ENOblock synergizes with colistin to treat *Acinetobacter baumannii* infections

Irene Molina Panadero, Antonio Moreno Rodríguez, Angela Rey Hidalgo, Mercedes de la Cruz, Pilar Sánchez, Laura Tomás Gallardo, Thanadon Samernate, Milan Sencanski, Sanja Glisic, Olga Genilloud, Poochit Nonejuie, Antonio J Perez-Pulido, Abdelkrim Hmadcha, and Younes SMANI

Corresponding author: Younes SMANI (ysma@upo.es)

Review Timeline:

Submission Date:	5th Jun 25
Editorial Decision:	20th Jun 25
Revision Received:	18th Sep 25
Editorial Decision:	7th Oct 25
Revision Received:	13th Oct 25
Accepted:	21st Oct 25

Editor: Zeljko Durdevic

Transaction Report:

20th Jun 2025

Dear Dr. Smani,

Thank you for the submission of your manuscript to EMBO Molecular Medicine. We have now received feedback from the three reviewers who agreed to evaluate your manuscript. As you will see from the reports, all three referees recognize potential interest of the study, but they also raise serious and partially overlapping concerns that should be addressed in a major revision. If you would like to discuss further the points raised by the referees, I am available to do so via email or video. Let me know if you are interested in this option.

We would welcome the submission of a revised version within three months for further consideration. Please let us know if you require longer to complete the revision.

I look forward to receiving your revised manuscript.

Yours sincerely,

Zeljko Durdevic

Zeljko Durdevic
Senior Editor
EMBO Molecular Medicine

We require:

- 1) A .docx formatted version of the manuscript text (including legends for main figures, EV figures and tables). Please make sure that the changes are highlighted to be clearly visible.
- 2) Individual production quality figure files as .eps, .tif, .jpg (one file per figure). For guidance, download the 'Figure Guide PDF': (<https://www.embopress.org/page/journal/17574684/authorguide#figureformat>).
- 3) A .docx formatted letter INCLUDING the reviewers' reports and your detailed point-by-point responses to their comments. As part of the EMBO Press transparent editorial process, the point-by-point response is part of the Review Process File (RPF), which will be published alongside your paper.
- 4) A complete author checklist, which you can download from our author guidelines (<https://www.embopress.org/page/journal/17574684/authorguide#submissionofrevisions>). Please insert information in the checklist that is also reflected in the manuscript. The completed author checklist will also be part of the RPF.
- 5) Please note that all corresponding authors are required to supply an ORCID ID for their name upon submission of a revised manuscript.
- 6) It is mandatory to include a 'Data Availability' section after the Materials and Methods. Before submitting your revision, primary

datasets produced in this study need to be deposited in an appropriate public database, and the accession numbers and database listed under 'Data Availability'. Please remember to provide a reviewer password if the datasets are not yet public (see <https://www.embopress.org/page/journal/17574684/authorguide#dataavailability>).

12) Author contributions: You will be asked to provide CRediT (Contributor Role Taxonomy) terms in the submission system. These replace a narrative author contribution section in the manuscript.

13) A Conflict of Interest statement should be provided in the main text.

14) Every published paper now includes a 'Synopsis' to further enhance discoverability. Synopses are displayed on the journal webpage and are freely accessible to all readers. They include a short stand first (maximum of 300 characters, including space) as well as 2-5 one-sentences bullet points that summarizes the paper. Please write the bullet points to summarize the key NEW findings. They should be designed to be complementary to the abstract - i.e. not repeat the same text. We encourage inclusion of key acronyms and quantitative information (maximum of 30 words / bullet point). Please use the passive voice. Please attach

these in a separate file or send them by email, we will incorporate them accordingly.

15) Include a Reagents and Tools Table as part of the Methods section, which can be downloaded from our author guidelines (<https://www.embopress.org/page/journal/17574684/authorguide#structuredmethods>)

***** Reviewer's comments *****

Referee #1 (Comments on Novelty/Model System for Author):

Technically seems quite sound although some assays needed additional controls. Novelty is there for targeting *A. baumannii* but targeting enolase is already a known mechanism of inhibiting bacterial growth (PMID: 31745118) and ENOblock is known to target enolase (and has been reported to inhibit the activity of *E. coli* Enolase PMID: 30714720). The medical impact has potential but as the authors highlight more work is needed. The system appears adequate.

Referee #1 (Remarks for Author):

Overall this is a very interesting study that identifies ENOblock from a screen of approved drugs as an inhibitor of *A. baumannii* growth. The authors do extensive characterisation to explore the potential mechanism, proteomics to identify that Enolase is upregulated in response to host cell contact. The authors indicate membrane permeabilization is the cause of growth inhibition and the Eno enzyme is the (as anticipated) target for ENOblock, but how these two observations are linked is not clear. They also present some compelling data to indicate a role in eukaryotic cell adhesion which is also subsequently impacted by deletion of eno or ENOblock. The title suggests a novel antibiotic class however what the criteria are for this claim are not clear particularly given targeting enolase is already a known mechanism of inhibiting bacterial growth (PMID: 31745118) and ENOblock is known to target enolase (and has been reported to inhibit the activity of *E. coli* Enolase PMID: 30714720). Throughout the authors use 1xMIC for their assays, however my interpretation of 1xMIC is that this is a sufficient concentration to completely inhibit bacterial growth, however in several assay the bacteria appear unaffected by this concentration (see comments below). The demonstrate efficacy in vivo also.

Please include line numbers in any revised manuscripts to assist with directing reviewer queries.

Specific Comments

Abstract

- "in an animal infection model"

Introduction

- Restructure for more clarity "Compounding the problem of AMR with reported high number of deaths associated with bacterial AMR and/or attributable to bacterial AMR is the immediate threat of a reduction in the discovery and development of new antibiotics."

- Restructure for more clarity "Especially for pathogens like *Acinetobacter baumannii* that is reported with continuous increase in AMR and are accounted for most reported associated diseases and deaths (ECDC and WHO, 2023)". This is misleading as *A. baumannii* is not responsible for the greatest number of deaths amongst MDR pathogens.

- Introduction needs to introduce enolase and its role in the prokaryotic and eukaryotic cells before moving straight to inhibitors.

Results

High-throughput screening for repurposing drugs as antimicrobial agents

- Screen needs to be explained in more detail here.

- "We identified 33 compounds (1.32% of the total compounds) with inhibitory activity of {greater than or equal to}70% against at least one or both MDR strains of *A. baumannii* and *E. coli*." Where is this data? What were the strains? What were the compounds? More clarity needed here.

- What is enoblock and what is it typically used to treat? What concentrations are physiologically relevant?

ENOblock is active against *A. baumannii*

- What was the rationale for selecting colistin resistant mutants? Were these clinical isolates? Was the resistant mechanism characterised or were they all different resistance mechanisms? These strains don't seem to appear in the supplementary material. Similar questions apply to CRABs.

- Consistency needed when describing units (mg/L or mg/l)

- In figure 2A why does ATCC17978 seem to recover very quickly from 1xMIC to such an extent that it has outgrown the untreated by 24 hours? This would suggest resistance can develop extremely quickly. More detail needed in legends as it is not clear what unit is being measured? Is this OD600? Recovered CFU? The methods would suggest CFU but how these values

were determined is not clear. Not clear how the MIC of ENOblock against 17978 is 8mg/L when in Figure S2 it appears that CFU recovery at this concentration is quite high.

- Data needs to be shown if available for toxicity data, and how does this compare to current treatment concentrations?
- Again the data is quite confusing, the authors state "The reference strain Ab ATCC 17978 present an ENOblock MIC of 8 mg/L." yet in Figure 2E it appears to grow unaffected at this concentration? Do the authors have a different definition of MIC other than the classical "lowest drug concentration that prevents visible bacterial growth after overnight incubation"
- ENOblock inhibits the growth of *A. baumannii* via distinct mechanism of action
- Not clear why 8mg/L (1 X MIC) is being used for the microscopy when the cells can grow unaffected in this concentration.
- Figure 3 legend needs to state how many cells were analysed per replicate.
- Legend or the results text does not outline what strain this is?
- Figure 3 legend needs to state what concentration of each antibiotic is used and how this relates to the MIC. What does M1_1 and M1_2 mean? Given that colistin is known to permeabilise the membrane then why is there no Sytox uptake?
- Why was Fosfomycin not tested based on PMID 31745118?
- Membrane blebs not clear from image shown and should be highlighted with arrows. Why does meropenem not induce membrane blebs as reported previously (PMID: 25884840)?
- Distinct mechanism of action would indicate it operates differently to other enolase inhibitors, to make this claim a control of another known antibacterial enolase inhibitor (2-aminothiazoles) would need to be used?
- ENOblock exhibits rapid permeabilization activity against *A. baumannii*
- "Although the fluorescence levels continued to increase after 30 minutes, this rise was not statistically significant (Figure 3D)" They don't rise they decrease according to Figure 3D.
- It would be useful is possible to include a positive control here to give a relative sense of levels of membrane permeabilization (colistin/polymyxin B for example)
- ENOblock acts on *A. baumannii* through the inhibition of enolase
- Ab ATCC17978 strain (Δ eno); How can the cell survive without eno when inhibiting its activity is lethal?
- There is still some level of growth inhibition in Figure 4 G, how is this occurring if the target enzyme is missing?
- Deletion of eno in *E. coli* does not appear to impact fitness, why?
- ENOblock affects the *A. baumannii*-host interaction
- The concentration used is 1xMIC so surely there is significant inhibition of bacterial growth leading to the observed effects, which as a result would be entirely anticipated.
- Did the concentration of ENOblock used in these assays have any impact on the epithelial cells or macrophages?
- Host cell interaction induces metabolic changes in *A. baumannii* and upregulates enolase expression
- This section is really nice, but perhaps for the narrative the results of reduced adherence with the enolase mutant could be included here as a validation.
- "Directly influences" may be overstating and could be softened as no evidence is presented that this is direct.
- What were the other 5 overexpressed proteins that were common to macrophages and epithelial cells? The full list of all DEPs should be included as supplemental.
- Enolase mediates the interaction of *A. baumannii* with host cells via binding to host proteins
- It needs to be made cleared here that the addition of exogenous enolase is to outcompete binding sites for the cell.
- "However, the incubation of Ab ATCC 17978 strain with 0.5x and 1xMIC during 2 hours did not reduce the bacterial concentration (data not shown)." Data needs to be shown.
- ENOblock presents therapeutic efficacy in vivo
- "To confirm the in vitro effect of ENOblock in monotherapy and in combination with colistin against *A. baumannii*, and to study this efficacy in a complete organism, we moved to an invertebrate model of infection by *A. baumannii*." But the combination with colistin is not tested here?
- Really nice data but what happens after 3 days? Why was this concentration selected which is significantly above the 1xMIC described below? A range was probably tested and if so it would be good to include this as it is not uncommon for the MIC in vitro to differ in vivo.
- An ENOblock only control us needed to demonstrate zero toxicity in the model and typically these assays have PBS control also.

Referee #2 (Comments on Novelty/Model System for Author):

The manuscript by Molina Panadero et al. presents a well-conducted and scientifically valuable study identifying ENOblock, a molecule previously recognized for anti-inflammatory, anti-fibrotic, anti-cancer, and anti-diabetic properties, as a potential antimicrobial agent through high-throughput screening. The work is primarily focused on *Acinetobacter baumannii*, a WHO-listed critical pathogen, and includes supporting data for *E. coli* and *K. pneumoniae*. The manuscript is of significant microbiological interest and could contribute meaningfully to the development of novel therapeutic strategies against multidrug-resistant

pathogens.

The main strengths of the study are the innovative screening approach and the thorough characterization of ENOblock's antimicrobial effects on *A. baumannii*. However, several points need clarification or further experimental support to strengthen the conclusions. In particular, the SYTOX Green membrane permeability assay requires better controls, the Informational Spectrum Method analysis needs additional explanation, and the data on resistance development appear inconsistent and warrant revision. Additional points involve improving methodological clarity and addressing minor inconsistencies in data presentation.

If these concerns are addressed, the manuscript could make a valuable contribution to the field.

Referee #2 (Remarks for Author):

Major Comments

1. Scope and emphasis on *A. baumannii*

The study should better highlight the importance of its findings specifically for *A. baumannii*, considering its WHO critical priority status. The results obtained for *E. coli* and *K. pneumoniae* could be moved to a dedicated section or presented as a caveat, to maintain focus.

2. SYTOX Green membrane permeability assay (Figure 3D)

It is unexpected that while the authors state SYTOX permeabilizes membranes, *A. baumannii* treated with colistin showed no green fluorescence. The authors should include additional positive controls (e.g., colistin at higher concentrations or combined with lysozyme). The paper by Hahala et al. (Frontiers in Microbiology, 2021) suggests appropriate controls. Also, it would be helpful to know whether bacterial survival was assessed after the assay, to correlate fluorescence with viability.

3. Informational Spectrum Method (ISM) analysis

While I am not fully familiar with ISM, the presence of a peak at F 0.270 in panel C (fibrinogen) appears comparable to that seen for plasminogen and fibronectin, although the authors only discuss the F 0.435 peak. It would strengthen the manuscript if they explained why the F 0.270 peak was not considered relevant.

4. Resistance development to ENOblock

There is a contradiction between lines 347-349 and 356-357. The manuscript should clarify whether ENOblock induces resistance under selective pressure. I recommend culturing bacteria at sub-MIC concentrations with viable counts over time to better assess resistance development.

5. ENOblock target specificity and in vivo relevance

It would be valuable to include a sequence alignment comparing bacterial enolase to its human counterpart in the supplementary materials. Additionally, the potential for in vivo application could be enhanced by proposing a murine acute pneumonia model to assess antimicrobial efficacy and toxicity.

6. Invasion assay duration

The rationale for limiting the invasion assay to 30 minutes should be clarified, as such assays are typically extended to at least 24 hours to capture meaningful intracellular persistence.

Minor Comments

7. Line 45: Define CC50 upon first mention.

8. Line 100: Replace "present" with "presents" or "presented" according to context.

9. Line 127: Move the definition of FIC here from line 544.

10. Lines 148-150: This section appears incomplete - please revise.

11. Lines 326-328: Content seems missing or unclear - revise.

12. Line 490: Add "respectively" at the end. Also, clarify the rationale for using *P. aeruginosa* and why the inoculum was 5×10^5 instead of 1×10^6 .

13. Line 1023: Remove highlighted text.

14. Introduction: It would be helpful to add a brief summary of previous studies on ENOblock's biological activities to better contextualize the current findings.

15. Use of *A. baumannii* Ab: Redundant. Define abbreviation once, then use either *A. baumannii* or Ab consistently.

16. Figure 4B: The legend appears incomplete - please revise.

17. Figure 7K: No statistical significance indicators are shown - clarify.

18. Controls in Membrane Permeability Assays: Ensure 1% DMSO (used in 100 μ M ENOblock samples) is added to controls for consistency.

19. Tables 1 and 2: Consider merging for clarity.

20. EUCAST 2013 Breakpoints: Provide rationale for using this version over more recent updates.

21. Binding Mode of ENOblock: Do the authors have any preliminary evidence or hypothesis regarding whether ENOblock binding to enolase is reversible, irreversible, or covalent?

Referee #3 (Remarks for Author):

This study identified the anticancer agent ENOblock as a novel enolase-targeting antibiotic class, which exhibits significant antibacterial activity against multidrug-resistant *Acinetobacter baumannii*. Additionally, the agent also displays significant synergistic effect with the last resort antibiotic colistin by targeting enolase. The new findings in this study repurposed ENOblock as alternative antibiotic candidate, and more importantly, revealed the potential of enolase as a promising bacterial target for new antimicrobials development. In overall, the manuscript is well executed and innovative. However, there are still some issues need improving prior to publication, as detailed below:

1, The section spanning "ENOblock exhibits rapid permeabilization activity" to "ENOblock acts... through inhibition of enolase" requires expanded explanation. Specifically, cite established literature confirming enolase localization to the outer membrane and its role in membrane regulation in Gram-negative bacteria. This will bridge the observed permeabilization phenotype to enolase targeting.

2, Control for metabolic confounders in enolase-knockout studies:

The Δ eno strain exhibits reduced growth/adherence and increased ENOblock MIC. However, enolase deficiency disrupts central metabolism, making it unclear whether phenotypes stem from direct enolase inhibition or indirect metabolic collapse. To isolate these effects:

Add a control cohort where Δ eno is supplemented with phosphoenolpyruvate (PEP), the metabolic product downstream of enolase.

Compare ENOblock susceptibility and adherence between PEP-supplemented vs. unsupplemented Δ eno. If phenotypes persist only without PEP, metabolic disruption likely drives the effects.

3, Incomplete in vivo validation:

Dose-response deficit: Efficacy was tested solely at 32 mg/L in *G. mellonella*. Include graded doses (e.g., 8, 16, 32 mg/L) to establish therapeutic range and dose dependency.

Untested synergy: The reported in vitro synergy with colistin lacks in vivo validation. Co-administration of ENOblock and colistin in the *G. mellonella* model is essential to support clinical translatability.

Referee #1 (Comments on Novelty/Model System for Author):

Technically seems quite sound although some assays needed additional controls. Novelty is there for targeting *A. baumannii* but targeting enolase is already a known mechanism of inhibiting bacterial growth (PMID: 31745118) and ENOblock is known to target enolase (and has been reported to inhibit the activity of *E. coli* Enolase PMID: 30714720). The medical impact has potential but as the authors highlight more work is needed. The system appears adequate.

Referee #1 (Remarks for Author):

Overall this is a very interesting study that identifies ENOblock from a screen of approved drugs as an inhibitor of *A. baumannii* growth. The authors do extensive characterisation to explore the potential mechanism, proteomics to identify that Enolase is upregulated in response to host cell contact. The authors indicate membrane permeabilization is the cause of growth inhibition and the Eno enzyme is the (as anticipated) target for ENOblock, but how these two observations are linked is not clear. They also present some compelling data to indicate a role in eukaryotic cell adhesion which is also subsequently impacted by deletion of *eno* or ENOblock. The title suggests a novel antibiotic class however what the criteria are for this claim are not clear particularly given targeting enolase is already a known mechanism of inhibiting bacterial growth (PMID: 31745118) and ENOblock is known to target enolase (and has been reported to inhibit the activity of *E. coli* Enolase PMID: 30714720). Throughout the authors use 1xMIC for their assays, however my interpretation of 1xMIC is that this is a sufficient concentration to completely inhibit bacterial growth, however in several assay the bacteria appear unaffected by this concentration (see comments below). The demonstrate efficacy *in vivo* also.

We are grateful to the Reviewer for the insightful comments on our paper. We appreciate the dedicated time and effort on our manuscript. We have been able to perform new experiments and to incorporate changes to reflect most of the suggestions provided by the Reviewer. We have highlighted with red colour all modifications within the manuscript. Here is a point-by-point response to the reviewer's comments and concerns

Please include line numbers in any revised manuscripts to assist with directing reviewer queries.

As requested by the Reviewer, we have included the line numbers in the revised manuscript.

Specific Comments

Abstract

- "in an animal infection model"

As requested by the Reviewer, we have corrected the mistake in the abstract section (see line 32).

Introduction

- Restructure for more clarity "Compounding the problem of AMR with reported high number of deaths associated with bacterial AMR and/or attributable to bacterial AMR is the immediate threat of a reduction in the discovery and development of new antibiotics."

As requested by the Reviewer we have restructured for more clarity "*Compounding the problem of AMR with reported high number of deaths associated with bacterial AMR and/or attributable to bacterial AMR is the immediate threat of a reduction in the discovery and development of new antibiotics.*" as follow:

"The challenge of AMR is further exacerbated by the alarming number of deaths attributed to bacterial AMR, coupled with the urgent threat posed by the declining discovery and development of new antibiotics" (see lines 49-51).

- Restructure for more clarity "Especially for pathogens like *Acinetobacter baumannii* that is reported with continuous increase in AMR and are accounted for most reported associated diseases and deaths (ECDC and WHO, 2023)". This is misleading as *A. baumannii* is not responsible for the greatest number of deaths amongst MDR pathogens.

As requested by the Reviewer we have restructured for more clarity "*Especially for pathogens like *Acinetobacter baumannii* that is reported with continuous increase in AMR and are accounted for most reported associated diseases and deaths (ECDC and WHO, 2023).*" as follow:

*"This is particularly critical for pathogens such as *Acinetobacter baumannii* (Ab), *Escherichia coli* and *Klebsiella pneumoniae*, among others, which have shown a continuous increase in antimicrobial resistance and is responsible for a significant proportion of reported infections and associated deaths (ECDC and WHO, 2023)"* (see lines 58-61).

- Introduction needs to introduction enolase and its role in the prokaryotic and eukaryotic cells before moving straight to inhibitors.

As requested by the Reviewer, we have provided the introduction of enolase and its role in the prokaryotic and eukaryotic cells before moving straight to inhibitors (see lines 68-78).

Results

High-throughput screening for repurposing drugs as antimicrobial agents

-Screen needs to be explained in more detail here.

As requested by the Reviewer, we have explained in more detail the HTS (See lines 100-102).

- "We identified 33 compounds (1.32% of the total compounds) with inhibitory activity of {greater than or equal to}70% against at least one or both MDR strains of *A. baumannii* and *E. coli*." Where is this data? What were the strains? What were the compounds? More clarity needed here.

As requested by the Reviewer, we have included the name of strains in the revised manuscript (see lines 103, 105, 106-107).

To preserve the possibility of future patent protection, we have not disclosed in the manuscript the names of the 33 compounds showing $\geq 70\%$ inhibitory activity against at least one or both MDR strains of *A. baumannii* (Ab186) and *E. coli* (MCR1⁺), nor the names of the seven compounds with MICs ≤ 100 μM against both reference and MDR strains of *A. baumannii* and *E. coli*, except for ENOblock. Below are the two lists of the respective 33 and 7 compounds, respectively:

Thirty-three compounds: *Bisindolylmaleimide IX*, *bithionol*, *cabotegravir*, *carfilzomib*, *chloroxine*, *chlorquinaldol*, *CUDC-907*, *decynium 22*, *demethoxycurcumin*, *diaveridine*, *diiodohydroxyquinoline*, *diphenyleneiodonium*, *dolutegravir*, *ellipticine*, *enoblock*, *hexachlorophene*, *h-89 dihydrochloride*, *gentian violet*, *hinokitiol*, *methacycline hydrochloride*, *3-methyltoxoflavin*, *ML069*, *ML311*, *NPS-2143*, *NSC228155*, *oxyquinoline*, *pyrimethamine*, *pyrithione*, *semapimod*, *terpyridine triapine*, *trimetrexate*, *sulfasuccinamide*.

Seven compounds: *Chlorexine*, *diphenyleneiodonium*, *ENOblock*, *hexachlorophene*, *3-methyltoxoflavin*, *semapimod* and *trimetrexate*.

- What is ENOblock and what is it typically used to treat? What concentrations are physiologically relevant?

ENOblock (also known as AP-III-a4) is a small-molecule inhibitor that targets enolase and is considered a non-active site inhibitor. It has been studied preclinically in various disease models, including cancer, diabetes, adipogenesis, and autoimmune and inflammatory diseases [Ni et al, 2023; Cho et al, 2017; Jung et al, 2013; Cho et al, 2019].

Most *in vitro* studies use ENOblock in the range of 1–100 μ M, with significant biological effects typically observed between 10–50 μ M. In cancer cell lines, concentrations of 10–25 μ M are commonly effective in reducing glycolytic activity and cell viability [Jung et al, 2013]. *In vivo* data are limited, but some animal studies have used doses of 8 mg/kg to achieve similar serum levels [Cho et al, 2017].

Bibliographic references

1. Cho H, Lee JH, Um J, Kim S, Kim Y, Kim WH, Kim YS, Pagire HS, Ahn JH, Ahn Y, Chang YT, Jung DW, Williams DR ENOblock inhibits the pathology of diet-induced obesity. *Sci. Rep.* 2019; 9(1):493.
2. Cho H, Um J, Lee JH, Kim WH, Kang WS, Kim SH, Ha HH, Kim YC, Ahn YK, Jung DW, Williams DR. ENOblock, a unique small molecule inhibitor of the non-glycolytic functions of enolase, alleviates the symptoms of type 2 diabetes. *Sci. Rep.* 2017; 7:44186.
3. Jung DW, Kim WH, Park SH, Lee J, Kim J, Su D, et al. A unique small molecule inhibitor of enolase clarifies its role in fundamental biological processes. *ACS Chem. Biol.* 2013; 8(6):1271–82.
4. Ni J, Huang Y, Li C, Yin Q, Ying J. Beyond ENO1, emerging roles and targeting strategies of other enolases in cancers. *Mol Ther Oncolytics* 2023; 31:100750.

ENOblock is active against *A. baumannii*

- What was the rationale for selecting colistin resistant mutants? Were these clinical isolates? Was the resistant mechanism characterised or were they all different resistance mechanisms? These strains don't seem to appear in the supplementary material. Similar questions apply to CRABs.

Thank you for your thoughtful questions. The colistin-resistant mutants included in our study were selected to represent a range of resistance mechanisms and genetic backgrounds.

We aimed to include both clinically relevant and experimentally derived colistin and carbapenem-resistant strains to evaluate the robustness of our findings across diverse resistance profiles.

The colistin and carbapenem-resistant mutants were indeed clinical isolates obtained from collaborating institutions. This information is included in the original manuscript version (see lines 482-485).

The mechanisms of resistance were characterized for the majority of the strains, including mutations in the *pmrAB* and presence of carbapenemases OXA-24, OXA-51 and OXA-58. However, a few strains without identified resistance mechanisms were included to reflect the variability observed in clinical settings. We included in the table 1 and 2 of the revised manuscript the characterized resistance mechanisms of these strains.

- Consistency needed when describing units (mg/L or mg/l)

Thank you for pointing this out. We have reviewed the manuscript thoroughly and standardized all units to use mg/L consistently throughout the text, figures, and tables.

- In figure 2A why does ATCC17978 seem to recover very quickly from 1xMIC to such an extent that it has outgrown the untreated by 24 hours? This would suggest resistance can develop extremely quickly. More detail needed in legends as it is not clear what unit is being measured? Is this OD600? Recovered CFU? The methods would suggest CFU but how these values were determined is not clear. Not clear how the MIC of ENOblock against 17978 is 8mg/L when in Figure S2 it appears that CFU recovery at this concentration is quite high.

Thank you for your careful analysis of Figure 2A. We appreciate the opportunity to clarify these points:

The apparent overgrowth of the treated ATCC17978 strain compared to the untreated control at 24 hours is likely due to bacterial regrowth following an initial bacteriostatic effect of the compound at 1×MIC. This phenomenon may reflect a tolerance-like behavior rather than true resistance development within that time frame. The MIC of ENOblock against ATCC 17978 at 24 hours of growth is 8 mg/L which suggest that not acquiring resistance is present (This information is discussed see lines 399-403). In addition, this hypothesis is validated by the results of selective pressure experiments where treatment against *A. baumannii* did not induce resistance to ENOblock (Figure S2A).

Regarding the measurement units and methodology, the Reviewer is correct. The measurements in Figure 2A reflect CFU counts. We have revised the figure legend to clearly state that CFUs were determined at each time point and that values are presented as mean from two biological replicates (see lines 1021-1022).

Clarification on MIC and CFU recovery in Figure S2: The MIC of ENOblock against ATCC17978 was determined by broth microdilution, following standard protocols. The apparent CFU recovery at 8 mg/L of ENOblock in Figure S2 reflects survival of a subpopulation rather than complete inhibition, which is not uncommon near the MIC threshold. We have clarified this discrepancy in the discussion section, emphasizing that MIC determination is based on visible growth inhibition in broth culture, while CFU assays may detect surviving subpopulations even at MIC levels (see lines 399-403).

- Data needs to be shown if available for toxicity data, and how does this compare to current treatment concentrations?

Thank you for your comment. We agree that toxicity data are essential for assessing the potential clinical applicability of ENOblock. In the revised manuscript, we have included available *in vitro* cytotoxicity data (see Supplementary Figure S1), where ENOblock was tested on HeLa cells and THP-1 cells differentiated into macrophages. The compound exhibited CC_{50} values of 112.7 mg/L in HeLa cells and 116.2 mg/L in macrophages, which are at least three times higher than the MIC_{90} of ENOblock against all tested isolates in our antimicrobial assays.

We have also added a comparison with the cytotoxicity profiles of currently used antibiotics (e.g., colistin and tigecycline), showing that the CC_{50} values of ENOblock fall above the range reported for both antibiotics, supporting its potential safety profile. This discussion has been incorporated into the revised manuscript (see lines 364-374) as follows:

“For instance, colistin has been reported to exhibit CC_{50} values between 50 and 100 mg/L in HK-2 cells (human kidney proximal tubule cells) [Lee et al, 2015]. Moreover, colistin at 8 mg/L and polymyxin B at 40 mg/L have been shown to reduce viability by 50% in both HeLa cells and PMA-differentiated THP-1 cells [Luo et al, 2013; Kagi et al, 2013]. Regarding tigecycline, CC_{50} values of 44 and 51 mg/L have been reported in NCM-356 cells (an epithelial cell line derived from normal colon mucosa) [Ruiz Malagón et al, 2023], and in PC12 cells (cell line derived from a pheochromocytoma of the rat adrenal medulla) [Huang et al, 2022], respectively. Furthermore, prolonged treatment with tigecycline (3 days) decreased its CC_{50} to 1.7 mg/L in Jurkat T cells and to a range of 1.2–5.5 mg/L in peripheral blood mononuclear cells (PBMCs) [Shao et al, 2025].”

References

1. Huang Q, Zhang X, Jia A, Huang Q, Jiang Y, Xie L. The Pharmacokinetics/Pharmacodynamics and Neurotoxicity of Tigecycline Intraventricular Injection for the Treatment of Extensively Drug-Resistant

- Acinetobacter baumannii* Intracranial Infection. *Infect. Drug Resist.* 2022; 15:4809-4817.
2. Kagi T, Naganuma R, Inoue A, Noguchi T, Hamano S, Sekiguchi Y, Hwang GW, Hirata Y, Matsuzawa A. The polypeptide antibiotic polymyxin B acts as a pro-inflammatory irritant by preferentially targeting macrophages. *J. Antibiot.* 2022; 75(1):29-39.
 3. Lee SH, Kim JS, Ravichandran K, Gil HW, Song HY, Hong SY. P-Glycoprotein Induction Ameliorates Colistin Induced Nephrotoxicity in Cultured Human Proximal Tubular Cells. *PLoS One.* 2015; 10(8):e0136075.
 4. Luo Y, Wang C, Peng P, Hossain M, Jiang T, Fu W, Liao Y, Su M. Visible light mediated killing of multidrug-resistant bacteria using photoacids. *J. Mater. Chem. B.* 2013; 1(7):997-1001.
 5. Ruiz-Malagón AJ, Hidalgo-García L, Rodríguez-Sojo MJ, Molina-Tijeras JA, García F, Diez-Echave P, Vezza T, Becerra P, Marchal JA, Redondo-Cerezo E, Hausmann M, Rogler G, Garrido-Mesa J, Rodríguez-Cabezas ME, Rodríguez-Nogales A, Gálvez J. Tigecycline reduces tumorigenesis in colorectal cancer via inhibition of cell proliferation and modulation of immune response. *Biomed. Pharmacother.* 2023; 163:114760.
 6. Shao Q, Khawaja A, Nguyen MD, Singh V, Zhang J, Liu Y, Nordin J, Adori M, Axel Innis C, Castro Dopico X, Rorbach J. T cell toxicity induced by tigecycline binding to the mitochondrial ribosome. *Nat. Commun.* 2025;16(1):4080.

- Again the data is quite confusing, the authors state "The reference strain Ab ATCC 17978 present an ENOblock MIC of 8 mg/L." yet in Figure 2E it appears to grow unaffected at this concentration? Do the authors have a different definition of MIC other than the classical "lowest drug concentration that prevents visible bacterial growth after overnight incubation"

Thank you for your comment. We apologize for the confusion regarding the interpretation of the MIC value and the data shown in Figure 2E.

As correctly stated, the MIC of ENOblock for *A. baumannii* ATCC 17978 strain was determined as 8 mg/L using the standard broth microdilution method, following EUCAST guidelines, where MIC is defined as the lowest concentration that prevents visible bacterial growth after overnight incubation (typically 18–20 hours). The apparent growth observed at 8 mg/L in Figure 2E reflects a bacterial growth experiment, which differs in both design and readout from the MIC determination. While MIC assays assess end-point growth inhibition in static conditions, bacterial growth curves provide dynamic information over 24 hours and can reveal regrowth of bacterial subpopulations even at concentrations near or above the MIC, particularly in the case of bacteriostatic agents or compounds with delayed effects.

To clarify this difference, we have revised the figure legend and the main text to clearly state that Figure 2E presents density absorbance over time and that delayed growth at MIC concentrations does not contradict the classical MIC definition, but rather highlights the complexity of the compound's activity and possible tolerance or persistence phenomena (see lines 614-617, 1027 and 1029-1030).

ENOblock inhibits the growth of *A. baumannii* via distinct mechanism of action

- Not clear why 8mg/L (1 X MIC) is being used for the microscopy when the cells can grow unaffected in this concentration.

We thank the Reviewer for pointing this out. The concentration used for microscopy in the original manuscript was 32 mg/L. This higher concentration was selected because treatment at 8 mg/L did not result in detectable morphological changes, and thus higher concentration of the compound was necessary to visualize the cellular effects of ENOblock. This observation aligns with the reviewer's earlier comment that ENOblock at its MIC (8 mg/L) does not significantly affect bacterial growth.

To better reflect the morphological effects of ENOblock at a concentration more comparable to 1xMIC of standard antibiotics, we have revised the figure using 16 mg/L (2xMIC) instead of 32 mg/L (4xMIC) for the cytological analysis. Notably, cells treated with 16 mg/L ENOblock displayed morphological changes identical to those observed at 32 mg/L, supporting the robustness of the phenotype.

All related information and the corresponding figure legend have been revised accordingly (see lines 162-163, 173, 182-187, 202-205, 1035-1055, and new figure 3).

- Figure 3 legend needs to state how many cells were analysed per replicate.

We agreed with the Reviewer and more detail regarding number of cells analyzed was added into the revised legend (see lines 1035-1055).

- Legend or the results text does not outline what strain this is?

We agreed with the Reviewer and more detail regarding specific cell strain analyzed was added into the revised legend (see lines 1035-1055).

- Figure 3 legend needs to state what concentration of each antibiotic is used and how this relates to the MIC. What does M1_1 and M1_2 mean? Given that colistin is known to permeabilise the membrane then why is there no Sytox uptake?

We agreed with the Reviewer and more detail regarding antibiotic concentration and its MIC relationship was added into the revised legend (see lines 1035-1055).

Regarding “M1_1” and “M1_2”: The M1_1 and M1_2 groups refer to subclusters identified in the BCP (bacterial cytological profiling) data for colistin-treated *A. baumannii* cells. In our previous work, these two subclusters were observed: M1_1 exhibited distinct morphological changes consistent with colistin action, while M1_2 appeared more similar to untreated cells. This suggests intrapopulation variation in antibiotic response. To avoid confusion regarding subcluster labeling in both the colistin and meropenem datasets, we have removed the subcluster designations from the revised figure (see new figure 3).

Regarding SYTOX uptake with colistin: Colistin’s primary mechanism of action involves binding to lipid A, leading to outer membrane permeabilization and typically allowing the influx of dyes such as SYTOX Green. However, in our single-cell BCP experiments on *A. baumannii*, all antibiotics, including colistin, were tested at 1xMIC (0.625 mg/L), specifically to capture early or subtle morphological changes while minimizing complete cell lysis, which would interfere with analysis.

At this concentration, we did not observe a clear increase in SYTOX Green or DAPI uptake. Nevertheless, colistin-treated cells frequently displayed a doublet-like morphology, likely reflecting alterations in the outer membrane. Although this membrane change was insufficient to allow entry of large DNA-binding dyes, it still appeared to affect envelope-associated processes, resulting in the observed morphological phenotype without full membrane permeability.

- Why was Fosfomycin not tested based on PMID 31745118?

We thank the Reviewer for this insightful comment. PMID 31745118 demonstrated that SF2312, when combined with fosfomycin in the presence of glucose-6-phosphate, shows significant synergy, suggesting these two agents act via distinct mechanisms of action. This indicates that fosfomycin, a cell wall-targeting antibiotic, functions differently from enolase inhibitors such as SF2312.

In our current study, we focused on comparing the BCP signature of ENOblock-treated *A. baumannii* cells with previously published profiles of standard antibiotics. Our aim was to highlight the uniqueness of the ENOblock-induced profile relative to these existing reference datasets. As fosfomycin had not been profiled in our existing *A. baumannii* BCP reference set, it was not included in the current analysis.

Inspired by the Reviewer's suggestion, we recognize the value of adding fosfomycin to our BCP reference panel for *A. baumannii*. We will prioritize generating and incorporating the fosfomycin-treated profile in future work to further strengthen and expand the utility of our BCP database.

- Membrane blebs not clear from image shown and should be highlighted with arrows. Why does meropenem not induce membrane blebs as reported previously (PMID: 25884840)?

We thank the Reviewer for this helpful observation. Upon careful review, we acknowledge that the images did not show true "membrane blebs" as classically defined (i.e., spherical protrusions or bulges of the plasma membrane extending outward from the cell envelope). Instead, what we observed in ENOblock-treated cells was uneven accumulation or pooling of FM4-64 dye along the membrane, suggesting localized membrane perturbation rather than distinct bleb formation.

To better reflect our actual findings and avoid misinterpretation, we have revised the text in the manuscript. The original sentence: "*In particular, membrane blebs were observed, as was the high SYTOX green signal of ENOblock-treated cells (Figure 3A), indicating the loss of membrane integrity.*" has been changed to: "*In particular, high SYTOX Green signal of ENOblock-treated cells was observed (Figure 3A), indicating the loss of membrane integrity.*" (see lines 162-163).

We believe this revision provides a clearer and more accurate description of the observed phenotype.

Regarding the second comment in relation with meropenem, we thank the Reviewer for this important question. In PMID: 25884840, the authors employed higher concentrations of β -lactam antibiotics and examined cells at later time points. Under those conditions, membrane blebs were observed as a consequence of severe peptidoglycan disruption and subsequent cell lysis. In contrast, our study used antibiotics at 1xMIC and analyzed cells at an earlier time point (1 hour). This approach was specifically designed to capture the primary, early-stage effects of PBP inhibition before extensive cell wall damage and lysis occur. For example, inhibition of PBP3 by piperacillin in our assay resulted in clear cell elongation without septation defects, an expected early morphological change. The differences in antibiotic concentration and exposure duration likely explain the absence of membrane blebs in our β -lactam-treated controls, including meropenem, in this study.

- Distinct mechanism of action would indicate it operates differently to other enolase inhibitors, to make this claim a control of another known antibacterial enolase inhibitor (2-aminothiazoles) would need to be used?

We thank the Reviewer for this insightful comment. In the BCP section, we aimed to investigate whether ENOblock produces morphological changes resembling those caused by conventional antibiotics targeting major bacterial pathways, given that enolase inhibitors have not previously been profiled morphologically. Our intention was not to claim that ENOblock acts through a mechanism distinct from other known antibacterial enolase inhibitors (e.g., 2-aminothiazoles), but rather to compare its effects with established antibiotic classes.

In agreement with the Reviewer's suggestion and to more accurately reflect the scope of our interpretation, we have revised both the main text and the subsection header to clarify that the observed morphological differences highlight ENOblock's distinction from the conventional antibiotic controls used in this study (see lines 152-153).

ENOblock exhibits rapid permeabilization activity against *A. baumannii*

- "Although the fluorescence levels continued to increase after 30 minutes, this rise was not statistically significant (Figure 3D)" They don't rise they decrease according to Figure 3D.

We thank the Reviewer for this careful observation. We agree that in Figure 3D, the SYTOX Green intensity in ENOblock-treated cells peaks at approximately 10 minutes and does not show a significant increase or decrease at later time points. This behavior is consistent with prior observations (e.g., with mansonone G) [Htoo et al, 2022] where, following rapid permeabilization and maximum dye uptake, subsequent time points show minimal or statistically insignificant fluctuations in fluorescence intensity.

In this study, our primary goal was to demonstrate that ENOblock induces rapid membrane permeabilization within 10 minutes, as evidenced by the significant SYTOX Green increase compared to untreated controls. We did not intend to emphasize subtle temporal changes beyond this initial permeabilization event.

Reference

1. Htoo HH, Tuyet NNT, Nakprasit K, Aonbangkhen C, Chaikeratisak V, Chavasiri W, Nonejuie P. Mansonone G and its derivatives exhibit membrane permeabilizing activities against bacteria. PLoS One. 2022; 17(9):e0273614.

- It would be useful is possible to include a positive control here to give a relative sense of levels of membrane permeabilization (colistin/polymyxin B for example)

We agree with the Reviewer that including a positive control, such as colistin at a high concentration, would provide a useful reference point for assessing the relative extent of membrane permeabilization.

To address this, we performed additional experiments using colistin at 5xMIC as a positive control, alongside ENOblock at 8, 16, and 32 mg/L. This allowed us to better compare the membrane-permeabilizing effects of ENOblock across concentrations relative to a well-established permeabilizing agent.

All related data and the corresponding figure legend have been revised accordingly in the manuscript (see lines 182-187, figure 3C and 3D).

ENOblock acts on *A. baumannii* through the inhibition of enolase

Thank you for these insightful questions regarding the Δeno mutant and the role of enolase in *A. baumannii* and *E. coli*.

- Ab ATCC17978 strain (Δeno); How can the cell survive without eno when inhibiting its activity is lethal?

The apparent contradiction between gene deletion viability and inhibitor lethality might be attributed to compensatory metabolic adaptations that occur in knockout strains over time, allowing survival despite the loss of a key glycolytic enzyme. In contrast, acute pharmacological inhibition in wild-type cells does not allow for such adaptations, often resulting in lethality.

- There is still some level of growth inhibition in Figure 4 G, how is this occurring if the target enzyme is missing?

The slight growth inhibition observed in the Δeno strain upon ENOblock (32 mg/L) treatment may suggest off-target effects of ENOblock beyond enolase inhibition. While enolase is the presumed primary target, small molecules can affect multiple pathways, especially under stress conditions. Alternatively, a metabolic imbalance due to the gene knockout could sensitize cells to additional stress imposed by the compound. This is supported by the comparison of the growth curves of the wt strain and the Δeno strain, where the mutant strain exhibits delayed and reduced growth (see figures 4F,G).

- Deletion of *eno* in *E. coli* does not appear to impact fitness, why?

This observation likely reflects species-specific metabolic flexibility; even though the growth curve of Δ *eno* strain is inferior during 24 h. Of note, *E. coli* can efficiently switch to alternative pathway (e.g., the Entner–Doudoroff pathway) to bypass parts of glycolysis. Moreover, enolase might not be essential under the tested growth conditions, particularly in nutrient-rich media that provide precursors for bacterial cell component biosynthesis.

ENOblock affects the *A. baumannii*-host interaction

- The concentration used is 1xMIC so surely there is significant inhibition of bacterial growth leading to the observed effects, which as a result would be entirely anticipated.

Thank you for your comment. While it is true that 1xMIC typically results in significant inhibition of bacterial growth, the effects observed in our study suggest that ENOblock also impacts the *A. baumannii*-host interaction. A two-hour treatment with ENOblock at 1xMIC reduced the invasion of the ATCC 17978 strain into epithelial and macrophage cells by 76% and 67%, respectively (Figures 5B and 5C), which exceeds the 45% reduction in bacterial growth ($-2 \log$ CFU/mL) observed in the time-kill curve after two hours of ENOblock exposure (Figure 2A).

- Did the concentration of ENOblock used in these assays have any impact on the epithelial cells or macrophages?

Thank you for your comment and for raising this important point. We have assessed the potential cytotoxicity of ENOblock on epithelial cells at the concentrations used in the adherence and invasion assays. The compound did not exhibit significant toxicity at 1xMIC, as confirmed by cell viability assay. Cell viability remained above 94% for epithelial cell line after 2 hours of treatment (see figure 1), indicating that the observed reduction in bacterial adherence and invasion was not due to host cell damage but rather to a specific effect of ENOblock on the bacterial infection process.

Figure 1. Viability of epithelial cells in presence of ENOblock. HeLa cells were pretreated with 1xMIC ENOblock during 2 hours. The cell viability was determined used the MTT assay.

Host cell interaction induces metabolic changes in *A. baumannii* and upregulates enolase expression

- This section is really nice, but perhaps for the narrative the results of reduced adherence with the enolase mutant could be included here as a validation.

Thank you for your positive feedback and helpful suggestion. We agree that including the reduced adherence results from the enolase mutant would strengthen the narrative and provide additional validation of enolase's role in host–pathogen interaction. In the revised manuscript, we have integrated these findings into the section, highlighting that the Δeno mutant also showed significantly reduced adherence to host cells, supporting the impact of enolase on bacterial adhesion and invasion. This addition helps reinforce the proposed role of enolase in mediating *A. baumannii*–host interactions (see lines 286-291 and 1098-1102).

- "Directly influences" may be overstating and could be softened as no evidence is presented that this is direct.

Thank you for your comment. We agree that the phrase "*directly influences*" may be too strong given the current data, and we have revised the text accordingly to reflect a more cautious interpretation. The sentence now reads: "*might influence*" (see line 292-293).

- What were the other 5 overexpressed proteins that were common to macrophages and epithelial cells? The full list of all DEPs should be included as supplemental.

In addition to enolase, the other five overexpressed proteins common to both macrophage- and epithelial cell-exposed *A. baumannii* were: *tyrosine-tRNA ligase*, *malonyl CoA-acyl carrier protein transacylase*, *N-succinylarginine dihydrolase*, *carbonic anhydrase*, and *biotin carboxylase*. Their shared upregulation suggests a conserved bacterial adaptation to host cell contact. We have included this information in the revised manuscript for clarity (see lines 1086-1091).

As requested by the Reviewer, we have now included the complete list of DEPs identified in both epithelial and macrophage interaction conditions as Supplementary Table S5 and Table S6. Both tables include fold changes, accession number and description where available, to facilitate further interpretation and reproducibility.

Enolase mediates the interaction of *A. baumannii* with host cells via binding to host proteins

- It needs to be made cleared here that the addition of exogenous enolase is to outcompete binding sites for the cell.

Thank you for your comment and for highlighting the need for clarification.

We have revised the text to make it explicitly clear that the addition of exogenous enolase in our experiments was intended to competitively inhibit the interaction between bacterial surface-associated enolase and the ECM proteins (see line 325).

- "However, the incubation of Ab ATCC 17978 strain with 0.5x and 1xMIC during 2 hours did not reduce the bacterial concentration (data not shown)." Data needs to be shown.

First, we would like to specify that the incubation of Ab ATCC 17978 strain with 0.5x and 1xMIC ENOblock is during 30 minutes rather 120 minutes, that after is washed twice in sterile PBS in order to remove to non-binded ENOblock. We clarified this in Materials and methods section (see lines 916 and 919-921).

In response to your comment, we have now added the previously omitted data showing that incubation of *A. baumannii* ATCC 17978 with 0.5x and 1xMIC of ENOblock for 30 minutes did not significantly reduce bacterial concentration. These results are now presented in Supplementary Figure S8.

ENOblock presents therapeutic efficacy in vivo

- "To confirm the *in vitro* effect of ENOblock in monotherapy and in combination with colistin against *A. baumannii*, and to study this efficacy in a complete organism, we moved to an invertebrate model of infection by *A. baumannii*." But the combination with colistin is not tested here?

Thank you for your observation. That is correct. The *in vivo* experiment using the *Galleria mellonella* model was conducted with ENOblock monotherapy only, and not in combination with colistin. To address this, we performed an additional experiment using combination therapy: ENOblock at 16 and 32 mg/L was administered together with colistin at 0.12 mg/L (one of the colistin concentrations that showed synergy with ≥ 4 mg/L ENOblock) one hour after *G. mellonella* infection with *A. baumannii*. We observed that the combination of ENOblock (16 and 32 mg/L) with colistin significantly increased larval survival to 75% and 100%, respectively, compared to colistin monotherapy at 0.12 mg/L (25% survival), and ENOblock monotherapy at 16 and 32 mg/L (28.57% and 57.14% survival, respectively). We have revised the text accordingly (see lines 338-347 and 932-941 and new Figures 8D and 8E).

-Really nice data but what happens after 3 days? Why was this concentration selected which is significantly above the 1xMIC described below? A range was probably tested and if so it would be good to include this as it is not uncommon for the MIC in vitro to differ in vivo.

The three-day time point was chosen as a standard duration for assessing acute infection outcomes in the animal models as we previously described in different works [PMID: 38786115, PMID: 35234505, PMID: 34991848, PMID: 33810067, PMID: 33671416, PMID: 32240294, PMID: 30188994, PMID: 29867823, PMID: 29089624, PMID: 28128094, PMID: 27161639, PMID: 25896698]. While we acknowledge the value of extended observation, our primary aim was to evaluate early survival trends and therapeutic efficacy.

Regarding the ENOblock concentration: you are right that the dose used (32 mg/L equivalent of 4xMIC) is higher than the 1xMIC. This concentration was selected based on the *in vitro* bacterial growth assay (Figure 4E), which showed higher efficacy than 1xMIC. To respond to your comment, an a 1x and 2xMIC (8 and 16 mg/L) of ENOblock have been also assessed (see response to your previous comment).

- An ENOblock only control is needed to demonstrate zero toxicity in the model and typically these assays have PBS control also.

Thank you for your comment. In our *in vivo* assays using the *G. mellonella* model, we included a PBS control group (see line 937).

We agree with your suggestion that an ENOblock-only control group need to be performed. No toxicity was observed in larvae treated with ENOblock (8, 16 and 32 mg/L) alone, indicating that the compound is not toxic at the concentrations used. This information is included in the revised manuscript to clarify this point (see lines 346-347 and 939).

Referee #2 (Comments on Novelty/Model System for Author):

The manuscript by Molina Panadero et al. presents a well-conducted and scientifically valuable study identifying ENOblock, a molecule previously recognized for anti-inflammatory, anti-fibrotic, anti-cancer, and anti-diabetic properties, as a potential antimicrobial agent through high-throughput screening. The work is primarily focused on *Acinetobacter baumannii*, a WHO-listed critical pathogen, and includes supporting data for *E. coli* and *K. pneumoniae*. The manuscript is of significant microbiological interest and could contribute meaningfully to the development of novel therapeutic strategies against multidrug-resistant pathogens.

The main strengths of the study are the innovative screening approach and the thorough characterization of ENOblock's antimicrobial effects on *A. baumannii*. However, several points need clarification or further experimental support to strengthen the conclusions. In particular, the SYTOX Green membrane permeability assay requires better controls, the Informational Spectrum Method analysis needs additional explanation, and the data on resistance development appear inconsistent and warrant revision. Additional points involve improving methodological clarity and addressing minor inconsistencies in data presentation. If these concerns are addressed, the manuscript could make a valuable contribution to the field.

We are grateful to the Reviewer for the insightful comments on our paper. We appreciate the dedicated time and effort on our manuscript. We have been able to perform new experiments and to incorporate changes to reflect most of the suggestions provided by the Reviewer. We have highlighted with red colour all modifications within the manuscript. Here is a point-by-point response to the reviewer's comments and concerns.

Referee #2 (Remarks for Author):

Major Comments

1. Scope and emphasis on *A. baumannii*

The study should better highlight the importance of its findings specifically for *A. baumannii*, considering its WHO critical priority status. The results obtained for *E. coli* and *K. pneumoniae* could be moved to a dedicated section or presented as a caveat, to maintain focus.

Thank you for this valuable suggestion. We agree that emphasizing the relevance of our findings in the context of *A. baumannii*, a WHO critical priority pathogen, is essential. In the revised manuscript, we have restructured some section that

includes *E. coli* and *K. pneumoniae* results by framing them more clearly as comparative or contextual findings, to keep the main focus on *A. baumannii*. We like to highlight that all results obtained for *E. coli* and *K. pneumoniae* are presented as supplemental data (see lines 225-227 and 244-245).

2. SYTOX Green membrane permeability assay (Figure 3D)

It is unexpected that while the authors state SYTOX permeabilizes membranes, *A. baumannii* treated with colistin showed no green fluorescence. The authors should include additional positive controls (e.g., colistin at higher concentrations or combined with lysozyme). The paper by Hahala et al. (Frontiers in Microbiology, 2021) suggests appropriate controls. Also, it would be helpful to know whether bacterial survival was assessed after the assay, to correlate fluorescence with viability.

We agree with the Reviewer that including a positive control, such as colistin at a high concentration, would provide a useful reference point for assessing the relative extent of membrane permeabilization.

To address this, we performed additional experiments using colistin at 5xMIC as a positive control, alongside ENOblock at 8, 16, and 32 mg/L. This allowed for a clearer comparison of the membrane-permeabilizing effects of ENOblock at various concentrations relative to a well-established permeabilizing agent.

Furthermore, all additional SYTOX Green intensity data are now accompanied by corresponding CFU counts at each time point. As suggested by the Reviewer, a correlation between compound concentration, SYTOX Green intensity, and remaining viable bacterial counts was observed, which provides a more comprehensive view of membrane damage and its impact on cell viability.

All related data and figure legends have been revised accordingly in the manuscript (see lines 176-205 and Figure 3).

3. Informational Spectrum Method (ISM) analysis

While I am not fully familiar with ISM, the presence of a peak at F 0.270 in panel C (fibrinogen) appears comparable to that seen for plasminogen and fibronectin, although the authors only discuss the F 0.435 peak. It would strengthen the manuscript if they explained why the F 0.270 peak was not considered relevant.

Thank you for this insightful observation. You are right that a peak at F 0.270 is also present in the fibrinogen interaction profile shown in panel C.

The signal-to-noise (S/N) ratio between the enolase–ENOblock system and the fibrinogen–enolase–ENOblock system shows a more pronounced change at F(0.435) than at F(0.270). The respective ratios are 2.58 and 1.45. Therefore, F(0.435) was selected as the more specific indicator for blocking protein–protein interactions. This explanation is included in the revised manuscript (see lines 310-313 and table S7).

4. Resistance development to ENOblock

There is a contradiction between lines 347-349 and 356-357. The manuscript should clarify whether ENOblock induces resistance under selective pressure. I recommend culturing bacteria at sub-MIC concentrations with viable counts over time to better assess resistance development.

Thank you for your valuable comment. You are right in pointing out the need for clarity. To your knowledge, we have already cultured *A. baumannii* at sub-MIC concentration with viable counts over time to better assess resistance development (see figure S2, and lines 549-555). We have revised the manuscript to avoid confusion (see lines 415-417)

5. ENOblock target specificity and in vivo relevance

It would be valuable to include a sequence alignment comparing bacterial enolase to its human counterpart in the supplementary materials. Additionally, the potential for in vivo application could be enhanced by proposing a murine acute pneumonia model to assess antimicrobial efficacy and toxicity.

Thank you for your suggestion. We agree that comparing the bacterial enolase sequence with its human counterpart is crucial for evaluating the selectivity and safety of ENOblock. Accordingly, we have included a sequence alignment between *A. baumannii*, human and *Galleria mellonella* enolases in the supplementary materials (see figure S9).

Regarding your second comment, we fully recognize the importance of validating the therapeutic efficacy and safety of ENOblock in a mammalian model. While our current study focuses on the *G. mellonella* model, future work will aim to establish an acute murine pneumonia model to further assess both antimicrobial efficacy and potential host toxicity *in vivo* (see line 468).

6. Invasion assay duration

The rationale for limiting the invasion assay to 30 minutes should be clarified, as such assays are typically extended to at least 24 hours to capture meaningful intracellular persistence.

We thank the Reviewer for this comment. We apologize for the lack of clarity in our initial description. The invasion assay was not limited to 30 minutes; rather, the bacteria were allowed to invade for 2 hours, which is a standard duration used to assess internalization before the addition of antibiotics to eliminate extracellular bacteria, as we previously described in different works [PMID: 36668965, PMID: 35234505, PMID: 32324853, PMID: 30082478, PMID: 29600279, PMID: 23908480, PMID: 22689572]. Our intent was to measure invasion efficiency rather than long-term intracellular persistence, which indeed requires extended incubation (≥ 24 hours). We have revised the Methods section to clearly state that the invasion step was carried out for 2 hours (see lines 825-826).

Minor Comments

7. Line 45: Define CC₅₀ upon first mention.

As requested by the Reviewer, we have defined CC₅₀ as cytotoxic concentration 50% (see line 123).

8. Line 100: Replace "present" with "presents" or "presented" according to context.

As requested by the Reviewer, we have replaced "present" by "presents" (see line 117).

9. Line 127: Move the definition of FIC here from line 544.

As requested by the Reviewer, we have moved "the definition of FIC here from line 544 (see lines 144-145).

10. Lines 148-150: This section appears incomplete - please revise.

As requested by the Reviewer, we have completed this section (see line 168-171).

11. Lines 326-328: Content seems missing or unclear - revise.

As requested by the Reviewer, we have revised and made clear the sentence (see lines 388-391).

12. Line 490: Add "respectively" at the end. Also, clarify the rationale for using *P. aeruginosa* and why the inoculum was 5×10^5 instead of 1×10^6 .

As requested by the Reviewer, we have added "respectively" at the end (see lines 545-546), and we have clarified the rationale for using *P. aeruginosa* (see line 547) and the inoculum was 5×10^5 instead of 1×10^6 as its recommended by CLSI and EUCAST (see line 543).

13. Line 1023: Remove highlighted text.

As requested by the Reviewer, we have removed the highlighted text (see line 1122).

14. Introduction: It would be helpful to add a brief summary of previous studies on ENOblock's biological activities to better contextualize the current findings.

Thank you for your suggestion. We agree that providing background on ENOblock's previously reported biological activities would enhance the context and relevance of our study. Accordingly, we have revised the Introduction to include a brief summary of prior research highlighting ENOblock's roles, including its reported anti-cancer, anti-inflammatory, and metabolic effects (see lines 80-86).

15. Use of *A. baumannii* Ab: Redundant. Define abbreviation once, then use either *A. baumannii* or Ab consistently.

As requested by the Reviewer, we have defined the abbreviation of *Acinetobacter baumannii* one by "Ab" in the whole of the revised the manuscript.

16. Figure 4B: The legend appears incomplete - please revise.

Thank you for your comment. We have revised the sentence of the figure 4B legend to be sure that is complete.

17. Figure 7K: No statistical significance indicators are shown - clarify.

Thank you for your comment, we have added the statistical significance indicators (see figure 7K).

18. Controls in Membrane Permeability Assays: Ensure 1% DMSO (used in 100 μ M ENOblock samples) is added to controls for consistency.

We thank the Reviewer for this comment. We have clarified that the PBS used as negative control is combined with 1%DMSO (see lines 658-659).

19. Tables 1 and 2: Consider merging for clarity.

We thank the Reviewer for this comment. As recommended by the Reviewer 1, we have added the resistance mechanisms for colistin and carbapenems in the studied isolates. To maintain clarity, we chose to keep both tables separate.

20. EUCAST 2013 Breakpoints: Provide rationale for using this version over more recent updates.

This is a typo. We meant 2023 rather than 2013. We apologize for this mistake, which has been corrected in the revised manuscript (see line 543).

21. Binding Mode of ENOblock: Do the authors have any preliminary evidence or hypothesis regarding whether ENOblock binding to enolase is reversible, irreversible, or covalent?

We thank the Reviewer for this insightful question. While we have not yet performed experiments specifically designed to determine whether ENOblock binding to enolase is reversible or irreversible, our current data suggest a non-covalent and likely reversible interaction.

This hypothesis is supported by the following observations: (i) the interaction was detected under non-denaturing conditions, and (ii) the binding appears dose-dependent. Nonetheless, we recognize the importance of elucidating the precise binding mode and are planning further biophysical studies (e.g., surface plasmon resonance) to address this question in more detail.

Referee #3 (Remarks for Author):

This study identified the anticancer agent ENOblock as a novel enolase-targeting antibiotic class, which exhibits significant antibacterial activity against multidrug-resistant *Acinetobacter baumannii*. Additionally, the agent also displays significant synergistic effect with the last resort antibiotic colistin by targeting enolase. The new findings in this study repurposed ENOblock as alternative antibiotic candidate, and more importantly, revealed the potential of enolase as a promising bacterial target for new antimicrobials development. In overall, the manuscript is well executed and innovative. However, there are still some issues need improving prior to publication, as detailed below:

We are grateful to the Reviewer for the insightful comments on our paper. We appreciate the dedicated time and effort on our manuscript. We have been able to perform new experiments and to incorporate changes to reflect most of the suggestions provided by the Reviewer. We have highlighted with red colour all modifications within the manuscript. Here is a point-by-point response to the reviewer's comments and concerns.

1, The section spanning "ENOblock exhibits rapid permeabilization activity" to "ENOblock acts... through inhibition of enolase" requires expanded explanation. Specifically, cite established literature confirming enolase localization to the outer membrane and its role in membrane regulation in Gram-negative bacteria. This will bridge the observed permeabilization phenotype to enolase targeting.

We thank the Reviewer for this insightful suggestion. To be in line with the comment #4 of the Reviewer 1, we have expanded the Introduction section to provide supporting literature that confirms the presence of enolase on the outer membrane of Gram-negative bacteria and its proposed roles in membrane-associated functions.

Notably, previous studies have demonstrated that enolase can localize to the bacterial surface, functioning in adhesion, plasminogen binding, and interaction with host tissues [Ehinger et al., 2004; Pancholi, 2001]. Furthermore, enolase has been implicated in maintaining membrane dynamics and virulence of several species [Ayón-Núñez et al, 2018; Liu et al, 2021]. These findings support the hypothesis that ENOblock-induced permeabilization may be a consequence of its interaction with surface-associated enolase, thereby linking our observed phenotype to a plausible mechanistic target. Relevant citations have been added in the revised manuscript (see lines 68–78).

References

1. Ayón-Núñez DA, Fragoso G, Bobes RJ, Lacleste JP. Plasminogen-binding proteins as an evasion mechanism of the host's innate immunity in infectious diseases. *Biosci Rep*. 2018 Oct 2;38(5):BSR20180705.
2. Ehinger S, Schubert WD, Bergmann S, Hammerschmidt S. Plasmin(ogen) acquisition by pathogenic streptococci through surface enolase. *Blood* 2004; 104(5):1327–1333.
3. Liu H, Lei S, Jia L, Xia X, Sun Y, Jiang H, Zhu R, Li S, Qu G, Gu J, Sun C, Feng X, Han W, Langford PR, Lei L. Streptococcus suis serotype 2 enolase interaction with host brain microvascular endothelial cells and RPSA-induced apoptosis lead to loss of BBB integrity. *Vet. Res*. 2021; 52(1):30.
4. Pancholi V. Multifunctional α -enolase: its role in diseases. *Cellular and Molecular Life Sciences* 2001; 58(7):902–920.

2, Control for metabolic confounders in enolase-knockout studies:

The Δ eno strain exhibits reduced growth/adherence and increased ENOblock MIC. However, enolase deficiency disrupts central metabolism, making it unclear whether phenotypes stem from direct enolase inhibition or indirect metabolic collapse. To isolate these effects: Add a control cohort where Δ eno is supplemented with phosphoenolpyruvate (PEP), the metabolic product downstream of enolase. Compare ENOblock susceptibility and adherence between PEP-supplemented vs. unsupplemented Δ eno. If phenotypes persist only without PEP, metabolic disruption likely drives the effects.

We thank the Reviewer for this insightful suggestion. We have added a control in which the Δ eno strain was supplemented with 4 mM PEP. We have compared the adherence of *A. baumannii* in PEP-supplemented vs. unsupplemented Δ eno strain. Supplementation with PEP partially restored non-significantly the initial phenotype of the Ab ATCC 17978 strain, increasing adherence of the Δ eno strain to 33.8 +17.53%. These new data have been added to the revised manuscript (see lines 286-291 and 814-815, and figure 6E).

To further clarify whether the observed effects in the Δ eno strain reflect direct consequences of enolase loss in addition to broader metabolic perturbations, we also compared ENOblock susceptibility between PEP-supplemented and unsupplemented Δ eno strain. The MIC of ENOblock remained unchanged at 32 mg/L. These data suggest that the PEP supplementation does not affect ENOblock susceptibility in the Δ eno strain, and that the observed phenotypes in the mutant strain stem primarily from direct enolase inhibition. This new information has been included in the revised manuscript (see lines 233-234).

3, Incomplete in vivo validation:

Dose-response deficit: Efficacy was tested solely at 32 mg/L in *G. mellonella*. Include graded doses (e.g., 8, 16, 32 mg/L) to establish therapeutic range and dose dependency.

Thank you for this insightful comment. We agree that assessing the dose-response relationship is important to establish the therapeutic range and efficacy. In response, we have conducted additional experiments using graded doses of ENOblock (8, 16, and 32 mg/L) in the *Galleria mellonella* infection model. The results demonstrated a clear dose-dependent increase in survival, supporting the therapeutic relevance of ENOblock (see new data in figure 8D and lines 340-343).

Untested synergy: The reported in vitro synergy with colistin lacks in vivo validation. Co-administration of ENOblock and colistin in the *G. mellonella* model is essential to support clinical translatability.

Thank you for your observation. You are right. The *in vivo* experiment using the *Galleria mellonella* model was conducted with ENOblock monotherapy only, and not in combination with colistin. To address this, we performed an additional experiment using combination therapy: ENOblock at 16 and 32 mg/L was administered together with colistin at 0.12 mg/L one hour after *G. mellonella* infection with *A. baumannii*. We observed that the combination of ENOblock (16 and 32 mg/L) with colistin significantly increased larval survival to 75% and 100%, respectively, compared to colistin monotherapy at 0.12 mg/L (25% survival), and ENOblock monotherapy at 16 and 32 mg/L (28.57% and 57.14% survival, respectively). We have revised the text accordingly (see lines 322-331 and 932-941 and new figure 8E).

7th Oct 2025

Dear Dr. Smani,

Thank you for the submission of your revised manuscript to EMBO Molecular Medicine. I am pleased to inform you that we will be able to accept your manuscript pending the following final amendments:

- 1) Please implement all referee #1 and #2 suggestions. In particular, we agree with referee #1 comments about "novel antibiotic class". This phrasing should be removed from the title and from the rest of the text. Alternative title could be: "ENOblock synergizes with colistin to treat *Acinetobacter baumannii* infections".
- 2) Authors: E-mail correspondence to Mercedes de la Cruz could not be delivered. We received following message: This is an automated message to kindly inform you that Mercedes de la Cruz is no longer with our company. Please update their e-mail addresses and make sure to enter correct e-mail addresses for all authors in our submission system.
- 3) Figures: Please submit several Appendix figures (e.g. 5) as EV Figures. EV figures should be uploaded as individual, high resolution figure files and their legends placed after the main figure legends in the manuscript file under the heading "Expanded View Figure Legends". Please check "Author Guidelines" for more information.
<https://www.embopress.org/page/journal/17574684/authorguide#expandedview>
- 4) In the main manuscript file, please do the following:
 - Please address all comments suggested by our data editors listed below:
 - o Figure legends:
 1. Please note that the error bars are not defined in the legends of figures 2E, 4E, F.
 2. Please note that the exact p values are not provided in the legends of figures 3D, 4C, 5B, C; 6E, 7G-L.
 3. Please note that the scale bar needs to be defined for figure 3A.
 4. Please note that scale bar and its definition are missing for figure.
 - Limit keywords to max. 5.
 - Please make sure that all figure callouts are correct and in sequential order.
 - Author contributions: Please remove it from the manuscript and specify author contributions in our submission system. CRediT has replaced the traditional author contributions section because it offers a systematic machine-readable author contributions format that allows for more effective research assessment. You are encouraged to use the free text boxes beneath each contributing author's name to add specific details on the author's contribution. More information is available in our guide to authors:
<https://www.embopress.org/page/journal/17574684/authorguide#authorshipguidelines>
 - Indicate in legends number and nature of replicates and exact p= values, not a range, along with the statistical test used. To keep the figures "clear" some authors found providing an Appendix table Sx with all exact p-values preferable. You are welcome to do this if you want to.
 - Please include structured Methods section that includes a Reagents and Tools Table (should be uploaded as a separate file) followed by a Methods and Protocols section. More information on how to adhere to this format as well as downloadable templates (.docx) for the Reagents and Tools Table can be found in our author guidelines:
<https://www.embopress.org/page/journal/17574684/authorguide#structuredmethods>
 - An example of a paper with Structured Methods can be found here:
<https://www.embopress.org/doi/full/10.1038/s44320-024-00037-6#sec-4>
 - Rename "FIGURE LIST" heading to "Figure legends" for the main figure legends and move them to the end of the manuscript file.
 - Remove the numbering from the references.
- 5) Tables: Please rename Table S5 and Table S6 to Dataset EV1 and EV2, upload them as excel files with the legend in a separate tab and update their callouts in the main text.
- 6) Appendix: The rest of the supplementary figures and tables should be compiled in an Appendix and renamed to Appendix Figure S1 etc. and Appendix Table S1 etc. All figures should have legends placed below each figure. Appendix should be uploaded as PDF file with the table of contents with page numbers on the title page. Please also update Appendix Figure and Table callouts in the main text.
- 7) Synopsis: Every published paper now includes a 'Synopsis' to further enhance discoverability. Synopses are displayed on the journal webpage and are freely accessible to all readers. They include separate synopsis image and synopsis text.
 - Synopsis image: Please provide a visual abstract as a high-resolution jpeg file 550 px-wide x 300-600 pixels high to illustrate your article.
 - Synopsis text: Please provide a short standfirst (maximum of 300 characters, including space) as well as 2-5 one sentence bullet points that summarise the paper as a .doc file. Please write the bullet points to summarise the key NEW findings. They should be designed to be complementary to the abstract - i.e. not repeat the same text. We encourage inclusion of key acronyms and quantitative information (maximum of 30 words / bullet point). Please use the passive voice.
 - Please check your synopsis text and image before submission with your revised manuscript. Please be aware that in the proof stage minor corrections only are allowed (e.g., typos).
- 8) As part of the EMBO Publications transparent editorial process initiative (see our Editorial at <http://embomolmed.embopress.org/content/2/9/329>), EMBO Molecular Medicine will publish online a Review Process File (RPF)

to accompany accepted manuscripts. This file will be published in conjunction with your paper and will include the anonymous referee reports, your point-by-point response and all pertinent correspondence relating to the manuscript. Let us know whether you agree with the publication of the RPF and as here, if you want to remove or not any figures from it prior to publication. Please note that the Authors checklist will be published at the end of the RPF.

9) Please provide a point-by-point letter INCLUDING my comments as well as the reviewer's reports and your detailed responses (as Word file).

I look forward to reading a new revised version of your manuscript as soon as possible.

Yours sincerely,

Zeljko Durdevic

Zeljko Durdevic
Senior Editor
EMBO Molecular Medicine

*** Instructions to submit your revised manuscript ***

- 1) a .docx formatted version of the manuscript text (including Figure legends and tables)
- 2) Separate figure files*
- 3) supplemental information as Expanded View and/or Appendix. Please carefully check the authors guidelines for formatting Expanded view and Appendix figures and tables at <https://www.embopress.org/page/journal/17574684/authorguide#expandedview>
- 4) a letter INCLUDING the reviewer's reports and your detailed responses to their comments (as Word file).
- 5) The paper explained: EMBO Molecular Medicine articles are accompanied by a summary of the articles to emphasize the major findings in the paper and their medical implications for the non-specialist reader. Please provide a draft summary of your article highlighting
 - the medical issue you are addressing,
 - the results obtained and
 - their clinical impact.This may be edited to ensure that readers understand the significance and context of the research. Please refer to any of our published articles for an example.
- 6) Author contributions: the contribution of every author must be detailed in a separate section.
- 7) EMBO Molecular Medicine now requires a complete author checklist (<https://www.embopress.org/page/journal/17574684/authorguide>) to be submitted with all revised manuscripts. Please use the checklist as guideline for the sort of information we need WITHIN the manuscript. The checklist should only be filled with page

numbers were the information can be found. This is particularly important for animal reporting, antibody dilutions (missing) and exact values and n that should be indicated instead of a range.

8) Every published paper now includes a 'Synopsis' to further enhance discoverability. Synopses are displayed on the journal webpage and are freely accessible to all readers. They include a short stand first (maximum of 300 characters, including space) as well as 2-5 one sentence bullet points that summarise the paper. Please write the bullet points to summarise the key NEW findings. They should be designed to be complementary to the abstract - i.e. not repeat the same text. We encourage inclusion of key acronyms and quantitative information (maximum of 30 words / bullet point). Please use the passive voice. Please attach these in a separate file or send them by email, we will incorporate them accordingly.

You are also welcome to suggest a striking image or visual abstract to illustrate your article. If you do please provide a jpeg file 550 px-wide x 300-600px high.

9) A Conflict of Interest statement should be provided in the main text

10) Please note that we now mandate that all corresponding authors list an ORCID digital identifier. This takes <90 seconds to complete. We encourage all authors to supply an ORCID identifier, which will be linked to their name for unambiguous name identification.

Currently, our records indicate that the ORCID for your account is 0000-0001-9302-8384.

Link Not Available

11) Include a Reagents and Tools Table as part of the Methods section, which can be downloaded from our author guidelines (<https://www.embopress.org/page/journal/17574684/authorguide#structuredmethods>)

Photos 400-800 DPI

*Additional important information regarding figures and illustrations can be found at

<https://bit.ly/EMBOPressFigurePreparationGuideline>. See also figure legend preparation guidelines:

<https://www.embopress.org/page/journal/17574684/authorguide#figureformat>

***** Reviewer's comments *****

Referee #1 (Remarks for Author):

Overall the authors should be commended for their efforts to address the reviewers comments and as it stands I feel the study should be accepted but I do think some text changes need to be included to better reflect the novelty of the study.

- Unfortunately the points about the novelty have not been addressed in the opening part of my rebuttal. Specifically I stated "The title suggests a novel antibiotic class however what the criteria are for this claim are not clear particularly given targeting enolase is already a known mechanism of inhibiting bacterial growth (PMID: 31745118) and ENOblock is known to target enolase (and has been reported to inhibit the activity of E. coli Enolase PMID: 30714720)." Therefore if accepted claims to primacy or "novel class" need to be removed. Also need to be removed in Lines 27-28. PMID: 30714720/ Krucinska et al is cited again on line 90 but the fact that this study reports the inhibition of E. coli growth by ENOblock is ignored and obviously then this contradicts the following statement in line 91-92 "In this work, we report the identification of ENOblock, an anticancer drug, as a novel antibiotic class.". The impact of enolase inhibitors on bacterial cells needs to be outlined in detail in the introduction, this was not done in the revised version.

- The question on the essentiality of eno also needs to be addressed in much greater depth in the discussion and outlining as a limitation of the work. The rationale given is that compensatory metabolic adaptations may be occurring in the eno mutant, however surely these would have also occurred in the resistance assays if this were the case. This highlights a dichotomy that is now present in the study where the authors indicate that Ab cannot evolve resistance to ENOblock (421-424) but throughout the

study they reference the evidence for resistance that they have seen (Line 406-407, 413-414) as a means to explain some of their findings.

Minor

- Line 31 "in animal an infection model" Correct sentence structure.
- Line 110 rationale for ENOblock selection needs to be given particularly as it doesn't appear the data for the other 6 compounds is presented, which is fine the authors want to maintain the novelty of these hits for future studies but given this is a question every reader will have the rationale for selecting ENOblock must be clear. Did it perform better?
- Line 121: I think MIC50 and MIC90 are being misinterpreted here as they are not the concentration that is "effective for 50 and 90% of isolates tested" but the lowest concentration of an antimicrobial substance that is required to inhibit the growth of 50% of a specific microorganism's population within a defined period, typically 24 hours.
- Line 126: "Notably, the ENOblock MIC90 is three times lower than the cytotoxic concentration 50% (CC50) of ENOblock in HeLa and macrophage cells (Figure S1)," I think it would be much easier to follow if the exact numbers were included here as it is hard to rationalize the statement without these, for example the previous sentences states "the MIC50 and MIC90 for colistin were 256 and >256 mg/L, and for carbapenems were 16 and 64 mg/L", and based on Figure S1 the CC50= 112.7 mg/L for HeLA cells so how the "three times lower" evaluation is reached isn't clear.
- Line 132 and Figure 2A Thank you for updating the legend however it still needs more detail as just says CFU, CFU per what? If the initial inoculum was 5×10^5 CFU then how can some of the values read -5.5? I am assuming (as it doesn't say it in the legend) that the values presents are Log₁₀ CFU/mL?
- Line 164 Unique is not the correct term here, please change to "different".
- Figure 3A. This legend is now much clearer but which is 2X MIC being used for ENOblock and 1X MIC for all the others? How do the authors know that at 2 X MIC for the other antibiotics, particularly colistin that the morphology would be different and maybe closer to that seen for ENOblock? Figure 3C suggests this is the case indeed in addressing the original reviewer comment on this they have shown that at 1X MIC the impact on the cell is minimal so this is likely to be the case for the other antibiotics also. Therefore the discussion (Line 432) needs to highlight that the Enolase concentration used was a higher MIC than that use for any of the antibiotics.
- Line 187: According to EUCAST guidelines, the clinical breakpoint for colistin against *Acinetobacter baumannii* is {less than or equal to} 2 mg/L, how is 3.125 mg/L 5X MIC?
- Line 324 Alphafold not Alfafold
- The inclusion of mechanism on table 1 and 2 is great but can the source also be specified? Particularly as some are from other institutions.

Referee #2 (Comments on Novelty/Model System for Author):

1. The technical quality of the work is high. The experimental design is sound, the methodology is adequately detailed, and the statistical analyses appear appropriate to support the conclusions. Importantly, the authors have added new controls (e.g., in the SYTOX Green assay) and clarified ambiguous sections, which has significantly increased the robustness of the data. Minor issues remain regarding figure clarity and consistency (e.g., Figures S1, 2C, and 3A/3C), but these are correctable and do not detract from the overall quality.
2. The manuscript makes a novel contribution by identifying ENOblock, a compound with known anti-inflammatory and anti-cancer properties, as a potential antimicrobial agent against *A. baumannii*, a WHO-designated critical pathogen. The combination of high-throughput screening, mechanistic assays, and bioinformatic approaches represents an original and valuable angle. While ENOblock has been studied in other biomedical contexts, its repositioning as an antimicrobial agent against multidrug-resistant bacteria is both innovative and timely.
3. The medical relevance of this work is high. Multidrug-resistant *A. baumannii* infections remain a pressing global health problem with very limited therapeutic options. The identification of a compound with activity against this pathogen could have important translational implications. Although in vivo validation in mammalian models is still pending, the findings provide a strong basis for further preclinical development and potential therapeutic applications.
4. The choice of model systems is appropriate for the current stage of investigation. The primary focus on *A. baumannii* is well justified, and the use of *Galleria mellonella* offers a convenient and ethically acceptable in vivo model for initial proof-of-concept studies. The addition of a sequence alignment with human enolase strengthens confidence in the selectivity of the compound. That said, future studies in mammalian infection models (e.g., murine pneumonia, human airway organoids) would be important to validate efficacy and safety in a clinically relevant context. This does not represent a limitation of the present manuscript but rather a logical next step in the research pipeline.

The models employed (in vitro assays, *G. mellonella*) raise no particular ethical concerns.

Referee #2 (Remarks for Author):

The authors have significantly improved the manuscript in response to my and the other reviewers' comments. The additional experiments and clarifications have strengthened the scientific value of the work, and the study now provides a clearer and more convincing characterization of ENOblock's antimicrobial activity, particularly against *A. baumannii*. The revised introduction is more contextualized, the new control experiments increase the robustness of the data, and the supplementary materials now offer useful comparative insights.

Before final acceptance, however, I would like to draw attention to some minor issues that still require clarification or correction in order to improve accuracy, consistency, and readability:

1. Lines 31-32: Although I noted the other reviewer's comment, I think that the correct phrasing is "in an animal model" (in animal an infection model).
2. Lines 58-62: Use "are responsible" instead of "is responsible."
3. Figure 1: Please correct with "2464."
4. Lines 116-117: Add "respectively" at the end of the sequence.
5. Figure S1: The graph is unclear. How is it possible that HeLa cell viability values start at >100% at 0 concentration of ENOblock? This appears inconsistent, especially when compared with the figure shown on page 13 (Fig. 1) of the authors' point-by-point responses, where the data presentation is much clearer.
6. Figure 2C: In the second step, a tube with broth is shown, while the text states "plate." Please correct for consistency.
7. FIC definition: Thank you for including the definition; however, in Figure 2D the term "FICI" is used. Please harmonize terminology throughout the manuscript and figures.
8. Figures 3A vs. 3C: It remains unclear how long bacteria were exposed to the antibiotics (panel A). This should be clarified both in the Methods and in the figure legends. Moreover, the authors should more explicitly define the differences in experimental setup between Figure 3A (with ENOblock) and Figure 3C.

In summary, the manuscript has been considerably strengthened. If the authors address these remaining minor issues, I believe the work will be ready for acceptance.

Referee #3 (Remarks for Author):

Is suitable for publication

Referee #1 (Remarks for Author):

Overall the authors should be commended for their efforts to address the reviewers comments and as it stands I feel the study should be accepted but I do think some text changes need to be included to better reflect the novelty of the study.

We are grateful to the Reviewer for the insightful comments on our paper. We appreciate the dedicated time and effort on our manuscript. We have highlighted with red colour all modifications suggested by the Reviewer within the manuscript. Here is a point-by-point response to the reviewer's comments and concerns.

- **Unfortunately the points about the novelty have not been addressed in the opening part of my rebuttal. Specifically I stated "The title suggests a novel antibiotic class however what the criteria are for this claim are not clear particularly given targeting enolase is already a known mechanism of inhibiting bacterial growth (PMID: 31745118) and ENOblock is known to target enolase (and has been reported to inhibit the activity of E. coli Enolase PMID: 30714720)." Therefore if accepted claims to primacy or "novel class" need to be removed. Also need to be removed in Lines 27-28. PMID: 30714720/ Krucinska et al is cited again on line 90 but the fact that this study reports the inhibition of E. coli growth by ENOblock is ignored and obviously then this contradicts the following statement in line 91-92 "In this work, we report the identification of ENOblock, an anticancer drug, as a novel antibiotic class.". The impact of enolase inhibitors on bacterial cells needs to be outlined in detail in the introduction, this was not done in the revised version.**

As requested by the Reviewer, we have removed the term "novel antibiotic class" from the title and throughout the manuscript. The title has been changed to "***ENOblock synergizes with colistin to treat Acinetobacter baumannii infections***" (see lines 1, 27, 96-97 and 479).

Regarding the impact of enolase inhibitors on bacterial cells, this has been detailed in the introduction, as follows:

Specific enolase inhibitors, such as 2-aminothiazoles, disrupt bacterial ATP production and viability in *Mycobacterium tuberculosis* (Wescott et al., 2018). Three tropolone derivatives showing 53–78% enolase inhibition displayed antibacterial activity against major Gram-negative pathogens (*A. baumannii*, *E. coli*, *Pseudomonas aeruginosa*, and *K. pneumoniae*) with MICs of 11.3-45.2 mg/L, and induced filamentation in *E. coli*, suggesting effects on cell wall biosynthesis or division (Krucinska et al., 2019a). The natural inhibitor SF2312, produced by *Micromonospora*, showed limited activity (MIC 50→400 mg/L) but improved potency against *E. coli* and *Staphylococcus aureus*—though not

against *A. baumannii* and *P. aeruginosa*—when glucose-6-phosphate was added (Krucinska et al., 2019b). Additionally, PEIP-expressing bacteria exhibited growth attenuation and thinner cell walls due to impaired peptidoglycan synthesis in *Bacillus subtilis* (Zhang et al., 2022) (see lines 78–86).

- **The question on the essentiality of *eno* also needs to be addressed in much greater depth in the discussion and outlining as a limitation of the work. The rationale given is that compensatory metabolic adaptations may be occurring in the *eno* mutant, however surely these would have also occurred in the resistance assays if this were the case. This highlights a dichotomy that is now present in the study where the authors indicate that Ab cannot evolve resistance to ENOblock (421-424) but throughout the study they reference the evidence for resistance that they have seen (Line 406-407, 413-414) as a means to explain some of their findings.**

We thank the Reviewer for this insightful comment. We agree that the essentiality of enolase and the apparent inconsistency between the resistance observations and the claim of limited evolutionary warrant further clarification.

In the revised Discussion, we have expanded this section to better contextualize the essentiality of enolase and to explicitly acknowledge this as a limitation of our study. Specifically, we now discuss that enolase is widely reported as essential in several bacterial species, including *A. baumannii* (Pancholi, 2001). However, conditional or partial loss of enolase function may still occur under certain experimental conditions, allowing the bacteria to survive through compensatory metabolic adaptations (Hottes et al, 2013). These adaptations likely differ in nature and timescale between our constructed Δeno mutant and the resistance assays of this study, where short-term selective pressure might not allow for the same degree of metabolic compensation (see lines 460-466).

We have also revised the text to resolve the apparent dichotomy regarding resistance to ENOblock. Our results suggest that while stable genetic resistance was not observed under the tested conditions, transient phenotypic adaptations, potentially involving metabolic redirecting or stress responses, could affect susceptibility to ENOblock (see 425-428).

Minor

- **Line 31 "in animal an infection model" Correct sentence structure.**

Corrected (see line 31).

- **Line 110 rationale for ENOblock selection needs to be given particularly as it doesn't appear the data for the other 6 compounds is presented, which is fine the authors want to maintain the novelty of these hits for**

future studies but given this is a question every reader will have the rationale for selecting ENOblock must be clear. Did it perform better?

We thank the Reviewer for this comment. The rationale for selecting ENOblock has now been clarified in the revised manuscript, as follows:

ENOblock was chosen based on preliminary antimicrobial activity against *A. baumannii* and *E. coli* in our initial screens. While the data for the other six compounds are not presented in detail to preserve the novelty of these hits for future studies, ENOblock was prioritized because it demonstrated the most consistent *in vitro* activity across multiple assays, making it the most suitable candidate for detailed mechanistic and time-kill studies (see lines 114-115 and 557-559).

• Line 121: I think MIC50 and MIC90 are being misinterpreted here as they are not the concentration that is "effective for 50 and 90% of isolates tested" but the lowest concentration of an antimicrobial substance that is required to inhibit the growth of 50% of a specific microorganism's population within a defined period, typically 24 hours.

We thank the Reviewer for this clarification. We acknowledge that our previous description of MIC₅₀ and MIC₉₀ was imprecise. As correctly noted, MIC₅₀ and MIC₉₀ refer to the lowest concentrations of an antimicrobial agent required to inhibit the growth of 50% and 90% of the tested population of a specific microorganism within 24 hours, rather than the concentration "effective" for a percentage of isolates. We have revised the text accordingly in the revised manuscript (see lines 125-126).

• Line 126: "Notably, the ENOblock MIC90 is three times lower than the cytotoxic concentration 50% (CC50) of ENOblock in HeLa and macrophage cells (Figure S1),"I think it would be much easier to follow if the exact numbers were included here as it is hard to rationalize the statement without these, for example the previous sentences states "the MIC50 and MIC90 for colistin were 256 and >256 mg/L, and for carbapenems were 16 and 64 mg/L", and based on Figure S1 the CC50= 112.7 mg/L for HeLa cells so how the "three times lower" evaluation is reached isn't clear.

We thank the Reviewer for pointing this out. We agree that providing the exact numerical values improves clarity and allows readers to better assess the therapeutic window of ENOblock. In the revised manuscript, we have updated the text to include the precise values: the MIC₉₀ of ENOblock against *A. baumannii* was 32 mg/L, while the CC₅₀ in HeLa cells was 112.7 mg/L and in macrophages was 121.3 mg/L (Figure S1) (see lines 130-131).

• Line 132 and Figure 2A Thank you for updating the legend however it still needs more detail as just says CFU, CFU per what? If the initial inoculum

was 5×10^5 CFU then how can some of the values read -5.5? I am assuming (as it doesn't say it in the legend) that the values presents are Log₁₀ CFU/mL?

We thank the Reviewer for this comment and agree that the legend required more detail for clarity. The values presented in the figure represent Log₁₀ CFU/mL, and the initial inoculum was 5×10^5 CFU/mL. We have revised the figure legend to clearly indicate that the data are expressed as Log₁₀ CFU per mL and have included a note explaining how the bacterial concentrations are determined for each time point (see lines 1286 and 1287-1288).

- **Line 164 Unique is not the correct term here, please change to "different".**

As requested by the Reviewer, we have replaced “unique” by “different” (see line 169).

- **Figure 3A. This legend is now much clearer but which is 2X MIC being used for ENOblock and 1X MIC for all the others? How do the authors know that at 2 X MIC for the other antibiotics, particularly colistin that the morphology would be different and maybe closer to that seen for ENOblock? Figure 3C suggests this is the case indeed in addressing the original reviewer comment on this they have shown that at 1X MIC the impact on the cell is minimal so this is likely to be the case for the other antibiotics also. Therefore the discussion (Line 432) needs to highlight that the Enolase concentration used was a higher MIC than that use for any of the antibiotics.**

We thank the Reviewer for this insightful comment. We agree that the difference in MIC multiples used for ENOblock and the comparator antibiotics should be clearly highlighted. In the revised manuscript, we have clarified that ENOblock was tested at 2xMIC, whereas the other antibiotics were tested at 1xMIC in the morphology assays (see lines 169-170). In addition, we have updated the Discussion (Line 432-433) to explicitly note that the ENOblock concentration used was a higher MIC than that use for any of the antibiotics.

- **Line 187: According to EUCAST guidelines, the clinical breakpoint for colistin against *Acinetobacter baumannii* is {less than or equal to} 2 mg/L, how is 3.125 mg/L 5XMIC?**

We thank the Reviewer for this comment. The discrepancy arises because in our study the MIC of colistin against the tested *A. baumannii* strain was 0.625 mg/L under our experimental conditions, which was determined *in vitro* using standard broth microdilution assay. Therefore, 3.125 mg/L corresponds to 5 x the experimentally determined MIC for this specific strain, rather than the EUCAST clinical breakpoint.

- **Line324 Alphafold not Alfafold**

Corrected (see line 917).

- **The inclusion of mechanism on table 1 and 2 is great but can the source also be specified? Particularly as some are from other institutions.**

As requested by the Reviewer, we have included in the tables 1 and 2 the sources of clinical isolates.

Referee #2 (Comments on Novelty/Model System for Author):

1. The technical quality of the work is high. The experimental design is sound, the methodology is adequately detailed, and the statistical analyses appear appropriate to support the conclusions. Importantly, the authors have added new controls (e.g., in the SYTOX Green assay) and clarified ambiguous sections, which has significantly increased the robustness of the data. Minor issues remain regarding figure clarity and consistency (e.g., Figures S1, 2C, and 3A/3C), but these are correctable and do not detract from the overall quality.

2. The manuscript makes a novel contribution by identifying ENOblock, a compound with known anti-inflammatory and anti-cancer properties, as a potential antimicrobial agent against *A. baumannii*, a WHO-designated critical pathogen. The combination of high-throughput screening, mechanistic assays, and bioinformatic approaches represents an original and valuable angle. While ENOblock has been studied in other biomedical contexts, its repositioning as an antimicrobial agent against multidrug-resistant bacteria is both innovative and timely.

3. The medical relevance of this work is high. Multidrug-resistant *A. baumannii* infections remain a pressing global health problem with very limited therapeutic options. The identification of a compound with activity against this pathogen could have important translational implications. Although in vivo validation in mammalian models is still pending, the findings provide a strong basis for further preclinical development and potential therapeutic applications.

4. The choice of model systems is appropriate for the current stage of investigation. The primary focus on *A. baumannii* is well justified, and the use of *Galleria mellonella* offers a convenient and ethically acceptable in vivo model for initial proof-of-concept studies. The addition of a sequence alignment with human enolase strengthens confidence in the selectivity of the compound. That said, future studies in mammalian infection models (e.g., murine pneumonia, human airway organoids) would be important to validate efficacy and safety in a clinically relevant context. This does not represent a limitation of the present manuscript but rather a logical next step in the research pipeline.

The models employed (in vitro assays, *G. mellonella*) raise no particular ethical concerns.

We are grateful to the Reviewer for the insightful comments on our paper.

Referee #2 (Remarks for Author):

The authors have significantly improved the manuscript in response to my and the other reviewers' comments. The additional experiments and clarifications have strengthened the scientific value of the work, and the study now provides a clearer and more convincing characterization of ENOblock's antimicrobial activity, particularly against *A. baumannii*. The revised introduction is more contextualized, the new control experiments increase the robustness of the data, and the supplementary materials now offer useful comparative insights.

Before final acceptance, however, I would like to draw attention to some minor issues that still require clarification or correction in order to improve accuracy, consistency, and readability:

We appreciate the dedicated time and effort made by the Reviewer on our manuscript. We have highlighted with red colour all modifications suggested by the Reviewer within the manuscript. Here is a point-by-point response to the reviewer's comments and concerns.

1. Lines 31-32: Although I noted the other reviewer's comment, I think that the correct phrasing is "in an animal model" (in animal an infection model).

The Reviewer is right. We have corrected the phrase (see line 31).

2. Lines 58-62: Use "are responsible" instead of "is responsible."

As requested by the Reviewer, we have replaced "is" by "are".

3. Figure 1: Please correct with "2464."

Corrected (see lines 106).

4. Lines 116-117: Add "respectively" at the end of the sequence.

As requested by the Reviewer, we have added "respectively" at the end of the sequence (see line 121).

5. Figure S1: The graph is unclear. How is it possible that HeLa cell viability values start at >100% at 0 concentration of ENOblock? This appears inconsistent, especially when compared with the figure shown on page 13 (Fig. 1) of the authors' point-by-point responses, where the data presentation is much clearer.

We thank the Reviewer for this observation. The apparent HeLa cell viability at 0 mg/L ENOblock is 100%, while the subsequent four concentrations of ENOblock show viability values slightly above 100%. We believe these values cannot be directly compared with the figure shown on page 13 (Figure 1 of the authors' point-by-point responses) because the duration of the assay in Figure

S1 was 24 hours, whereas in Figure 1 of the point-by-point responses, it was only 2 hours.

6. Figure 2C: In the second step, a tube with broth is shown, while the text states "plate." Please correct for consistency.

Corrected.

7. FIC definition: Thank you for including the definition; however, in Figure 2D the term "FICI" is used. Please harmonize terminology throughout the manuscript and figures.

We thank the Reviewer for this comment. We have harmonized the terminology throughout the manuscript and figures.

8. Figures 3A vs. 3C: It remains unclear how long bacteria were exposed to the antibiotics (panel A). This should be clarified both in the Methods and in the figure legends. Moreover, the authors should more explicitly define the differences in experimental setup between Figure 3A (with ENOblock) and Figure 3C.

We have revised the manuscript according to the Reviewer comment. In the method, we have now split the fluorescent microscopy part into two separate experiments (BCP and SYTOX) to highlight the experimental setup difference (see lines 643-645, 653-662, 665 and 677-680).

In the figure legend, exposure times at 60 min was added in Figure 3A (see line 1304).

In summary, the manuscript has been considerably strengthened. If the authors address these remaining minor issues, I believe the work will be ready for acceptance.

Thank you.

21st Oct 2025

Dear Dr. Smani,

We are pleased to inform you that your manuscript is accepted for publication and is now being sent to our publisher to be included in the next available issue of EMBO Molecular Medicine.

Zeljko Durdevic
Senior Editor
EMBO Molecular Medicine
